# FASTER LAST-ITERATE CONVERGENCE OF POLICY OPTIMIZATION IN ZERO-SUM MARKOV GAMES

**Shicong Cen**
Carnegie Mellon University
`shicongc@andrew.cmu.edu`

**Yuejie Chi**
Carnegie Mellon University
`yuejiechi@cmu.edu`

**Simon S. Du**
University of Washington
`ssdu@cs.washington.edu`

**Lin Xiao**
Meta AI Research
`linx@fb.com`

## ABSTRACT

Multi-Agent Reinforcement Learning (MARL)—where multiple agents learn to interact in a shared dynamic environment—permeates across a wide range of critical applications. While there has been substantial progress on understanding the global convergence of policy optimization methods in single-agent RL, designing and analysis of efficient policy optimization algorithms in the MARL setting present significant challenges, which unfortunately, remain highly inadequately addressed by existing theory. In this paper, we focus on the most basic setting of competitive multi-agent RL, namely two-player zero-sum Markov games, and study equilibrium finding algorithms in both the infinite-horizon discounted setting and the finite-horizon episodic setting. We propose a single-loop policy optimization method with symmetric updates from both agents, where the policy is updated via the entropy-regularized optimistic multiplicative weights update (OMWU) method and the value is updated on a slower timescale. We show that, in the full-information tabular setting, the proposed method achieves a finite-time last-iterate linear convergence to the quantal response equilibrium of the regularized problem, which translates to a sublinear last-iterate convergence to the Nash equilibrium by controlling the amount of regularization. Our convergence results improve upon the best known iteration complexities, and lead to a better understanding of policy optimization in competitive Markov games.

**Keywords:** zero-sum Markov game, entropy regularization, policy optimization, global convergence, multiplicative updates

## 1 INTRODUCTION

Policy optimization methods (Williams, 1992; Sutton et al., 2000; Kakade, 2002; Peters and Schaal, 2008; Konda and Tsitsiklis, 2000), which cast sequential decision making as value maximization problems with regards to (parameterized) policies, have been instrumental in enabling recent successes of reinforcement learning (RL). See e.g., Schulman et al. (2015; 2017); Silver et al. (2016). Despite its empirical popularity, the theoretical underpinnings of policy optimization methods remain elusive until very recently. For single-agent RL problems, a flurry of recent works has made substantial progress on understanding the global convergence of policy optimization methods under the framework of Markov Decision Processes (MDP) (Agarwal et al., 2020; Bhandari and Russo, 2019; Mei et al., 2020; Cen et al., 2021a; Lan, 2022; Bhandari and Russo, 2020; Zhan et al., 2021; Khodadadian et al., 2021; Xiao, 2022). Despite the nonconcave nature of value maximization, (natural) policy gradient methods are shown to achieve global convergence at a sublinear rate (Agarwal et al., 2020; Mei et al., 2020) or even a linear rate in the presence of regularization (Mei et al., 2020; Cen et al., 2021a; Lan, 2022; Zhan et al., 2021) when the learning rate is constant.

---

Author are sorted alphabetically.

Moving beyond single-agent RL, Multi-Agent Reinforcement Learning (MARL) is the next frontier—where multiple agents learn to interact in a shared dynamic environment—permeating across critical applications such as multi-agent networked systems, autonomous vehicles, robotics, and so on. Designing and analysis of efficient policy optimization algorithms in the MARL setting present significant challenges and new desiderata, which unfortunately, remain highly inadequately addressed by existing theory.

## 1.1 POLICY OPTIMIZATION FOR COMPETITIVE RL

In this work, we focus on one of the most basic settings of competitive multi-agent RL, namely two-player zero-sum Markov games (Shapley, 1953), and study equilibrium finding algorithms in both the infinite-horizon discounted setting and the finite-horizon episodic setting. In particular, our designs gravitate around algorithms that are *single-loop*, *symmetric*, with *finite-time last-iterate* convergence to the Nash Equilibrium (NE) or Quantal Response Equilibrium (QRE) under bounded rationality, two prevalent solution concepts in game theory. These design principles naturally come up as a result of pursuing simple yet efficient algorithms: *single-loop* updates preclude sophisticated interleaving of rounds between agents; *symmetric* updates ensure no agent will compromise its rewards in the learning process, which can be otherwise exploited by a faster-updating opponent; in addition, asymmetric updates typically lead to one-sided convergence, i.e., only one of the agents is guaranteed to converge to the minimax equilibrium in a non-asymptotic manner, which is less desirable; moreover, *last-iterate convergence* guarantee absolves the need for agents to switch between learning and deployment; last but not least, it is desirable to converge as fast as possible, where the iteration complexities are *non-asymptotic* with clear dependence on salient problem parameters.

Substantial algorithmic developments have been made for finding equilibria in two-player zero-sum Markov games, where Dynamical Programming (DP) techniques have long been used as a fundamental building block, leading to prototypical iterative schemes such as Value Iteration (VI) (Shapley, 1953) and Policy Iteration (PI) (Van Der Wal, 1978; Patek and Bertsekas, 1999). Different from their single-agent counterparts, these methods require solving a two-player zero-sum matrix game for every state per iteration. A considerable number of recent works (Zhao et al., 2022; Alacaoglu et al., 2022; Cen et al., 2021b; Chen et al., 2021a) are based on these DP iterations, by plugging in various (gradient-based) solvers of two-player zero-sum matrix games. However, these methods are inherently nested-loop, which are less convenient to implement. In addition, PI-based methods are asymmetric and come with only one-sided convergence guarantees (Patek and Bertsekas, 1999; Zhao et al., 2022; Alacaoglu et al., 2022).

Going beyond nested-loop algorithms, single-loop policy gradient methods have been proposed recently for solving two-player zero-sum Markov games. Here, we are interested in finding an $\epsilon$-optimal NE or QRE in terms of the duality gap, i.e. the difference in the value functions when either of the agents deviates from the solution policy.

- For the infinite-horizon discounted setting, Daskalakis et al. (2020) demonstrated that the independent policy gradient method, with direct parameterization and asymmetric learning rates, finds an $\epsilon$-optimal NE within a polynomial number of iterations. Zeng et al. (2022) improved over this rate using an entropy-regularized policy gradient method with softmax parameterization and asymmetric learning rates. On the other end, Wei et al. (2021b) proposed an optimistic gradient descent ascent (OGDA) method (Rakhlin and Sridharan, 2013) with direct parameterization and symmetric learning rates,[1] which achieves a last-iterate convergence at a rather pessimistic iteration complexity.
- For the finite-horizon episodic setting, Zhang et al. (2022); Yang and Ma (2022) showed that the weighted average-iterate of the optimistic Follow-The-Regularized-Leader (FTRL) method, when combined with slow critic updates, finds an $\epsilon$-optimal NE in a polynomial number of iterations.

A more complete summary of prior results can be found in Table 1 and Table 2. In brief, while there have been encouraging progresses in developing computationally efficient policy gradient methods

---

[1]To be precise, Wei et al. (2021b) proved the average-iterate convergence of the duality gap, as well as the last-iterate convergence of the policy in terms of the Euclidean distance to the set of NEs, where it is possible to translate the latter last-iterate convergence to the duality gap (see Appendix. G). The resulting iteration complexity, however, is much worse than that of the average-iterate convergence in terms of the duality gap, with a problem-dependent constant that can scale pessimistically with salient problem parameters.

for solving zero-sum Markov games, achieving fast finite-time last-iterate convergence with single-loop and symmetric update rules remains a challenging goal.

## 1.2 OUR CONTRIBUTIONS

Motivated by the positive role of entropy regularization in enabling faster convergence of policy optimization in single-agent RL (Cen et al., 2021a; Lan, 2022) and two-player zero-sum games (Cen et al., 2021b), we propose a single-loop policy optimization algorithm for two-player zero-sum Markov games in both the infinite-horizon and finite-horizon settings. The proposed algorithm follows the style of actor-critic (Konda and Tsitsiklis, 2000), with the actor updating the policy via the entropy-regularized optimistic multiplicative weights update (OMWU) method (Cen et al., 2021b) and the critic updating the value function on a slower timescale. Both agents execute multiplicative and symmetric policy updates, where the learning rates are carefully selected to ensure a fast last-iterate convergence. In both the infinite-horizon and finite-horizon settings, we prove that the last iterate of the proposed method learns the optimal value function and converges at a linear rate to the unique QRE of the entropy-regularized Markov game, which can be further translated into finding the NE by setting the regularization sufficiently small.

- For the infinite-horizon discounted setting, the last iterate of our method takes at most

$$\widetilde{\mathcal{O}}\left(\frac{|\mathcal{S}|}{(1-\gamma)^4 \tau} \log \frac{1}{\epsilon}\right)$$

iterations for finding an $\epsilon$-optimal QRE under entropy regularization, where $\widetilde{\mathcal{O}}(\cdot)$ hides logarithmic dependencies. Here, $|\mathcal{S}|$ is the size of the state space, $\gamma$ is the discount factor, and $\tau$ is the regularization parameter. Moreover, this implies the last-iterate convergence with an iteration complexity of

$$\widetilde{\mathcal{O}}\left(\frac{|\mathcal{S}|}{(1-\gamma)^5 \epsilon}\right)$$

for finding an $\epsilon$-optimal NE.

- For the finite-horizon episodic setting, the last iterate of our method takes at most

$$\widetilde{\mathcal{O}}\left(\frac{H^2}{\tau} \log \frac{1}{\epsilon}\right)$$

iterations for finding an $\epsilon$-optimal QRE under entropy regularization, where $H$ is the horizon length. Similarly, this implies the last-iterate convergence with an iteration complexity of

$$\widetilde{\mathcal{O}}\left(\frac{H^3}{\epsilon}\right)$$

for finding an $\epsilon$-optimal NE.

Detailed comparisons between the proposed method and prior arts are provided in Table 1 and Table 2. To the best of our knowledge, this work presents the first method that is simultaneously single-loop, symmetric, and achieves fast finite-time last-iterate convergence in terms of the duality gap in both infinite-horizon and finite-horizon settings. From a technical perspective, the infinite-horizon discounted setting is in particular challenging, where ours is the first single-loop algorithm that guarantees an iteration complexity of $\widetilde{\mathcal{O}}(1/\epsilon)$ for last-iterate convergence in terms of the duality gap, with clear and improved dependencies on other problem parameters in the meantime. In contrast, several existing works introduce additional problem-dependent constants (Daskalakis et al., 2020; Wei et al., 2021b; Zeng et al., 2022) in the iteration complexity, which can scale rather pessimistically—sometimes even exponentially—with problem dimensions (Li et al., 2021).

Our technical developments require novel ingredients that deviate from prior tools such as error propagation analysis for Bellman operators (Perolat et al., 2015; Patek and Bertsekas, 1999) from a dynamic programming perspective, as well as the gradient dominance condition (Daskalakis et al., 2020; Zeng et al., 2022) from a policy optimization perspective. Importantly, at the core of our analysis lies a carefully-designed one-step error contraction bound for policy learning, together with a set of recursive error bounds for value learning, all of which tailored to the non-Euclidean OMWU update rules that have not been well studied in the setting of Markov games.

| Solution type | Reference | Iteration complexity | Single loop | Symmetric | Last-iterate convergence |
|---|---|---|---|---|---|
| $\epsilon$-NE | **PI-based Methods** **Zhao et al. (2022)** **Alacaoglu et al. (2022)** | $\widetilde{\mathcal{O}}\left(\frac{\lVert 1/\rho \rVert_\infty}{(1-\gamma)^3 \epsilon}\right)^*$ | ✗ | ✗ | ✓ |
| | **VI-based Methods** **Cen et al. (2021b)** **Chen et al. (2021a)** | $\widetilde{\mathcal{O}}\left(\frac{1}{(1-\gamma)^3 \epsilon}\right)$ | ✗ | ✓ | ✓ |
| | **Daskalakis et al. (2020)** | Polynomial* | ✓ | ✗ | ✗ |
| | **Zeng et al. (2022)** | $\widetilde{\mathcal{O}}\left(\frac{\lvert\mathcal{S}\rvert^2 \lVert 1/\rho\rVert_\infty^5}{(1-\gamma)^{14} c^4 \epsilon^3}\right)^*$ | ✓ | ✗ | ✓ |
| | **Wei et al. (2021b)** | $\widetilde{\mathcal{O}}\left(\frac{\lvert\mathcal{S}\rvert^3}{(1-\gamma)^9 \epsilon^2}\right)$ | ✓ | ✓ | ✗ |
| | | $\widetilde{\mathcal{O}}\left(\frac{\lvert\mathcal{S}\rvert^5 (\lvert\mathcal{A}\rvert+\lvert\mathcal{B}\rvert)^{1/2}}{(1-\gamma)^{16} c^4 \epsilon^2}\right)$ | ✓ | ✓ | ✓ |
| | **This Work** | $\widetilde{\mathcal{O}}\left(\frac{\lvert\mathcal{S}\rvert}{(1-\gamma)^5 \epsilon}\right)$ | ✓ | ✓ | ✓ |
| $\epsilon$-QRE | **VI-based Methods** **Cen et al. (2021b)** | $\widetilde{\mathcal{O}}\left(\frac{1}{(1-\gamma)^3} \log^2 \frac{1}{\epsilon}\right)$ | ✗ | ✓ | ✓ |
| | **Zeng et al. (2022)** | $\widetilde{\mathcal{O}}\left(\frac{\lvert\mathcal{S}\rvert^2 \lVert 1/\rho\rVert_\infty^5}{(1-\gamma)^{11} c^4 \tau^3} \log \frac{1}{\epsilon}\right)^*$ | ✓ | ✗ | ✓ |
| | **This Work** | $\widetilde{\mathcal{O}}\left(\frac{\lvert\mathcal{S}\rvert}{(1-\gamma)^4 \tau} \log \frac{1}{\epsilon}\right)$ | ✓ | ✓ | ✓ |

Table 1: Comparison of policy optimization methods for finding an $\epsilon$-optimal NE or QRE of two-player zero-sum discounted Markov games in terms of the duality gap. Note that $*$ implies one-sided convergence, i.e., only one of the agents is guaranteed to achieve finite-time convergence to the equilibrium. Here, $c > 0$ refers to some problem-dependent constant. For simplicity and a fair comparison, we replace various notions of concentrability coefficient and distribution mismatch coefficient with a crude upper bound $\lVert 1/\rho\rVert_\infty$, where $\rho$ is the initial state distribution.

| Solution type | Reference | Iteration complexity | Single loop | Symmetric | Last-iterate convergence |
|---|---|---|---|---|---|
| $\epsilon$-NE | **Zhang et al. (2022)** **OFTRL** | $\widetilde{\mathcal{O}}\left(\frac{H^{28/5}}{\epsilon^{6/5}}\right)$ | ✓ | ✓ | ✗ |
| | **Zhang et al. (2022)** **modified OFTRL** | $\widetilde{\mathcal{O}}\left(\frac{H^4}{\epsilon}\right)$ | ✓ | ✓ | ✗ |
| | **Yang and Ma (2022)** **OFTRL** | $\widetilde{\mathcal{O}}\left(\frac{H^5}{\epsilon}\right)$ | ✓ | ✓ | ✗ |
| | **This Work** | $\widetilde{\mathcal{O}}\left(\frac{H^3}{\epsilon}\right)$ | ✓ | ✓ | ✓ |
| $\epsilon$-QRE | **This Work** | $\widetilde{\mathcal{O}}\left(\frac{H^2}{\tau} \log \frac{1}{\epsilon}\right)$ | ✓ | ✓ | ✓ |

Table 2: Comparison of policy optimization methods for finding an $\epsilon$-optimal NE or QRE of two-player zero-sum episodic Markov games in terms of the duality gap.

## 1.3 RELATED WORKS

**Learning in two-player zero-sum matrix games.** Freund and Schapire (1999) showed that the average iterate of Multiplicative Weight Update (MWU) method converges to NE at a rate of $\mathcal{O}(1/\sqrt{T})$, which in principle holds for many other no-regret algorithms as well. Daskalakis et al. (2011) deployed the excessive gap technique of Nesterov and improved the convergence rate to $\mathcal{O}(1/T)$, which is achieved later by (Rakhlin and Sridharan, 2013) with a simple modification of MWU method, named Optimistic Mirror Descent (OMD) or more commonly, OMWU. Moving beyond average-iterate convergence, Bailey and Piliouras (2018) demonstrated that MWU updates, despite converging in an ergodic manner, diverge from the equilibrium. Daskalakis and Panageas (2018); Wei et al. (2021a) explored the last-iterate convergence guarantee of OMWU, as-

suming uniqueness of NE. Cen et al. (2021b) established linear last-iterate convergence of entropy-regularized OMWU without uniqueness assumption. Sokota et al. (2022) showed that optimistic update is not necessary for achieving linear last-iterate convergence in the presence of regularization, albeit with a more strict restriction on the step size.

**Learning in two-player zero-sum Markov games.** In addition to the aforementioned works on policy optimization methods (policy-based methods) for two-player zero-sum Markov games (cf. Table 1 and Table 2), a growing body of works have developed model-based methods (Liu et al., 2021; Zhang et al., 2020; Li et al., 2022) and value-based methods (Bai and Jin, 2020; Bai et al., 2020; Chen et al., 2021b; Jin et al., 2021; Sayin et al., 2021; Xie et al., 2020), with a primary focus on learning NE in a sample-efficient manner. Our work, together with prior literatures on policy optimization, focuses instead on learning NE in a computation-efficient manner assuming full-information.

**Entropy regularization in RL and games.** Entropy regularization is a popular algorithmic idea in RL (Williams and Peng, 1991) that promotes exploration of the policy. A recent line of works (Mei et al., 2020; Cen et al., 2021a; Lan, 2022; Zhan et al., 2021) demonstrated that incorporating entropy regularization provably accelerates policy optimization in single-agent MDPs by enabling fast linear convergence. While the positive role of entropy regularization is also verified in various game-theoretic settings, e.g., two-player zero-sum matrix games (Cen et al., 2021b), zero-sum polymatrix games (Leonardos et al., 2021), and potential games (Cen et al., 2022), it remains highly unexplored the interplay between entropy regularization and policy optimization in Markov games with only a few exceptions (Zeng et al., 2022).

## 1.4 NOTATIONS

We denote the probability simplex over a set $\mathcal{A}$ by $\Delta(\mathcal{A})$. We use bracket with subscript to index the entries of a vector or matrix, e.g., $[x]_a$ for $a$-th element of a vector $x$, or simply $x(a)$ when it is clear from the context. Given two distributions $x, y \in \Delta(\mathcal{A})$, the Kullback-Leibler (KL) divergence from $y$ to $x$ is denoted by $\mathsf{KL}\big(x \,\|\, y\big) = \sum_{a \in \mathcal{A}} x(a)(\log x(a) - \log y(a))$. Finally, we denote by $\|A\|_\infty$ the maximum entrywise absolute value of a matrix $A$, i.e., $\|A\|_\infty = \max_{i,j} |A_{i,j}|$.

## 2 ALGORITHM AND THEORY: THE INFINITE-HORIZON SETTING

### 2.1 PROBLEM FORMULATION

**Two-player zero-sum discounted Markov game.** A two-player zero-sum discounted Markov game is defined by a tuple $\mathcal{M} = (\mathcal{S}, \mathcal{A}, \mathcal{B}, P, r, \gamma)$, with finite state space $\mathcal{S}$, finite action spaces of the two players $\mathcal{A}$ and $\mathcal{B}$, reward function $r : \mathcal{S} \times \mathcal{A} \times \mathcal{B} \to [0, 1]$, transition probability kernel $P : \mathcal{S} \times \mathcal{A} \times \mathcal{B} \to \Delta(\mathcal{S})$ and discount factor $0 \le \gamma < 1$. The action selection rule of the max player (resp. the min player) is represented by $\mu : \mathcal{S} \to \Delta(\mathcal{A})$ (resp. $\nu : \mathcal{S} \to \Delta(\mathcal{B})$), where the probability of selecting action $a \in \mathcal{A}$ (resp. $b \in \mathcal{B}$) in state $s \in \mathcal{S}$ is specified by $\mu(a|s)$ (resp. $\nu(b|s)$). The probability of transitioning from state $s$ to a new state $s'$ upon selecting the action pair $(a, b) \in \mathcal{A}, \mathcal{B}$ is given by $P(s'|s, a, b)$.

**Value function and Q-function.** For a given policy pair $\mu, \nu$, the state value of $s \in \mathcal{S}$ is evaluated by the expected discounted sum of rewards with initial state $s_0 = s$:

$$\forall s \in \mathcal{S}: \qquad V^{\mu,\nu}(s) = \mathbb{E}\left[\sum_{t=0}^{\infty} \gamma^t r(s_t, a_t, b_t) \big| s_0 = s\right], \qquad (1)$$

the quantity the max player seeks to maximize while the min player seeks to minimize. Here, the trajectory $(s_0, a_0, b_0, s_1, \cdots)$ is generated according to $a_t \sim \mu(\cdot|s_t)$, $b_t \sim \nu(\cdot|s_t)$ and $s_{t+1} \sim P(\cdot|s_t, a_t, b_t)$. Similarly, the $Q$-function $Q^{\mu,\nu}(s, a, b)$ evaluates the expected discounted cumulative reward with initial state $s$ and initial action pair $(a, b)$:

$$\forall (s, a, b) \in \mathcal{S} \times \mathcal{A} \times \mathcal{B}: \qquad Q^{\mu,\nu}(s, a, b) = \mathbb{E}\left[\sum_{t=0}^{\infty} \gamma^t r(s_t, a_t, b_t) \big| s_0 = s, a_0 = a, b_0 = b\right]. \quad (2)$$

For notation simplicity, we denote by $Q^{\mu,\nu}(s) \in \mathbb{R}^{|\mathcal{A}| \times |\mathcal{B}|}$ the matrix $[Q^{\mu,\nu}(s, a, b)]_{(a,b) \in \mathcal{A} \times \mathcal{B}}$, so that

$$\forall s \in \mathcal{S}: \qquad V^{\mu,\nu}(s) = \mu(s)^\top Q^{\mu,\nu}(s)\nu(s).$$

Shapley (1953) proved the existence of a policy pair $(\mu^\star, \nu^\star)$ that solves the min-max problem $\max_\mu \min_\nu V^{\mu,\nu}(s)$ for all $s \in \mathcal{S}$ simultaneously, and that the mini-max value is unique. A set of such optimal policy pair $(\mu^\star, \nu^\star)$ is called the Nash equilibrium (NE) to the Markov game.

**Entropy regularized two-player zero-sum Markov game.** Entropy regularization is shown to provably accelerate convergence in single-agent RL (Geist et al., 2019; Mei et al., 2020; Cen et al., 2021a) and facilitate the analysis in two-player zero-sum matrix games (Cen et al., 2021b) as well as Markov games (Cen et al., 2021b; Zeng et al., 2022). The entropy-regularized value function $V_\tau^{\mu,\nu}(s)$ is defined as

$$\forall s \in \mathcal{S}: \qquad V_\tau^{\mu,\nu}(s) = \mathbb{E}\left[\sum_{t=0}^\infty \gamma^t \Big(r(s_t, a_t, b_t) - \tau \log \mu(a_t|s_t) + \tau \log \nu(b_t|s_t)\Big) \Big| s_0 = s\right], \tag{3}$$

where $\tau \geq 0$ is the regularization parameter. Similarly, the regularized $Q$-function $Q_\tau^{\mu,\nu}$ is given by

$$\forall (s, a, b) \in \mathcal{S} \times \mathcal{A} \times \mathcal{B}: \qquad Q_\tau^{\mu,\nu}(s) = r(s, a, b) + \gamma \mathbb{E}_{s' \sim P(\cdot|s,a,b)}\left[V_\tau^{\mu,\nu}(s')\right]. \tag{4}$$

It is known that (Cen et al., 2021b) there exists a unique pair of policy $(\mu_\tau^\star, \nu_\tau^\star)$ that solves the min-max entropy-regularized problem $\max_\mu \min_\nu V_\tau^{\mu,\nu}(s)$, or equivalently

$$\max_\mu \min_\nu \mu(s)^\top Q_\tau^{\mu,\nu}(s)\nu(s) + \tau \mathcal{H}\big(\mu(s)\big) - \tau \mathcal{H}\big(\nu(s)\big) \tag{5}$$

for all $s \in \mathcal{S}$, and we call $(\mu_\tau^\star, \nu_\tau^\star)$ the quantal response equilibrium (QRE) (McKelvey and Palfrey, 1995) to the entropy-regularized Markov game. We denote the associated regularized value function and Q-function by $V_\tau^\star(s) = V_\tau^{\mu_\tau^\star, \nu_\tau^\star}(s)$ and $Q_\tau^\star(s, a, b) = Q_\tau^{\mu_\tau^\star, \nu_\tau^\star}(s, a, b)$.

**Goal.** We seek to find an $\epsilon$-optimal QRE or $\epsilon$-QRE (resp. $\epsilon$-optimal NE or $\epsilon$-NE) $\zeta = (\mu, \nu)$ which satisfies

$$\max_{s \in \mathcal{S}, \mu', \nu'} \left(V_\tau^{\mu',\nu}(s) - V_\tau^{\mu,\nu'}(s)\right) \leq \epsilon \tag{6}$$

(resp. $\max_{s \in \mathcal{S}, \mu', \nu'} \left(V^{\mu',\nu}(s) - V^{\mu,\nu'}(s)\right) \leq \epsilon$) in a computationally efficient manner. In truth, the solution concept of $\epsilon$-QRE provides an approximation of $\epsilon$-NE with appropriate choice of the regularization parameter $\tau$. Basic calculations tell us that

$$V^{\mu',\nu}(s) - V^{\mu,\nu'}(s) = \left(V_\tau^{\mu',\nu}(s) - V_\tau^{\mu,\nu'}(s)\right) + \left(V^{\mu',\nu}(s) - V_\tau^{\mu',\nu}(s)\right) - \left(V^{\mu,\nu'}(s) - V_\tau^{\mu,\nu'}(s)\right)$$

$$\leq V_\tau^{\mu',\nu}(s) - V_\tau^{\mu,\nu'}(s) + \frac{\tau(\log |\mathcal{A}| + \log |\mathcal{B}|)}{1 - \gamma},$$

which guarantees that an $\epsilon/2$-QRE is an $\epsilon$-NE as long as $\tau \leq \frac{(1-\gamma)\epsilon}{2(\log |\mathcal{A}| + \log |\mathcal{B}|)}$. For technical convenience, we assume $\tau \leq \frac{1}{\max\{1, \log |\mathcal{A}| + \log |\mathcal{B}|\}}$ throughout the paper.

**Additional notation.** For notation convenience, we denote by $\zeta$ the concatenation of a policy pair $\mu$ and $\nu$, i.e., $\zeta = (\mu, \nu)$. The QRE to the regularized problem is denoted by $\zeta_\tau^\star = (\mu_\tau^\star, \nu_\tau^\star)$. We use shorthand notation $\mu(s)$ and $\nu(s)$ to denote $\mu(\cdot|s)$ and $\nu(\cdot|s)$. In addition, we write $\mathsf{KL}\big(\mu(s) \, \| \, \mu'(s)\big)$ and $\mathsf{KL}\big(\nu(s) \, \| \, \nu'(s)\big)$ as $\mathsf{KL}_s\big(\mu \, \| \, \mu'\big)$ and $\mathsf{KL}_s\big(\nu \, \| \, \nu'\big)$, and let

$$\mathsf{KL}_s\big(\zeta \, \| \, \zeta'\big) = \mathsf{KL}_s\big(\mu \, \| \, \mu'\big) + \mathsf{KL}_s\big(\nu \, \| \, \nu'\big).$$

By a slight abuse of notation, $\mathsf{KL}_\rho\big(\zeta \, \| \, \zeta'\big)$ denotes $\mathbb{E}_{s \sim \rho}\big[\mathsf{KL}_s\big(\zeta \, \| \, \zeta'\big)\big]$ for $\rho \in \Delta(\mathcal{S})$.

## 2.2 SINGLE-LOOP ALGORITHM DESIGN

In this section, we propose a single-loop policy optimization algorithm for finding the QRE of the entropy-regularized Markov game, which is generalized from the entropy-regularized OMWU method (Cen et al., 2021b) for solving entropy-regularized matrix games, with a careful orchestrating of the policy update and the value update.

---

**Algorithm 1:** Entropy-regularized OMWU for Discounted Two-player Zero-sum Markov Game

---

1  **Input:** Regularization parameter $\tau > 0$, learning rate for policy update $\eta > 0$, learning rate for value update $\{\alpha_t\}_{t=1}^{\infty}$.

2  **Initialization:** Set $\mu^{(0)}, \bar{\mu}^{(0)}, \nu^{(0)}$ and $\bar{\nu}^{(0)}$ as uniform policies; and set

$$Q^{(0)} = 0, \quad V^{(0)} = \tau(\log|\mathcal{A}| - \log|\mathcal{B}|).$$

3  **for** $t = 0, 1, \cdots$ **do**

4     **for** *all* $s \in \mathcal{S}$ **do in parallel**

5         When $t \geq 1$, update policy pair $\zeta^{(t)}(s)$ as:

$$\begin{cases} \mu^{(t)}(a|s) \propto \mu^{(t-1)}(a|s)^{1-\eta\tau} \exp(\eta[Q^{(t)}(s)\bar{\nu}^{(t)}(s)]_a) \\ \nu^{(t)}(b|s) \propto \nu^{(t-1)}(b|s)^{1-\eta\tau} \exp(-\eta[Q^{(t)}(s)^\top\bar{\mu}^{(t)}(s)]_b) \end{cases} . \tag{9a}$$

6         Update policy pair $\bar{\zeta}^{(t+1)}(s)$ as:

$$\begin{cases} \bar{\mu}^{(t+1)}(a|s) \propto \mu^{(t)}(a|s)^{1-\eta\tau} \exp(\eta[Q^{(t)}(s)\bar{\nu}^{(t)}(s)]_a) \\ \bar{\nu}^{(t+1)}(b|s) \propto \nu^{(t)}(b|s)^{1-\eta\tau} \exp(-\eta[Q^{(t)}(s)^\top\bar{\mu}^{(t)}(s)]_b) \end{cases} . \tag{9b}$$

7         Update $Q^{(t+1)}(s)$ and $V^{(t+1)}(s)$ as

$$\begin{cases} Q^{(t+1)}(s,a,b) &= r(s,a,b) + \gamma\mathbb{E}_{s'\sim P(\cdot|s,a,b)}\left[V^{(t)}(s')\right] \\ V^{(t+1)}(s) &= (1-\alpha_{t+1})V^{(t)}(s) \\ &\quad + \alpha_{t+1}\left[\bar{\mu}^{(t+1)}(s)^\top Q^{(t+1)}(s)\bar{\nu}^{(t+1)}(s) + \tau\mathcal{H}\left(\bar{\mu}^{(t+1)}(s)\right) - \tau\mathcal{H}\left(\bar{\nu}^{(t+1)}(s)\right)\right] \end{cases} . \tag{10}$$

---

**Review: entropy-regularized OMWU for two-player zero-sum matrix games.** We briefly review the algorithm design of entropy-regularized OMWU method for two-player zero-sum matrix game (Cen et al., 2021b). The problem of interest can be described as

$$\max_{\mu\in\Delta(\mathcal{A})} \min_{\nu\in\Delta(\mathcal{B})} \mu^\top A\nu + \tau\mathcal{H}(\mu) - \tau\mathcal{H}(\nu), \tag{7}$$

where $A \in \mathbb{R}^{|\mathcal{A}|\times|\mathcal{B}|}$ is the payoff matrix of the game. The update rule of entropy-regularized OMWU with learning rate $\eta > 0$ is defined as follows: $\forall a \in \mathcal{A}, b \in \mathcal{B}$,

$$\begin{cases} \mu^{(t)}(a) \propto \mu^{(t-1)}(a)^{1-\eta\tau} \exp(\eta[A\bar{\nu}^{(t)}]_a) \\ \nu^{(t)}(b) \propto \nu^{(t-1)}(b)^{1-\eta\tau} \exp(-\eta[A^\top\bar{\mu}^{(t)}]_b) \end{cases} , \tag{8a}$$

$$\begin{cases} \bar{\mu}^{(t+1)}(a) \propto \mu^{(t)}(a)^{1-\eta\tau} \exp(\eta[A\bar{\nu}^{(t)}]_a) \\ \bar{\nu}^{(t+1)}(b) \propto \nu^{(t)}(b)^{1-\eta\tau} \exp(-\eta[A^\top\bar{\mu}^{(t)}]_b) \end{cases} . \tag{8b}$$

We remark that the update rule can be alternatively motivated from the perspective of natural policy gradient (Kakade, 2002; Cen et al., 2021a) or mirror descent (Lan, 2022; Zhan et al., 2021) with optimistic updates. In particular, the midpoint $(\bar{\mu}^{(t+1)}, \bar{\nu}^{(t+1)})$ serves as a prediction of $(\mu^{(t+1)}, \nu^{(t+1)})$ by running one step of mirror descent. Cen et al. (2021b) established that the last iterate of entropy-regularized OMWU converges to the QRE of the matrix game (7) at a linear rate $(1 - \eta\tau)^t$, as long as the step size $\eta$ is no larger than $\min\left\{\frac{1}{2\|A\|_\infty+2\tau}, \frac{1}{4\|A\|_\infty}\right\}$.

**Single-loop algorithm for two-player zero-sum Markov games.** In view of the similarity in the problem formulations of (5) and (7), it is tempting to apply the aforementioned method to the Markov game in a state-wise manner, where the $Q$-function assumes the role of the payoff matrix. It is worth noting, however, that $Q$-function depends on the policy pair $\zeta = (\mu, \nu)$ and is hence changing concurrently with the update of the policy pair. We take inspiration from Bai et al. (2020); Wei et al. (2021b) and equip the entropy-regularized OMWU method with the following update rule that iteratively approximates the value function in an actor-critic fashion:

$$Q^{(t+1)}(s,a,b) = r(s,a,b) + \gamma\mathbb{E}_{s'\sim P(\cdot|s,a,b)}\left[V^{(t)}(s')\right],$$

where $V^{(t+1)}$ is updated as a convex combination of the previous $V^{(t)}$ and the regularized game value induced by $Q^{(t+1)}$ as well as the policy pair $\bar{\zeta}^{(t+1)} = (\bar{\mu}^{(t+1)}, \bar{\nu}^{(t+1)})$:

$$V^{(t+1)}(s) = (1 - \alpha_{t+1})V^{(t)}(s)$$
$$+ \alpha_{t+1}\left[\bar{\mu}^{(t+1)}(s)^\top Q^{(t+1)}(s)\bar{\nu}^{(t+1)}(s) + \tau\mathcal{H}\big(\bar{\mu}^{(t+1)}(s)\big) - \tau\mathcal{H}\big(\bar{\nu}^{(t+1)}(s)\big)\right]. \quad (11)$$

The update of $V$ becomes more conservative with a smaller learning rate $\alpha_t$, hence stabilizing the update of policies. However, setting $\alpha_t$ too small slows down the convergence of $V$ to $V_\tau^\star$. A key novelty—suggested by our analysis—is the choice of the constant learning rates $\alpha := \alpha_t = \eta\tau$ which updates at a slower timescale than the policy due to $\tau < 1$. This is in sharp contrast to the vanishing sequence $\alpha_t = \frac{2/(1-\gamma)+1}{2/(1-\gamma)+t}$ adopted in Wei et al. (2021b), which is essential in their analysis but inevitably leads to a much slower convergence. We summarize the detailed procedure in Algorithm 1. Last but not least, it is worth noting that the proposed method access the reward via "first-order information", i.e., either agent can only update its policy with the marginalized value function $Q(s)\nu(s)$ or $Q(s)^\top\mu(s)$. Update rules of this kind are instrumental in breaking the curse of multi-agents in the sample complexity when adopting sample-based estimates in (10), as we only need to estimate the marginalized Q-function rather than its full form (Li et al., 2022; Chen et al., 2021a).

## 2.3 THEORETICAL GUARANTEES

Below we present our main results concerning the last-iterate convergence of Algorithm 1 for solving entropy-regularized two-player zero-sum Markov games in the infinite-horizon discounted setting. The proof is postponed to Appendix A.

**Theorem 1.** *Setting $0 < \eta \leq \frac{(1-\gamma)^3}{32000|\mathcal{S}|}$ and $\alpha_t = \eta\tau$, it holds for all $t \geq 0$ that*

$$\max\left\{\frac{1}{|\mathcal{S}|}\sum_{s\in\mathcal{S}}\mathsf{KL}_s\big(\zeta_\tau^\star \| \zeta^{(t)}\big), \frac{1}{2|\mathcal{S}|}\sum_{s\in\mathcal{S}}\mathsf{KL}_s\big(\zeta_\tau^\star \| \bar{\zeta}^{(t)}\big), \frac{3\eta}{|\mathcal{S}|}\sum_{s\in\mathcal{S}}\left\|Q^{(t)}(s) - Q_\tau^\star(s)\right\|_\infty\right\}$$
$$\leq \frac{3000}{(1-\gamma)^2\tau}\left(1 - \frac{(1-\gamma)\eta\tau}{4}\right)^t; \quad (12a)$$

*and*

$$\max_{s\in\mathcal{S},\mu,\nu}\left(V_\tau^{\mu,\bar{\nu}^{(t)}}(s) - V_\tau^{\bar{\mu}^{(t)},\nu}(s)\right) \leq \frac{6000|\mathcal{S}|}{(1-\gamma)^3\tau}\max\left\{\frac{8}{(1-\gamma)^2\tau}, \frac{1}{\eta}\right\}\left(1 - \frac{(1-\gamma)\eta\tau}{4}\right)^t. \quad (12b)$$

Theorem 1 demonstrates that as long as the learning rate $\eta$ is small enough, the last iterate of Algorithm 1 converges at a linear rate for the entropy-regularized Markov game. Compared with prior literatures investigating on policy optimization, our analysis focuses on the last-iterate convergence of non-Euclidean updates in the presence of entropy regularization, which appears to be the first of its kind. Several remarks are in order, with detailed comparisons in Table 1.

- **Linear convergence to the QRE.** Theorem 1 demonstrates that the last iterate of Algorithm 1 takes at most $\widetilde{\mathcal{O}}\left(\frac{1}{(1-\gamma)\eta\tau}\log\frac{1}{\epsilon}\right)$ iterations to yield an $\epsilon$-optimal policy in terms of the KL divergence to the QRE $\max_{s\in\mathcal{S}}\mathsf{KL}_s\big(\zeta_\tau^\star \| \bar{\zeta}^{(t)}\big) \leq \epsilon$, the entrywise error of the regularized Q-function $\left\|Q^{(t)} - Q_\tau^\star\right\|_\infty \leq \epsilon$, as well as the duality gap $\max_{s\in\mathcal{S},\mu,\nu}\left(V_\tau^{\mu,\bar{\nu}^{(t)}}(s) - V_\tau^{\bar{\mu}^{(t)},\nu}(s)\right) \leq \epsilon$ at once. Minimizing the bound over the learning rate $\eta$, the proposed method is guaranteed to find an $\epsilon$-QRE within $\widetilde{\mathcal{O}}\left(\frac{|\mathcal{S}|}{(1-\gamma)^4\tau}\log\frac{1}{\epsilon}\right)$ iterations, which significantly improves upon the one-side convergence rate of Zeng et al. (2022).

- **Last-iterate convergence to $\epsilon$-optimal NE.** By setting $\tau = \frac{(1-\gamma)\epsilon}{2(\log|\mathcal{A}|+\log|\mathcal{B}|)}$, this immediately leads to provable last-iterate convergence to an $\epsilon$-NE, with an iteration complexity of $\widetilde{\mathcal{O}}\left(\frac{|\mathcal{S}|}{(1-\gamma)^5\epsilon}\right)$, which again outperforms the convergence rate in Wei et al. (2021b).

**Remark 1.** *The learning rate $\eta$ is constrained to be inverse proportional to $|\mathcal{S}|$, which is for the worst case and can be potentially loosened for problems with a small concentration coefficient. We refer interested readers to Appendix A for details.*

## 3 ALGORITHM AND THEORY: THE EPISODIC SETTING

**Episodic two-player zero-sum Markov game.** An episodic two-player zero-sum Markov game is defined by a tuple $\{\mathcal{S}, \mathcal{A}, \mathcal{B}, H, \{P_h\}_{h=1}^H, \{r_h\}_{h=1}^H\}$, with $\mathcal{S}$ being a finite state space, $\mathcal{A}$ and $\mathcal{B}$ denoting finite action spaces of the two players, and $H > 0$ the horizon length. Every step $h \in [H]$ admits a transition probability kernel $P_h : \mathcal{S} \times \mathcal{A} \rightarrow \Delta(\mathcal{S})$ and reward function $r_h : \mathcal{S} \times \mathcal{A} \times \mathcal{B} \rightarrow [0,1]$. Furthermore, $\mu = \{\mu_h\}_{h=1}^H$ and $\{\nu_h\}_{h=1}^H$ denote the policies of the two players, where the probability of the max player choosing $a \in \mathcal{A}$ (resp. the min player choosing $b \in \mathcal{B}$) at time $h$ is specified by $\mu_h(a|s)$ (resp. $\nu_h(a|s)$).

**Entropy regularized value functions.** The value function and Q-function characterize the expected cumulative reward starting from step $h$ by following the policy pair $\mu, \nu$. For conciseness, we only present the definition of entropy-regularized value functions below and remark that the their un-regularized counterparts $V_h^{\mu,\nu}$ and $Q_h^{\mu,\nu}$ can be obtained by setting $\tau = 0$. We have

$$V_{h,\tau}^{\mu,\nu}(s) = \mathbb{E}\left[\sum_{h'=h}^H \left[r_{h'}(s_{h'}, a_{h'}, b_{h'}) - \tau \log \mu_{h'}(a_{h'}|s_{h'}) + \tau \log \nu_{h'}(b_{h'}|s_{h'})\right] \ \bigg| \ s_h = s\right];$$

$$Q_{h,\tau}^{\mu,\nu}(s,a,b) = r_h(s,a,b) + \mathbb{E}_{s' \sim P_h(\cdot|s,a,b)}\left[V_{h+1,\tau}^{\mu,\nu}(s')\right].$$

The solution concept of NE and QRE are defined in a similar manner by focusing on the episodic versions of value functions. We again denote the unique QRE by $\zeta_\tau^\star = (\mu_\tau^\star, \nu_\tau^\star)$.

**Proposed method and convergence guarantee** It is straightforward to adapt Algorithm 1 to the episodic setting with minimal modifications, with detailed procedure showcased in Algorithm 2 (cf. Appendix B). The analysis, which substantially deviates from the discounted setting, exploits the structure of finite-horizon MDP and time-inhomogeneous policies, enabling a much larger range of learning rates as showed in the following theorem.

**Theorem 2.** *Setting $0 < \eta \leq \frac{1}{8H}$ and $\alpha_t = \eta\tau$, it holds for all $h \in [H]$ and $t \geq T_h := (H-h)T_{\text{start}}$ with $T_{\text{start}} = \lceil \frac{1}{\eta\tau} \log H \rceil$ that*

$$\left\|Q_{h,\tau}^\star - Q_h^{(t)}\right\|_\infty \leq (1-\eta\tau)^{t-T_h} t^{H-h}; \tag{13a}$$

$$\max_{s \in \mathcal{S}, \mu, \nu} \left(V_{h,\tau}^{\mu, \bar{\nu}^{(t)}}(s) - V_{h,\tau}^{\bar{\mu}^{(t)}, \nu}(s)\right) \leq 4(1-\eta\tau)^{t-T_h} \max\left\{\frac{8H^2}{\tau}, \frac{1}{\eta}\right\}\left(\frac{8H}{\tau} + 6\eta t^{H-h+1}\right). \tag{13b}$$

Theorem 2 implies that the last iterate of Algorithm 2 takes no more than $\widetilde{\mathcal{O}}\left(HT_{\text{start}} + \frac{H}{\eta\tau}\log\frac{1}{\epsilon}\right) = \widetilde{\mathcal{O}}\left(\frac{H}{\eta\tau}\log\frac{1}{\epsilon}\right)$ iterations for finding an $\epsilon$-QRE. Minimizing the bound over the learning rate $\eta$, Algorithm 2 is guaranteed to find an $\epsilon$-QRE in $\widetilde{\mathcal{O}}\left(\frac{H^2}{\tau}\log\frac{1}{\epsilon}\right)$ iterations, which translates into an iteration complexity of $\widetilde{\mathcal{O}}\left(\frac{H^3}{\epsilon}\right)$ for finding an $\epsilon$-NE in terms of the duality gap, i.e., $\max_{s \in \mathcal{S}, h \in [H], \mu, \nu}\left(V_h^{\mu, \bar{\nu}^{(t)}}(s) - V_h^{\bar{\mu}^{(t)}, \nu}(s)\right) \leq \epsilon$, by setting $\tau = \mathcal{O}\left(\frac{\epsilon}{H(\log|\mathcal{A}| + \log|\mathcal{B}|)}\right)$.

## 4 DISCUSSION

This work develops policy optimization methods for zero-sum Markov games that feature single-loop and symmetric updates with provable last-iterate convergence guarantees. Our approach yields better iteration complexities in both infinite-horizon and finite-horizon settings, by adopting entropy regularization and non-Euclidean policy update. Important future directions include investigating whether larger learning rates are possible without knowing problem-dependent information a priori, extending the framework to allow function approximation, and designing sample-efficient implementations of the proposed method.

## ACKNOWLEDGMENTS

The authors would like to thank Gen Li and Zeyuan Allen-Zhu for valuable discussions. Part of this work was completed while S. Cen was an intern at Meta AI Research. S. Cen and Y. Chi are supported in part by the grants ONR N00014-18-1-2142 and N00014-19-1-2404, ARO W911NF-18-1-0303, NSF CCF-1901199, CCF-2007911, CCF-2106778 and CNS-2148212. S. Cen is also gratefully supported by Wei Shen and Xuehong Zhang Presidential Fellowship, and Nicholas Minnici Dean's Graduate Fellowship in Electrical and Computer Engineering at Carnegie Mellon University.

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

# Appendix

## Table of Contents

## A  ANALYSIS FOR THE INFINITE-HORIZON SETTING

**Definition 1.** *Given $\rho \in \Delta(\mathcal{S})$ with $\rho(s) > 0, \forall s \in S$, concentrability coefficient $c_\rho(t)$ is defined as*

$$c_\rho(t) = \sup_{\substack{x^{(l)} \in \mathcal{A}^{\mathcal{S}}, 1 \leq l \leq t, \\ y^{(l)} \in \mathcal{B}^{\mathcal{S}}, 1 \leq l \leq t}} \left\| \frac{\rho P_{x^{(1)},y^{(1)}} \cdots P_{x^{(t)},y^{(t)}}}{\rho} \right\|_\infty,$$

*where $P_{x^{(l)},y^{(l)}} \in \mathbb{R}^{|\mathcal{S}| \times |\mathcal{S}|}$ is the state transition matrix induced by a pair of deterministic policy $x^{(l)}, y^{(l)}$:*

$$[P_{x^{(l)},y^{(l)}}]_{s,s'} = P(s'|s, x^{(l)}(s), y^{(l)}(s)).$$

*Let $\mathcal{C}_\rho$ be the maximum value of $c_\rho(t)$ over $t \geq 0$:*

$$\mathcal{C}_\rho = \sup_{t \geq 0} c_\rho(t).$$

*In addition, let $\Gamma(\rho)$ be the set of all possible distribution over $\mathcal{S}$ induced by initial distribution $\rho$ and deterministic policy sequences, i.e.,*

$$\Gamma(\rho) = \bigcup_{t=0}^{\infty} \left\{ \rho P_{x^{(1)},y^{(1)}} \cdots P_{x^{(t)},y^{(t)}} : x^{(l)} \in \mathcal{A}^{\mathcal{S}}, y^{(l)} \in \mathcal{B}^{\mathcal{S}}, \forall l \in [t] \right\}$$

We make note that Theorem 1 is the direct corollary of following theorems, by setting $\rho$ to the uniform distribution over $\mathcal{S}$, where $\mathcal{C}_\rho$ admits a trivial upper bounded $|\mathcal{S}|$.

**Theorem 3.** *With $0 < \eta \leq \frac{(1-\gamma)^3}{32000\mathcal{C}_\rho}$, and $\alpha_i = \eta\tau$, we have*

$$\max\left\{\mathsf{KL}_\rho\big(\zeta_\tau^\star \,\|\, \zeta^{(t)}\big), \frac{1}{2}\mathsf{KL}_\rho\big(\zeta_\tau^\star \,\|\, \bar\zeta^{(t)}\big), 3\eta \mathop{\mathbb{E}}_{s\sim\rho}\left[\big\|Q^{(t)}(s) - Q_\tau^\star(s)\big\|_\infty\right]\right\} \leq \frac{3000}{(1-\gamma)^2\tau}\left(1 - \frac{(1-\gamma)\eta\tau}{4}\right)^t.$$

**Definition 2.** *We define regularized minimax mismatch coefficient by*

$$\mathcal{C}_{\rho,\tau}^\dagger = \max\left\{\max_\mu \left\|\frac{d_\rho^{\mu,\nu_\tau^\dagger(\mu)}}{\rho}\right\|_\infty, \max_\nu \left\|\frac{d_\rho^{\mu_\tau^\dagger(\nu),\nu}}{\rho}\right\|_\infty\right\}.$$

*Here, $\nu_\tau^\dagger(\mu)$ denotes the optimal policy of the min player when the max player adopts policy $\mu$:*

$$\nu_\tau^\dagger(\mu) = \arg\min_\nu V_\tau^{\mu,\nu}(s),$$

*and $\mu_\tau^\dagger(\nu)$ is defined in a symmetric way. The discounted state visitation distribution $d_\rho^{\mu,\nu}$ is defined as*

$$d_\rho^{\mu,\nu}(s) = (1-\gamma) \mathop{\mathbb{E}}_{s_0\sim\rho}\left[\sum_{t=0}^\infty \gamma^t P(s_t = s | s_0)\right].$$

Note that this definition parallels that of the (unregularized) minimax mismatch coefficient in (Daskalakis et al., 2020).

**Theorem 4.** *With $0 < \eta \leq \frac{(1-\gamma)^3}{32000\mathcal{C}_\rho}$, and $\alpha_i = \eta\tau$, we have*

$$\max_{s\in\mathcal{S},\mu,\nu}\left(V_\tau^{\mu,\bar\nu^{(t)}}(s) - V_\tau^{\bar\mu^{(t)},\nu}(s)\right) \leq \frac{6000\|1/\rho\|_\infty}{(1-\gamma)^3\tau}\max\left\{\frac{8}{(1-\gamma)^2\tau}, \frac{1}{\eta}\right\}\left(1 - \frac{(1-\gamma)\eta\tau}{4}\right)^t,$$

*and*

$$\max_{\mu,\nu}\left(V_\tau^{\mu,\bar\nu^{(t)}}(\rho) - V_\tau^{\bar\mu^{(t)},\nu}(\rho)\right) \leq \frac{6000\mathcal{C}_{\rho,\tau}^\dagger}{(1-\gamma)^3\tau}\max\left\{\frac{8}{(1-\gamma)^2\tau}, \frac{1}{\eta}\right\}\left(1 - \frac{(1-\gamma)\eta\tau}{4}\right)^t.$$

We start with the following lemma. The proof can be found in Appendix C.1. For notational simplicity, we set $Q^{(-1)} = 0$, $\bar\zeta^{(-1)} = \bar\zeta^{(0)}$ and $\alpha_0 = 1$. It follows from the update rule (9a) that $\bar\zeta^{(1)} = \zeta^{(0)} = \bar\zeta^{(0)}$.

**Lemma 1.** *It holds for any step size $0 < \eta \leq 1/\tau$ and $t \geq 0$ that*

$$\mathsf{KL}_\rho\big(\zeta_\tau^\star \,\|\, \zeta^{(t+1)}\big) - (1-\eta\tau)\mathsf{KL}_\rho\big(\zeta_\tau^\star \,\|\, \zeta^{(t)}\big) + \left(1 - \eta\tau - \frac{4\eta}{1-\gamma}\right)\mathsf{KL}_\rho\big(\bar\zeta^{(t+1)} \,\|\, \bar\zeta^{(t)}\big)$$

$$+ \eta\tau\mathsf{KL}_\rho\big(\bar\zeta^{(t+1)} \,\|\, \zeta_\tau^\star\big) + \left(1 - \frac{2\eta}{1-\gamma}\right)\mathsf{KL}_\rho\big(\zeta^{(t+1)} \,\|\, \bar\zeta^{(t+1)}\big) + (1-\eta\tau)\mathsf{KL}_\rho\big(\bar\zeta^{(t)} \,\|\, \zeta^{(t)}\big)$$

$$- \frac{2\eta}{1-\gamma}\mathsf{KL}_\rho\big(\bar\zeta^{(t)} \,\|\, \bar\zeta^{(t-1)}\big)$$

$$\leq \mathop{\mathbb{E}}_{s\sim\rho}\left[2\eta\big\|Q^{(t+1)}(s) - Q_\tau^\star(s)\big\|_\infty + \frac{4\eta^2}{1-\gamma}\big\|Q^{(t)}(s) - Q^{(t+1)}(s)\big\|_\infty + \frac{12\eta^2}{1-\gamma}\big\|Q^{(t-1)}(s) - Q^{(t)}(s)\big\|_\infty\right].$$
$$(14)$$

It remains to bound the terms on the right hand side of (14). By a slight abuse of notation, we denote

$$\big\|Q^{(t+1)} - Q_\tau^\star\big\|_{\Gamma(\rho)} = \sup_{\chi\in\Gamma(\rho)} \mathop{\mathbb{E}}_{s\sim\chi}\left[\big\|Q^{(t+1)}(s) - Q_\tau^\star(s)\big\|_\infty\right],$$

and

$$\big\|Q^{(t+1)} - Q^{(t)}\big\|_{\Gamma(\rho)} = \sup_{\chi\in\Gamma(\rho)} \mathop{\mathbb{E}}_{s\sim\chi}\left[\big\|Q^{(t+1)}(s) - Q^{(t)}(s)\big\|_\infty\right].$$

The following two lemmas establish a set of recursive bounds that relate $\big\{\big\|Q^{(l+1)}(s) - Q_\tau^\star(s)\big\|_{\Gamma(\rho)}\big\}_{l=0,\cdots,t}$ and $\big\{\big\|Q^{(l+1)}(s) - Q^{(l)}(s)\big\|_{\Gamma(\rho)}\big\}_{l=0,\cdots,t}$ with $\big\{\mathsf{KL}_\rho\big(\bar\zeta^{(l+1)} \,\|\, \bar\zeta^{(l)}\big)\big\}_{l=0,\cdots,t-1}$:

**Lemma 2.** *With $0 < \eta \leq \min\{(1-\gamma)/180, (1-\gamma)^2/48\}$, it holds for all $t \geq 1$ that*

$$\left\|Q^{(t+1)} - Q^{(t)}\right\|_{\Gamma(\rho)} \leq \frac{1+\gamma}{2} \sum_{l=1}^{t} \alpha_{l,t} \left\|Q^{(l)} - Q^{(l-1)}\right\|_{\Gamma(\rho)} + \frac{4\mathcal{C}_\rho}{\eta} \cdot \sum_{l=1}^{t} \alpha_{l,t} \mathsf{KL}_\rho\big(\bar{\zeta}^{(l)} \,\|\, \bar{\zeta}^{(l-1)}\big),$$

(15)

*Here, $\alpha_{l,t}$ is defined as*

$$\alpha_{l,t} = \alpha_l \prod_{i=l+1}^{t} (1 - \alpha_i).$$

*When $t = 0$, we have $\left\|Q^{(1)}(s) - Q^{(0)}(s)\right\|_{\Gamma(\rho)} \leq 2$.*

*Proof.* The proof can be found in Appendix C.2. □

**Lemma 3.** *With $0 < \eta \leq (1-\gamma)^2/16$, it holds for all $t \geq 1$ that*

$$\left\|Q^{(t+1)} - Q_\tau^\star\right\|_{\Gamma(\rho)}$$
$$\leq \frac{1+\gamma}{2} \cdot \sum_{l=0}^{t} \alpha_{l,t} \Big(\left\|Q^{(l)} - Q_\tau^\star\right\|_{\Gamma(\rho)} + \frac{2\eta}{1-\gamma}\left\|Q^{(l)} - Q^{(l-1)}\right\|_{\Gamma(\rho)}\Big) + 2\alpha_{0,t}.$$

(16)

*When $t = 0$, we have $\left\|Q^{(t+1)} - Q_\tau^\star\right\|_{\Gamma(\rho)} \leq \frac{2\gamma}{1-\gamma}$.*

*Proof.* The proof can be found in Appendix C.3. □

The following lemma further demystify the complicated recursive bounds showed in Lemma 2 and 3.

**Lemma 4.** *Let $\lambda_{l,t}$ be defined as*

$$\lambda_{l,t} = \alpha_l \prod_{i=l+1}^{t} \Big(1 - \frac{1-\gamma}{4} \cdot \alpha_i\Big).$$

*Under the assumption of Lemma 2 and 3, it holds for all $t \geq 0$ that*

$$\sum_{l=0}^{t} \lambda_{l+1,t+1}\Big[\eta\left\|Q_\tau^\star - Q^{(l+1)}\right\|_{\Gamma(\rho)} + \frac{12\eta^2}{(1-\gamma)^2}\left\|Q^{(l+1)} - Q^{(l)}\right\|_{\Gamma(\rho)}\Big]$$
$$\leq \frac{6250\eta\mathcal{C}_\rho}{(1-\gamma)^3} \sum_{l=0}^{t-1} \lambda_{l+1,t+1} \mathsf{KL}\big(\bar{\zeta}^{(l+1)} \,\|\, \bar{\zeta}^{(l)}\big) + \frac{550\eta}{(1-\gamma)^2}\lambda_{0,t+1}$$

*Proof.* The proof can be found in Appendix C.4. □

We are now ready to prove our main results. Averaging (14) with weight $\lambda$ gives

$$\sum_{l=0}^{t} \lambda_{l+1,t+1}\bigg[\mathsf{KL}_\rho\big(\zeta_\tau^\star \,\|\, \zeta^{(l+1)}\big) - (1 - \eta\tau)\mathsf{KL}_\rho\big(\zeta_\tau^\star \,\|\, \zeta^{(l)}\big)$$
$$+ \Big(1 - \frac{2\eta}{1-\gamma}\Big)\mathsf{KL}_\rho\big(\zeta^{(l+1)} \,\|\, \bar{\zeta}^{(l+1)}\big) + 3\eta \mathop{\mathbb{E}}_{s\sim\rho}\Big[\left\|Q^{(l+1)}(s) - Q_\tau^\star(s)\right\|_\infty\Big]$$
$$+ \Big(1 - \eta\tau - \frac{4\eta}{1-\gamma}\Big)\mathsf{KL}_\rho\big(\bar{\zeta}^{(l+1)} \,\|\, \bar{\zeta}^{(l)}\big) - \frac{2\eta}{1-\gamma}\mathsf{KL}_\rho\big(\bar{\zeta}^{(l)} \,\|\, \bar{\zeta}^{(l-1)}\big)\bigg]$$
$$\leq \sum_{l=0}^{t} \lambda_{l+1,t+1} \mathop{\mathbb{E}}_{s\sim\rho}\Big[5\eta\left\|Q^{(l+1)}(s) - Q_\tau^\star(s)\right\|_\infty + \frac{4\eta^2}{1-\gamma}\left\|Q^{(l+1)}(s) - Q^{(l)}(s)\right\|_\infty + \frac{13\eta^2}{1-\gamma}\left\|Q^{(l-1)}(s) - Q^{(l)}(s)\right\|_\infty\Big]$$
$$\leq 5\sum_{l=0}^{t} \lambda_{l+1,t+1} \mathop{\mathbb{E}}_{s\sim\rho}\Big[\eta\left\|Q_\tau^\star(s) - Q^{(l+1)}(s)\right\|_{\Gamma(\rho)} + \frac{12\eta^2}{(1-\gamma)^2}\left\|Q^{(l+1)}(s) - Q^{(l)}(s)\right\|_{\Gamma(\rho)}\Big]$$

$$\leq \frac{31250\eta\mathcal{C}_\rho}{(1-\gamma)^3} \sum_{l=0}^{t-1} \lambda_{l+1,t+1} \mathsf{KL}_\rho\big(\bar{\zeta}^{(l+1)} \,\|\, \bar{\zeta}^{(l)}\big) + \frac{2750\eta}{(1-\gamma)^2} \lambda_{0,t+1}$$

for all $t \geq 0$. Rearranging terms, we have

$$\alpha_{t+1} \Big[ \mathsf{KL}_\rho\big(\zeta_\tau^\star \,\|\, \zeta^{(t+1)}\big) + \Big(1 - \frac{2\eta}{1-\gamma}\Big) \mathsf{KL}_\rho\big(\zeta^{(t+1)} \,\|\, \bar{\zeta}^{(t+1)}\big) + 3\eta \underset{s\sim\rho}{\mathbb{E}} \Big[ \big\|Q^{(t+1)}(s) - Q_\tau^\star(s)\big\|_\infty \Big] \Big]$$

$$+ \sum_{l=1}^{t} (\lambda_{l,t+1} - (1-\eta\tau)\lambda_{l+1,t+1}) \mathsf{KL}_\rho\big(\zeta_\tau^\star \,\|\, \zeta^{(l)}\big)$$

$$+ \sum_{l=0}^{t-1} \Big[ \lambda_{l+1,t+1}\Big(1 - \eta\tau - \frac{4\eta}{1-\gamma} - \frac{31250\eta\mathcal{C}_\rho}{(1-\gamma)^3}\Big) - \lambda_{l+2,t+1}\frac{2\eta}{1-\gamma} \Big] \mathsf{KL}\big(\bar{\zeta}^{(l+1)} \,\|\, \bar{\zeta}^{(l)}\big)$$

$$\leq \frac{2750\eta}{(1-\gamma)^2} \lambda_{0,t+1} + (1-\eta\tau)\lambda_{1,t+1} \mathsf{KL}_\rho\big(\zeta_\tau^\star \,\|\, \zeta^{(0)}\big) \leq \Big( \frac{2750\eta}{(1-\gamma)^2} + \eta \Big)\lambda_{0,t+1}.$$

With $0 < \eta \leq \frac{(1-\gamma)^3}{32000\mathcal{C}_\rho}$, and $\alpha_i = \eta\tau$, we have $\lambda_{l,t+1} - (1-\eta\tau)\lambda_{l+1,t+1} \geq 0$ (c.f. (40)), and

$$\lambda_{l+1,t+1}\Big(1 - \eta\tau - \frac{4\eta}{1-\gamma} - \frac{31250\eta\mathcal{C}_\rho}{(1-\gamma)^3}\Big) - \lambda_{l+2,t+1}\frac{2\eta}{1-\gamma}$$

$$= \eta\tau \prod_{j=l+3}^{t+1} \Big(1 - \frac{1-\gamma}{4}\alpha_j\Big) \Big[\big(1 - \frac{1-\gamma}{4}\eta\tau\big)\Big(1 - \eta\tau - \frac{4\eta}{1-\gamma} - \frac{31250\eta\mathcal{C}_\rho}{(1-\gamma)^3}\Big) - \frac{2\eta}{1-\gamma}\Big] \geq 0.$$

It follows that

$$\mathsf{KL}_\rho\big(\zeta_\tau^\star \,\|\, \zeta^{(t+1)}\big) + \Big(1 - \frac{2\eta}{1-\gamma}\Big)\mathsf{KL}_\rho\big(\zeta^{(t+1)} \,\|\, \bar{\zeta}^{(t+1)}\big) + 3\eta \underset{s\sim\rho}{\mathbb{E}} \Big[ \big\|Q^{(t+1)}(s) - Q_\tau^\star(s)\big\|_\infty \Big]$$

$$\leq \Big( \frac{2750}{(1-\gamma)^2\tau} + \frac{1}{\tau} \Big)\Big(1 - \frac{(1-\gamma)\eta\tau}{4}\Big)^{t+1} < \frac{3000}{(1-\gamma)^2\tau}\Big(1 - \frac{(1-\gamma)\eta\tau}{4}\Big)^{t+1}. \tag{17}$$

This proves the bound of $\mathsf{KL}_\rho\big(\zeta_\tau^\star \,\|\, \zeta^{(t+1)}\big)$ and $3\eta \underset{s\sim\rho}{\mathbb{E}} \big[\big\|Q^{(t+1)}(s) - Q_\tau^\star(s)\big\|_\infty\big]$ in Theorem 3. Note that the bound holds trivially for $\mathsf{KL}_\rho\big(\zeta_\tau^\star \,\|\, \zeta^{(0)}\big)$ and $3\eta \underset{s\sim\rho}{\mathbb{E}} \big[\big\|Q^{(0)}(s) - Q_\tau^\star(s)\big\|_\infty\big]$. It remains to bound $\mathsf{KL}\big(\zeta_\tau^\star \,\|\, \bar{\zeta}^{(t+1)}\big)$.

**Lemma 5.** *With* $0 < \eta \leq (1-\gamma)/8$, *we have*

$$\frac{1}{2}\mathsf{KL}_s\big(\zeta_\tau^\star \,\|\, \bar{\zeta}^{(t+1)}\big) + \eta\tau\mathsf{KL}_s\big(\bar{\zeta}^{(t+1)} \,\|\, \zeta_\tau^\star\big)$$

$$\leq (1-\eta\tau)\mathsf{KL}_s\big(\zeta_\tau^\star \,\|\, \zeta^{(t)}\big) + \frac{2\eta}{1-\gamma}\mathsf{KL}_s\big(\zeta^{(t)} \,\|\, \bar{\zeta}^{(t)}\big) + 2\eta\big\|Q^{(t)}(s) - Q_\tau^\star(s)\big\|_\infty.$$

*Proof.* See Appendix C.5. □

Combining the above Lemma with (17) gives

$$\frac{1}{2}\mathsf{KL}_\rho\big(\zeta_\tau^\star \,\|\, \bar{\zeta}^{(t+1)}\big) + \eta\tau\mathsf{KL}_\rho\big(\bar{\zeta}^{(t+1)} \,\|\, \zeta_\tau^\star\big)$$

$$\leq (1-\eta\tau)\Big(\mathsf{KL}_\rho\big(\zeta_\tau^\star \,\|\, \zeta^{(t)}\big) + \Big(1 - \frac{2\eta}{1-\gamma}\Big)\mathsf{KL}_\rho\big(\zeta^{(t)} \,\|\, \bar{\zeta}^{(t)}\big) + 3\eta \underset{s\sim\rho}{\mathbb{E}} \Big[ \big\|Q^{(t)}(s) - Q_\tau^\star(s)\big\|_\infty \Big]\Big)$$

$$\leq \frac{3000}{(1-\gamma)^2\tau}\Big(1 - \frac{(1-\gamma)\eta\tau}{4}\Big)^{t+1}.$$

We are now ready to bound the duality gap. Before proceeding, we introduce the following two lemmas:

**Lemma 6.** *It holds for any policy pair $\mu, \nu$ that*

$$\max_{\mu',\nu'} \left( V_\tau^{\mu',\nu}(\rho) - V_\tau^{\mu,\nu'}(\rho) \right) \leq \frac{2\mathcal{C}_{\rho,\tau}^\dagger}{1-\gamma} \mathop{\mathbb{E}}_{s\sim\rho} \left[ \max_{\mu',\nu'} \left( f_s(Q_\tau^\star, \mu', \nu) - f_s(Q_\tau^\star, \mu, \nu') \right) \right] \tag{18}$$

*and*

$$\max_{s\in\mathcal{S},\mu',\nu'} \left( V_\tau^{\mu',\nu}(s) - V_\tau^{\mu,\nu'}(s) \right) \leq \frac{2\|1/\rho\|_\infty}{1-\gamma} \mathop{\mathbb{E}}_{s\sim\rho} \left[ \max_{\mu',\nu'} \left( f_s(Q_\tau^\star, \mu', \nu) - f_s(Q_\tau^\star, \mu, \nu') \right) \right]. \tag{19}$$

*Proof.* Note that (19) is a slight generalization of (Wei et al., 2021b, Lemma 32) . The proof can be found in Appendix C.6. □

**Lemma 7** ((Cen et al., 2021b, Lemma 4))**.** *It holds for all $s \in \mathcal{S}$ and policy pair $\mu, \nu$ that*

$$\max_{\mu',\nu'} \left( f_s(Q_\tau^\star, \mu', \nu) - f_s(Q_\tau^\star, \mu, \nu') \right) \leq \frac{4}{(1-\gamma)^2\tau} \mathsf{KL}_s\left( \zeta_\tau^\star \,\|\, \zeta \right) + \tau \mathsf{KL}_s\left( \zeta \,\|\, \zeta_\tau^\star \right).$$

Putting all pieces together, we arrive at

$$\max_{\mu,\nu} \left( V_\tau^{\mu,\bar\nu^{(t)}}(\rho) - V_\tau^{\bar\mu^{(t)},\nu}(\rho) \right) \leq \frac{2\mathcal{C}_{\rho,\tau}^\dagger}{1-\gamma} \left( \frac{4}{(1-\gamma)^2\tau} \mathsf{KL}_\rho\left( \zeta_\tau^\star \,\|\, \bar\zeta^{(t+1)} \right) + \tau \mathsf{KL}_\rho\left( \bar\zeta^{(t+1)} \,\|\, \zeta_\tau^\star \right) \right)$$

$$\leq \frac{2\mathcal{C}_{\rho,\tau}^\dagger}{1-\gamma} \max\left\{ \frac{8}{(1-\gamma)^2\tau}, \frac{1}{\eta} \right\} \left( \frac{1}{2}\mathsf{KL}_\rho\left( \zeta_\tau^\star \,\|\, \bar\zeta^{(t+1)} \right) + \eta\tau \mathsf{KL}_\rho\left( \bar\zeta^{(t+1)} \,\|\, \zeta_\tau^\star \right) \right)$$

$$\leq \frac{6000\mathcal{C}_{\rho,\tau}^\dagger}{(1-\gamma)^3\tau} \max\left\{ \frac{8}{(1-\gamma)^2\tau}, \frac{1}{\eta} \right\} \left( 1 - \frac{(1-\gamma)\eta\tau}{4} \right)^t.$$

We omit the proof for $\max_{s\in\mathcal{S},\mu,\nu} \left( V_\tau^{\mu,\bar\nu^{(t)}}(s) - V_\tau^{\bar\mu^{(t)},\nu}(s) \right)$ as it follows virtually the same argument.

# B  ANALYSIS FOR THE EPISODIC SETTING

Throughout the analysis, we restrict our choice of value update step size to $\alpha_t = \eta\tau$. We start with the following lemma which parallels Lemma 11 in the episodic Markov game setting:

**Lemma 8.** *With $0 < \eta \leq 1/\tau$, it holds for all $s \in \mathcal{S}$, $h \in [H]$ and $t \geq 0$ that*

$$\max\left\{ \left\|\bar\mu_h^{(t+1)}(s) - \mu_h^{(t+1)}(s)\right\|_1, \left\|\bar\nu_h^{(t+1)}(s) - \nu_h^{(t+1)}(s)\right\|_1 \right\} \leq 2\eta H. \tag{22}$$

*In addition, we have*

$$\max\{ \|\log \zeta_h^{(t)}(s)\|_\infty, \|\log \bar\zeta_h^{(t)}(s)\|_\infty \| \log \zeta_{h,\tau}^\star(s)\|_\infty \} \leq \frac{2H}{\tau} \tag{23}$$

**Lemma 9.** *With $0 < \eta \leq \frac{1}{8H}$, it holds for all $0 \leq t_1 \leq t_2$, $h \in [H]$ and $s \in \mathcal{S}$ that*

$$\mathsf{KL}_s\left( \zeta_{h,\tau}^\star \,\|\, \zeta_h^{(t_2)} \right) + (1-4\eta H)\mathsf{KL}_s\left( \zeta_h^{(t_2)} \,\|\, \bar\zeta_h^{(t_2)} \right)$$

$$\leq (1-\eta\tau)^{t_2-t_1} \left( \mathsf{KL}_s\left( \zeta_{h,\tau}^\star \,\|\, \zeta_h^{(t_1)} \right) + (1-4\eta H)\mathsf{KL}_s\left( \zeta_h^{(t_1)} \,\|\, \bar\zeta_h^{(t_1)} \right) \right)$$

$$+ 4\eta \sum_{l=t_1}^{t_2} (1-\eta\tau)^{t_2-l} \left\| Q_h^{(l)}(s) - Q_\tau^\star(s) \right\|_\infty.$$

*Proof.* See Appendix D.1. □

**Lemma 10.** *With $0 < \eta \leq \frac{1}{8H}$, it holds for all $0 < t_1 \leq t_2$, $2 \leq h \leq H$ and $s \in \mathcal{S}$ that*

$$\left| Q_{h-1}^{(t_2)}(s,a,b) - Q_{h-1,\tau}^\star(s,a,b) \right|$$

$$\leq 2(1-\eta\tau)^{t_2-t_1}H + 10\eta\tau \mathop{\mathbb{E}}_{s'\sim P_{h-1}(\cdot|s,a,b)} \left[ \sum_{l=t_1-1}^{t_2-1} (1-\eta\tau)^{t_2-1-l} \left\| Q_h^{(l)}(s) - Q_{h,\tau}^\star(s) \right\|_\infty \right]$$

$$+ \tau(1-\eta\tau)^{t_2-t_1} \mathop{\mathbb{E}}_{s'\sim P_{h-1}(\cdot|s,a,b)} \left[ \mathsf{KL}_s\left( \zeta_{h,\tau}^\star \,\|\, \zeta_h^{(t_1-1)} \right) + (1-4\eta H)\mathsf{KL}_s\left( \zeta_h^{(t_1-1)} \,\|\, \bar\zeta_h^{(t_1-1)} \right) \right].$$

---

**Algorithm 2:** Entropy-regularized OMWU for Episodic Two-player Zero-sum Markov Game

---

1 **Input:** Regularization parameter $\tau > 0$, learning rate for policy update $\eta > 0$, learning rate for value update $\{\alpha_t\}_{t=1}^{\infty}$.

2 **Initialization:** Set $\mu^{(0)}, \bar{\mu}^{(0)}, \nu^{(0)}$ and $\bar{\nu}^{(0)}$ as uniform policies; set

$$Q^{(0)} = 0, \quad V^{(0)} = \tau(\log|\mathcal{A}| - \log|\mathcal{B}|).$$

 **for** $t = 0, 1, \cdots$ **do**

3   **for** *all* $h \in [H]$, $s \in \mathcal{S}$ **do in parallel**

4    When $t \geq 1$, update policy pair $\zeta_h^{(t)}(s)$ as:

$$\begin{cases} \mu_h^{(t)}(a|s) \propto \mu_h^{(t-1)}(a|s)^{1-\eta\tau} \exp(\eta[Q_h^{(t)}(s)\bar{\nu}_h^{(t)}(s)]_a) \\ \nu_h^{(t)}(b|s) \propto \nu_h^{(t-1)}(b|s)^{1-\eta\tau} \exp(-\eta[Q_h^{(t)}(s)^\top \bar{\mu}_h^{(t)}(s)]_b) \end{cases} \quad . \quad (20a)$$

5    Update policy pair $\bar{\zeta}_h^{(t+1)}(s)$ as:

$$\begin{cases} \bar{\mu}_h^{(t+1)}(a|s) \propto \mu_h^{(t)}(a|s)^{1-\eta\tau} \exp(\eta[Q_h^{(t)}(s)\bar{\nu}_h^{(t)}(s)]_a) \\ \bar{\nu}_h^{(t+1)}(b|s) \propto \nu_h^{(t)}(b|s)^{1-\eta\tau} \exp(-\eta[Q_h^{(t)}(s)^\top \bar{\mu}_h^{(t)}(s)]_b) \end{cases} \quad . \quad (20b)$$

6    Update $Q_h^{(t+1)}(s)$ and $V_h^{(t+1)}(s)$ as

$$\begin{cases} Q_h^{(t+1)}(s,a,b) &= r_h(s,a,b) + \mathbb{E}_{s'\sim P_h(\cdot|s,a,b)}\left[V_{h+1}^{(t)}(s')\right] \\ V_h^{(t+1)}(s) &= (1-\alpha_{t+1})V_h^{(t)}(s) \\ &\quad + \alpha_{t+1}\left[\bar{\mu}_h^{(t+1)}(s)^\top Q_h^{(t+1)}(s)\bar{\nu}_h^{(t+1)}(s) + \tau\mathcal{H}\big(\bar{\mu}_h^{(t+1)}(s)\big) - \tau\mathcal{H}\big(\bar{\nu}_h^{(t+1)}(s)\big)\right] \end{cases} \quad .$$
$$(21)$$

---

*Proof.* See Appendix D.2.              $\square$

We prove Theorem 2 by induction. By definition, we have

$$\left\|Q_{H,\tau}^{\star} - Q_H^{(0)}\right\|_{\infty} = \left\|Q_{H,\tau}^{\star}\right\|_{\infty} \leq 1,$$

and $\left\|Q_{H,\tau}^{\star} - Q_H^{(t)}\right\|_{\infty} = \left\|r_H - r_H\right\|_{\infty} = 0$ for $t > 0$. So (13a) holds trivially for $h = H$. When the statement holds for some $h$, we can invoke Lemma 10 with $t_1 = T_h + 1$ and $t_2 = t \geq T_{h-1}$, which yields

$$\left\|Q_{h-1}^{(t)} - Q_{h-1,\tau}^{\star}\right\|$$

$$\leq 2(1-\eta\tau)^{t-T_h-1}H + 10\eta\tau \underset{s'\sim P(\cdot|s,a,b)}{\mathbb{E}}\left[\sum_{l=T_h}^{t-1}(1-\eta\tau)^{t-1-l}\big\|Q_h^{(l)}(s) - Q_{h,\tau}^{\star}(s)\big\|_{\infty}\right]$$

$$\quad + \tau(1-\eta\tau)^{t-T_h-1} \underset{s'\sim P(\cdot|s,a,b)}{\mathbb{E}}\left[\mathsf{KL}_s\big(\zeta_{h,\tau}^{\star} \| \zeta_h^{(T_h)}\big) + (1-4\eta H)\mathsf{KL}_s\big(\zeta_h^{(T_h)} \| \bar{\zeta}_h^{(T_h)}\big)\right]$$

$$\leq 2(1-\eta\tau)^{t-T_h-1}H + 10\eta\tau \underset{s'\sim P(\cdot|s,a,b)}{\mathbb{E}}\left[\sum_{l=T_h}^{t-1}(1-\eta\tau)^{t-T_h-1}l^{H-h}\right]$$

$$\quad + \tau(1-\eta\tau)^{t-T_h-1} \underset{s'\sim P(\cdot|s,a,b)}{\mathbb{E}}\left[\mathsf{KL}_s\big(\zeta_{h,\tau}^{\star} \| \zeta_h^{(T_h)}\big) + (1-4\eta H)\mathsf{KL}_s\big(\zeta_h^{(T_h)} \| \bar{\zeta}_h^{(T_h)}\big)\right]$$

$$\leq (1-\eta\tau)^{t-T_{h-1}}(1-\eta\tau)^{T_{\text{start}}-1}\left[10H + 10\eta\tau t^{H-h+1}\right],$$

where the last step results from

$$\tau\left(\mathsf{KL}_s\big(\zeta_{h,\tau}^{\star} \| \zeta_h^{(T_h)}\big) + (1-4\eta H)\mathsf{KL}_s\big(\zeta_h^{(T_h)} \| \bar{\zeta}_h^{(T_h)}\big)\right)$$

$$\leq \tau \Big( \big\| \log \mu_{h,\tau}^\star(s) - \log \mu_h^{(T_h)}(s) \big\|_\infty + \big\| \log \nu_{h,\tau}^\star(s) - \log \nu_h^{(T_h)}(s) \big\|_\infty$$

$$+ \big\| \log \mu_h^{(T_h)}(s) - \log \bar{\mu}_h^{(T_h)}(s) \big\|_\infty + \big\| \log \nu_h^{(T_h)}(s) - \log \bar{\nu}_h^{(T_h)}(s) \big\|_\infty \Big)$$

$$\leq 8H.$$

Therefore, with $T_{\text{start}} = \lceil \frac{1}{\eta\tau} \log H \rceil$ we can guarantee that

$$\big\| Q_{h-1}^{(t)} - Q_{h-1,\tau}^\star \big\| \leq 10(1-\eta\tau)^{t-T_{h-1}}(1-\eta\tau)^{T_{\text{start}}-1} \Big[ H + \eta\tau t^{H-h+1} \Big]$$

$$\leq (1-\eta\tau)^{t-T_{h-1}} t^{H-h+1}.$$

This completes the proof for (13a). Regarding (13b), we start by the following lemmas, which are simply Lemma 5 and Lemma 7 applied to the episodic setting:

**Lemma 5A.** *With $0 < \eta \leq \frac{1}{8H}$, we have*

$$\frac{1}{2}\mathsf{KL}_s\big(\zeta_{h,\tau}^\star \,\|\, \bar{\zeta}_h^{(t+1)}\big) + \eta\tau\mathsf{KL}_s\big(\bar{\zeta}_h^{(t+1)} \,\|\, \zeta_{h,\tau}^\star\big)$$

$$\leq (1-\eta\tau)\mathsf{KL}_s\big(\zeta_{h,\tau}^\star \,\|\, \zeta_h^{(t)}\big) + 2\eta H\mathsf{KL}_s\big(\zeta_h^{(t)} \,\|\, \bar{\zeta}_h^{(t)}\big) + 2\eta\big\|Q_h^{(t)}(s) - Q_{h,\tau}^\star(s)\big\|_\infty.$$

**Lemma 7A.** *It holds for all $h \in [H]$, $s \in \mathcal{S}$ and policy pair $\mu, \nu$ that*

$$\max_{\mu',\nu'} \big( f_s(Q_{h,\tau}^\star, \mu_h', \nu_h) - f_s(Q_\tau^\star, \mu_h, \nu_h') \big) \leq \frac{4H^2}{\tau}\mathsf{KL}_s\big(\zeta_{h,\tau}^\star \,\|\, \zeta_h\big) + \tau\mathsf{KL}_s\big(\zeta_h \,\|\, \zeta_{h,\tau}^\star\big).$$

Combining Lemma 9 with Lemma 5A and Lemma 7A, we conclude that for $0 \leq t_1 \leq t_2 - 1$,

$$\max_{\mu,\nu} \big( f_s(Q_{h,\tau}^\star, \mu_h, \bar{\nu}_h^{(t_2)}) - f_s(Q_\tau^\star, \bar{\mu}_h^{(t_2)}, \nu_h) \big)$$

$$\leq \frac{4H^2}{\tau}\mathsf{KL}_s\big(\zeta_{h,\tau}^\star \,\|\, \bar{\zeta}_h^{(t_2)}\big) + \tau\mathsf{KL}_s\big(\bar{\zeta}_h^{(t_2)} \,\|\, \zeta_{h,\tau}^\star\big)$$

$$\leq \max\Big\{ \frac{8H^2}{\tau}, \frac{1}{\eta} \Big\} \Big( \frac{1}{2}\mathsf{KL}_s\big(\zeta_{h,\tau}^\star \,\|\, \bar{\zeta}_h^{(t_2)}\big) + \eta\tau\mathsf{KL}_s\big(\bar{\zeta}_h^{(t_2)} \,\|\, \zeta_{h,\tau}^\star\big) \Big)$$

$$\leq \max\Big\{ \frac{8H^2}{\tau}, \frac{1}{\eta} \Big\} \Big( (1-\eta\tau)\mathsf{KL}_s\big(\zeta_{h,\tau}^\star \,\|\, \zeta_h^{(t_2-1)}\big) + 2\eta H\mathsf{KL}_s\big(\zeta_h^{(t_2-1)} \,\|\, \bar{\zeta}_h^{(t_2-1)}\big) + 2\eta\big\|Q_h^{(t_2-1)}(s) - Q_{h,\tau}^\star(s)\big\|_\infty \Big)$$

$$\leq \max\Big\{ \frac{8H^2}{\tau}, \frac{1}{\eta} \Big\} \Big( (1-\eta\tau)^{t_2-t_1} \Big( \mathsf{KL}_s\big(\zeta_{h,\tau}^\star \,\|\, \zeta_h^{(t_1)}\big) + (1-4\eta H)\mathsf{KL}_s\big(\zeta_h^{(t_1)} \,\|\, \bar{\zeta}_h^{(t_1)}\big) \Big)$$

$$+ 6\eta \sum_{l=t_1}^{t_2} (1-\eta\tau)^{t_2-l}\big\|Q_h^{(l)}(s) - Q_{h,\tau}^\star(s)\big\|_\infty \Big).$$

It is straightforward to verify that the above inequality holds for $0 \leq t_1 \leq t_2$, by omitting the third step. Substitution of (13a) into the above inequality yields

$$\max_{\mu,\nu} \big( f_s(Q_{h,\tau}^\star, \mu_h, \bar{\nu}_h^{(t)}) - f_s(Q_\tau^\star, \bar{\mu}_h^{(t)}, \nu_h) \big)$$

$$\leq \max\Big\{ \frac{8H^2}{\tau}, \frac{1}{\eta} \Big\} \Big( (1-\eta\tau)^{t-T_h} \Big( \mathsf{KL}_s\big(\zeta_{h,\tau}^\star \,\|\, \zeta_h^{(T_h)}\big) + (1-4\eta H)\mathsf{KL}_s\big(\zeta_h^{(T_h)} \,\|\, \bar{\zeta}_h^{(T_h)}\big) \Big)$$

$$+ 6\eta \sum_{l=T_h}^{t} (1-\eta\tau)^{t-l}(1-\eta\tau)^{l-T_h} l^{H-h} \Big)$$

$$\leq (1-\eta\tau)^{t-T_h} \max\Big\{ \frac{8H^2}{\tau}, \frac{1}{\eta} \Big\} \Big( \frac{8H}{\tau} + 6\eta t^{H-h+1} \Big). \tag{24}$$

We prove the following results instead, where (13b) is a direct conclusion of (25) by summing the two inequalities.

$$\begin{cases} \max_{s\in\mathcal{S},\mu} \big( V_{h,\tau}^{\mu,\bar{\nu}^{(t)}}(s) - V_{h,\tau}^\star(s) \big) \leq 2(1-\eta\tau)^{t-T_h} \max\Big\{ \frac{8H^2}{\tau}, \frac{1}{\eta} \Big\} \Big( \frac{8H}{\tau} + 6\eta t^{H-h+1} \Big) \\ \max_{s\in\mathcal{S},\mu} \big( V_{h,\tau}^\star(s) - V_{h,\tau}^{\bar{\mu}^{(t)},\nu}(s) \big) \leq 2(1-\eta\tau)^{t-T_h} \max\Big\{ \frac{8H^2}{\tau}, \frac{1}{\eta} \Big\} \Big( \frac{8H}{\tau} + 6\eta t^{H-h+1} \Big) \end{cases} \tag{25}$$

We prove by induction. Note that when $h = H$, we have $V_{H,\tau}^{\mu,\nu}(s) = f_s(r_H, \mu_H, \nu_H) = f_s(Q_{H,\tau}^\star, \mu_H, \nu_H)$ and the claim holds by invoking (24). When the claim holds for some $2 \leq h \leq H$, we have

$$V_{h-1,\tau}^{\mu,\bar{\nu}^{(t)}}(s) - V_{h-1,\tau}^\star(s)$$

$$= \mu_{h-1}(s)^\top Q_{h-1,\tau}^{\mu,\bar{\nu}^{(t)}}(s)\bar{\nu}_{h-1}^{(t)}(s) + \tau\mathcal{H}\big(\mu_{h-1}(s)\big) - \tau\mathcal{H}\big(\bar{\nu}_{h-1}^{(t)}(s)\big)$$
$$\quad - \mu_{h-1,\tau}^\star(s)^\top Q_{h-1,\tau}^\star(s)\nu_{h-1,\tau}^\star(s) + \tau\mathcal{H}\big(\mu_{h-1,\tau}^\star(s)\big) - \tau\mathcal{H}\big(\nu_{h-1,\tau}^\star(s)\big)$$

$$= f_s(Q_{h-1,\tau}^\star, \mu_{h-1}, \bar{\nu}_{h-1}^{(t)}) - f_s(Q_{h-1,\tau}^\star, \mu_{h-1,\tau}^\star, \nu_{h-1,\tau}^\star) + \mu_{h-1}(s)^\top\big(Q_{h-1,\tau}^{\mu,\bar{\nu}^{(t)}}(s) - Q_{h-1,\tau}^\star(s)\big)\bar{\nu}_{h-1}^{(t)}(s)$$

$$\leq f_s(Q_{h-1,\tau}^\star, \mu_{h-1}, \bar{\nu}_{h-1}^{(t)}) - f_s(Q_{h-1,\tau}^\star, \bar{\mu}_{h-1}^{(t)}, \nu_{h-1,\tau}^\star) + \max_{s'\in\mathcal{S}}\Big[V_{h,\tau}^{\mu,\bar{\nu}^{(t)}}(s') - V_{h,\tau}^\star(s')\Big]$$

$$\leq \max_{\mu'_{h-1},\nu'_{h-1}} \Big(f_s(Q_{h-1,\tau}^\star, \mu'_{h-1}, \bar{\nu}_{h-1}^{(t)}) - f_s(Q_{h-1,\tau}^\star, \bar{\mu}_{h-1}^{(t)}, \nu'_{h-1})\Big) + \max_{s'\in\mathcal{S}}\Big[V_{h,\tau}^{\mu,\bar{\nu}^{(t)}}(s') - V_{h,\tau}^\star(s')\Big]$$

$$\leq (1-\eta\tau)^{t-T_{h-1}}\max\Big\{\frac{8H^2}{\tau}, \frac{1}{\eta}\Big\}\Big(\frac{8H}{\tau} + 6\eta t^{H-h+2}\Big)$$
$$\quad + 2(1-\eta\tau)^{t-T_h}\max\Big\{\frac{8H^2}{\tau}, \frac{1}{\eta}\Big\}\Big(\frac{8H}{\tau} + 6\eta t^{H-h+1}\Big)$$

$$\leq 2(1-\eta\tau)^{t-T_{h-1}}\max\Big\{\frac{8H^2}{\tau}, \frac{1}{\eta}\Big\}\Big(\frac{8H}{\tau} + 6\eta t^{H-h+2}\Big).$$

Taking maximum over $\mu$ verifies the claim for $h-1$, thereby finishing the proof. The bound for $\max_{s\in\mathcal{S},\mu}\Big(V_{h,\tau}^\star(s) - V_{h,\tau}^{\bar{\mu}^{(t)},\nu}(s)\Big)$ can be established by following a similar argument and is therefore omitted.

## C  PROOF OF KEY LEMMAS FOR THE DISCOUNTED SETTING

### C.1  PROOF OF LEMMA 1

Before proceeding, we shall introduce the following lemma that quantifies the distance between two consecutive updates, whose proof can be found in Appendix E.1.

**Lemma 11.** *For $0 < \eta \leq 1/\tau$, it holds for all $s \in \mathcal{S}$ and $t \geq 0$ that*

$$\max\big\{\big\|\bar{\mu}^{(t+1)}(s) - \mu^{(t+1)}(s)\big\|_1, \big\|\bar{\nu}^{(t+1)}(s) - \nu^{(t+1)}(s)\big\|_1\big\} \leq \frac{2\eta}{1-\gamma}$$

*and that*

$$\max\big\{\big\|\bar{\mu}^{(t+1)}(s) - \bar{\mu}^{(t)}(s)\big\|_1, \big\|\bar{\nu}^{(t+1)}(s) - \bar{\nu}^{(t)}(s)\big\|_1\big\} \leq \frac{6\eta}{1-\gamma}.$$

For notational simplicity, we use $x \overset{\mathbf{1}}{=} y$ to denote equivalence up to a global shift for two vectors $x, y$:

$$x = y + c \cdot \mathbf{1}$$

for some constant $c \in \mathbb{R}$. Taking logarithm on the both sides of the update rule (9a), we get

$$\begin{cases} \log\mu^{(t+1)}(s) - (1-\eta\tau)\log\mu^{(t)}(s) & \overset{\mathbf{1}}{=} \eta Q^{(t+1)}(s)\bar{\nu}^{(t+1)}(s) \\ \log\nu^{(t+1)}(s) - (1-\eta\tau)\log\nu^{(t)}(s) & \overset{\mathbf{1}}{=} -\eta Q^{(t+1)}(s)^\top\bar{\mu}^{(t+1)}(s) \end{cases}. \tag{26}$$

On the other hand, it holds for the optimal policies $(\mu_\tau^\star, \nu_\tau^\star)$ that

$$\begin{cases} \eta\tau\log\mu_\tau^\star(s) & \overset{\mathbf{1}}{=} \eta Q_\tau^\star(s)\nu_\tau^\star(s) \\ \eta\tau\log\nu_\tau^\star(s) & \overset{\mathbf{1}}{=} -\eta Q_\tau^\star(s)^\top\mu_\tau^\star(s) \end{cases}. \tag{27}$$

Subtracting (27) from (26) and taking inner product with $\bar{\zeta}^{(t+1)}(s) - \zeta_\tau^\star(s)$ gives

$$\langle \log \zeta^{(t+1)}(s) - (1 - \eta\tau) \log \zeta^{(t)}(s) - \eta\tau \log \zeta_\tau^\star(s), \bar{\zeta}^{(t+1)}(s) - \zeta_\tau^\star(s) \rangle$$

$$= \eta \langle \bar{\mu}^{(t+1)}(s) - \mu_\tau^\star(s), Q^{(t+1)}(s)\bar{\nu}^{(t+1)}(s) - Q_\tau^\star(s)\nu_\tau^\star(s) \rangle$$

$$\quad - \eta \langle \bar{\nu}^{(t+1)}(s) - \nu_\tau^\star(s), Q^{(t+1)}(s)^\top \bar{\mu}^{(t+1)}(s) - Q_\tau^\star(s)^\top \mu_\tau^\star(s) \rangle$$

$$= \eta \langle \bar{\mu}^{(t+1)}(s) - \mu_\tau^\star(s), (Q^{(t+1)}(s) - Q_\tau^\star(s))\bar{\nu}^{(t+1)}(s) \rangle$$

$$\quad - \eta \langle \bar{\nu}^{(t+1)}(s) - \nu_\tau^\star(s), (Q^{(t+1)}(s) - Q_\tau^\star(s))^\top \bar{\mu}^{(t+1)}(s) \rangle$$

$$= -\eta \langle \mu_\tau^\star(s), (Q^{(t+1)}(s) - Q_\tau^\star(s))\bar{\nu}^{(t+1)}(s) \rangle + \eta \langle \nu_\tau^\star(s), (Q^{(t+1)}(s) - Q_\tau^\star(s))^\top \bar{\mu}^{(t+1)}(s) \rangle$$

$$\leq 2\eta \big\| Q^{(t+1)}(s) - Q_\tau^\star(s) \big\|_\infty.$$

$$(28)$$

We rewrite the LHS as

$$\langle \log \zeta^{(t+1)}(s) - (1 - \eta\tau) \log \zeta^{(t)}(s) - \eta\tau \log \zeta_\tau^\star(s), \bar{\zeta}^{(t+1)}(s) - \zeta_\tau^\star(s) \rangle$$

$$= -\langle \log \zeta^{(t+1)}(s) - (1 - \eta\tau) \log \zeta^{(t)}(s) - \eta\tau \log \zeta_\tau^\star(s), \zeta_\tau^\star(s) \rangle$$

$$\quad + \langle \log \bar{\zeta}^{(t+1)}(s) - (1 - \eta\tau) \log \bar{\zeta}^{(t)}(s) - \eta\tau \log \zeta_\tau^\star(s), \bar{\zeta}^{(t+1)}(s) \rangle$$

$$\quad + \langle \log \zeta^{(t+1)}(s) - \log \bar{\zeta}^{(t+1)}(s), \bar{\zeta}^{(t+1)}(s) \rangle$$

$$\quad - (1 - \eta\tau)\langle \log \zeta^{(t)}(s) - \log \bar{\zeta}^{(t)}(s), \bar{\zeta}^{(t+1)}(s) \rangle$$

$$= \mathsf{KL}_s\big(\zeta_\tau^\star \,\|\, \zeta^{(t+1)}\big) - (1 - \eta\tau)\mathsf{KL}_s\big(\zeta_\tau^\star \,\|\, \zeta^{(t)}\big)$$

$$\quad + (1 - \eta\tau)\mathsf{KL}_s\big(\bar{\zeta}^{(t+1)} \,\|\, \bar{\zeta}^{(t)}\big) + \eta\tau\mathsf{KL}_s\big(\bar{\zeta}^{(t+1)} \,\|\, \zeta_\tau^\star\big)$$

$$\quad + \mathsf{KL}_s\big(\zeta^{(t+1)} \,\|\, \bar{\zeta}^{(t+1)}\big) - \langle \log \bar{\zeta}^{(t+1)}(s) - \log \zeta^{(t+1)}(s), \bar{\zeta}^{(t+1)}(s) - \zeta^{(t+1)}(s) \rangle$$

$$\quad + (1 - \eta\tau)\mathsf{KL}_s\big(\bar{\zeta}^{(t)} \,\|\, \zeta^{(t)}\big) - (1 - \eta\tau)\langle \log \zeta^{(t)}(s) - \log \bar{\zeta}^{(t)}(s), \bar{\zeta}^{(t+1)}(s) - \bar{\zeta}^{(t)}(s) \rangle.$$

Rearranging terms, we have

$$\mathsf{KL}_s\big(\zeta_\tau^\star \,\|\, \zeta^{(t+1)}\big) - (1 - \eta\tau)\mathsf{KL}_s\big(\zeta_\tau^\star \,\|\, \zeta^{(t)}\big) + (1 - \eta\tau)\mathsf{KL}_s\big(\bar{\zeta}^{(t+1)} \,\|\, \bar{\zeta}^{(t)}\big)$$

$$\quad + \eta\tau\mathsf{KL}_s\big(\bar{\zeta}^{(t+1)} \,\|\, \zeta_\tau^\star\big) + \mathsf{KL}_s\big(\zeta^{(t+1)} \,\|\, \bar{\zeta}^{(t+1)}\big) + (1 - \eta\tau)\mathsf{KL}_s\big(\bar{\zeta}^{(t)} \,\|\, \zeta^{(t)}\big)$$

$$\quad - \langle \log \bar{\zeta}^{(t+1)}(s) - \log \zeta^{(t+1)}(s), \bar{\zeta}^{(t+1)}(s) - \zeta^{(t+1)}(s) \rangle$$

$$\quad - (1 - \eta\tau)\langle \log \zeta^{(t)}(s) - \log \bar{\zeta}^{(t)}(s), \bar{\zeta}^{(t+1)}(s) - \bar{\zeta}^{(t)}(s) \rangle$$

$$\quad \leq 2\eta \big\| Q^{(t+1)}(s) - Q_\tau^\star(s) \big\|_\infty.$$

It remains to bound $\langle \log \bar{\zeta}^{(t+1)}(s) - \log \zeta^{(t+1)}(s), \bar{\zeta}^{(t+1)}(s) - \zeta^{(t+1)}(s) \rangle$ and $\langle \log \zeta^{(t)}(s) - \log \bar{\zeta}^{(t)}(s), \bar{\zeta}^{(t+1)}(s) - \bar{\zeta}^{(t)}(s) \rangle$. Note that

$$\langle \log \bar{\mu}^{(t+1)}(s) - \log \mu^{(t+1)}(s), \bar{\mu}^{(t+1)}(s) - \mu^{(t+1)}(s) \rangle$$

$$= \eta \langle Q^{(t)}(s)\bar{\nu}^{(t)}(s) - Q^{(t+1)}(s)\bar{\nu}^{(t+1)}(s), \bar{\mu}^{(t+1)}(s) - \mu^{(t+1)}(s) \rangle \quad (29)$$

$$\leq \eta \big\| Q^{(t)}(s)\bar{\nu}^{(t)}(s) - Q^{(t+1)}(s)\bar{\nu}^{(t+1)}(s) \big\|_1 \big\| \bar{\mu}^{(t+1)}(s) - \mu^{(t+1)}(s) \big\|_1.$$

We bound $\big\| Q^{(t)}(s)\bar{\nu}^{(t)}(s) - Q^{(t+1)}(s)\bar{\nu}^{(t+1)}(s) \big\|_1$ as

$$\big\| Q^{(t)}(s)\bar{\nu}^{(t)}(s) - Q^{(t+1)}(s)\bar{\nu}^{(t+1)}(s) \big\|_1$$

$$\leq \big\| Q^{(t+1)}(s)\big(\bar{\nu}^{(t)}(s) - \bar{\nu}^{(t+1)}(s)\big) \big\|_1 + \big\| \big(Q^{(t)}(s) - Q^{(t+1)}(s)\big)\bar{\nu}^{(t)}(s) \big\|_1$$

$$\leq \frac{2}{1 - \gamma} \big\| \bar{\nu}^{(t)}(s) - \bar{\nu}^{(t+1)}(s) \big\|_1 + \big\| Q^{(t)}(s) - Q^{(t+1)}(s) \big\|_\infty.$$

Plugging the above inequality into (29) and invoking Young's inequality yields

$$\langle \log \bar{\mu}^{(t+1)}(s) - \log \mu^{(t+1)}(s), \bar{\mu}^{(t+1)}(s) - \mu^{(t+1)}(s) \rangle$$

$$\leq \frac{\eta}{1 - \gamma} \Big( \big\| \bar{\nu}^{(t+1)}(s) - \bar{\nu}^{(t)}(s) \big\|_1^2 + \big\| \bar{\mu}^{(t+1)}(s) - \mu^{(t+1)}(s) \big\|_1^2 \Big)$$

$$+ \eta \big\| Q^{(t)}(s) - Q^{(t+1)}(s) \big\|_\infty \big\| \bar{\mu}^{(t+1)}(s) - \mu^{(t+1)}(s) \big\|_1$$

$$\leq \frac{2\eta}{1-\gamma} \mathsf{KL}_s \big( \bar{\nu}^{(t+1)} \,\|\, \bar{\nu}^{(t)} \big) + \frac{2\eta}{1-\gamma} \mathsf{KL}_s \big( \mu^{(t+1)} \,\|\, \bar{\mu}^{(t+1)} \big) + \frac{2\eta^2}{1-\gamma} \big\| Q^{(t)}(s) - Q^{(t+1)}(s) \big\|_\infty,$$

where the last step results from Pinsker's inequality and Lemma 11. Similarly, we have

$$\big\langle \log \bar{\nu}^{(t+1)}(s) - \log \nu^{(t+1)}(s), \bar{\nu}^{(t+1)}(s) - \nu^{(t+1)}(s) \big\rangle$$

$$\leq \frac{2\eta}{1-\gamma} \mathsf{KL}_s \big( \bar{\mu}^{(t+1)} \,\|\, \bar{\mu}^{(t)} \big) + \frac{2\eta}{1-\gamma} \mathsf{KL}_s \big( \nu^{(t+1)} \,\|\, \bar{\nu}^{(t+1)} \big) + \frac{2\eta^2}{1-\gamma} \big\| Q^{(t)}(s) - Q^{(t+1)}(s) \big\|_\infty.$$

Combining the above two inequalities gives

$$\big\langle \log \bar{\zeta}^{(t+1)}(s) - \log \zeta^{(t+1)}(s), \bar{\zeta}^{(t+1)}(s) - \zeta^{(t+1)}(s) \big\rangle$$

$$\leq \frac{2\eta}{1-\gamma} \mathsf{KL}_s \big( \bar{\zeta}^{(t+1)} \,\|\, \bar{\zeta}^{(t)} \big) + \frac{2\eta}{1-\gamma} \mathsf{KL}_s \big( \zeta^{(t+1)} \,\|\, \bar{\zeta}^{(t+1)} \big) + \frac{4\eta^2}{1-\gamma} \big\| Q^{(t)}(s) - Q^{(t+1)}(s) \big\|_\infty.$$

By a similar argument, when $t \geq 1$:

$$\big\langle \log \zeta^{(t)}(s) - \log \bar{\zeta}^{(t)}(s), \bar{\zeta}^{(t+1)}(s) - \bar{\zeta}^{(t)}(s) \big\rangle$$

$$= \eta \big\langle Q^{(t)}(s)\bar{\nu}^{(t)}(s) - Q^{(t-1)}(s)\bar{\nu}^{(t-1)}(s), \bar{\mu}^{(t+1)}(s) - \bar{\mu}^{(t)}(s) \big\rangle$$

$$\quad - \eta \big\langle Q^{(t)}(s)^\top \bar{\mu}^{(t)}(s) - Q^{(t-1)}(s)^\top \bar{\mu}^{(t-1)}(s), \bar{\nu}^{(t+1)}(s) - \bar{\nu}^{(t)}(s) \big\rangle$$

$$\leq \frac{2\eta}{1-\gamma} \mathsf{KL}_s \big( \bar{\zeta}^{(t)} \,\|\, \bar{\zeta}^{(t-1)} \big) + \frac{2\eta}{1-\gamma} \mathsf{KL}_s \big( \bar{\zeta}^{(t+1)} \,\|\, \bar{\zeta}^{(t)} \big)$$

$$\quad + \eta \big( \big\| \bar{\mu}^{(t+1)}(s) - \bar{\mu}^{(t)}(s) \big\|_1 + \big\| \bar{\nu}^{(t+1)}(s) - \bar{\nu}^{(t)}(s) \big\|_1 \big) \big\| Q^{(t)}(s) - Q^{(t-1)}(s) \big\|_\infty$$

$$\leq \frac{2\eta}{1-\gamma} \mathsf{KL}_s \big( \bar{\zeta}^{(t)} \,\|\, \bar{\zeta}^{(t-1)} \big) + \frac{2\eta}{1-\gamma} \mathsf{KL}_s \big( \bar{\zeta}^{(t+1)} \,\|\, \bar{\zeta}^{(t)} \big)$$

$$\quad + \frac{12\eta^2}{1-\gamma} \big\| Q^{(t)}(s) - Q^{(t-1)}(s) \big\|_\infty.$$

Note that the above inequality trivially holds for $t = 0$, since $\log \zeta^{(0)}(s) = \log \bar{\zeta}^{(0)}(s)$.

Putting pieces together, we conclude for that

$$\mathsf{KL}_s \big( \zeta_\tau^\star \,\|\, \zeta^{(t+1)} \big) - (1 - \eta\tau) \mathsf{KL}_s \big( \zeta_\tau^\star \,\|\, \zeta^{(t)} \big) + \Big( 1 - \eta\tau - \frac{4\eta}{1-\gamma} \Big) \mathsf{KL}_s \big( \bar{\zeta}^{(t+1)} \,\|\, \bar{\zeta}^{(t)} \big)$$

$$\quad + \eta\tau \mathsf{KL}_s \big( \bar{\zeta}^{(t+1)} \,\|\, \zeta_\tau^\star \big) + \Big( 1 - \frac{2\eta}{1-\gamma} \Big) \mathsf{KL}_s \big( \zeta^{(t+1)} \,\|\, \bar{\zeta}^{(t+1)} \big) + (1 - \eta\tau) \mathsf{KL}_s \big( \bar{\zeta}^{(t)} \,\|\, \zeta^{(t)} \big)$$

$$\quad - \frac{2\eta}{1-\gamma} \mathsf{KL}_s \big( \bar{\zeta}^{(t)} \,\|\, \bar{\zeta}^{(t-1)} \big)$$

$$\leq 2\eta \big\| Q^{(t+1)}(s) - Q_\tau^\star(s) \big\|_\infty + \frac{4\eta^2}{1-\gamma} \big\| Q^{(t)}(s) - Q^{(t+1)}(s) \big\|_\infty + \frac{12\eta^2}{1-\gamma} \big\| Q^{(t-1)}(s) - Q^{(t)}(s) \big\|_\infty.$$

Averaging $s$ over the distribution $\rho$ completes the proof.

## C.2 Proof of Lemma 2

*Proof.* By definition of $Q$, it holds for $t \geq 1$ that

$$\big| Q^{(t+1)}(s, a, b) - Q^{(t)}(s, a, b) \big| \leq \gamma \mathbb{E}_{s' \sim P(\cdot | s, a, b)} \Big[ \big| V^{(t)}(s') - V^{(t-1)}(s') \big| \Big]. \tag{30}$$

We denote by $f_s(Q, \mu, \nu)$ the one-step entropy-regularized game value at state $s$, i.e.,

$$f_s(Q, \mu, \nu) = \mu(s)^\top Q(s)\nu(s) + \tau \mathcal{H}(\mu(s)) - \tau \mathcal{H}(\nu(s)).$$

We further simplify the notation by introducing

$$f_s^{(t)} = f_s(Q^{(t)}, \bar{\mu}^{(t)}, \bar{\nu}^{(t)})$$

By recursively applying the update rule $V^{(t)}(s) = (1 - \alpha_t)V^{(t-1)}(s) + \alpha_t f_s^{(t)}$, we get

$$V^{(t)}(s) = \alpha_{0,t}V^{(0)} + \sum_{l=1}^{t} \alpha_{l,t} f_s(Q^{(l)}, \bar{\mu}^{(l)}, \bar{\nu}^{(l)}) = \sum_{l=0}^{t} \alpha_{l,t} f_s^{(l)}.$$

Since $\alpha_0 = 1$, it follows that

$$\sum_{l=0}^{t} \alpha_{l,t} = \alpha_0 = 1$$

So we have

$$
\begin{aligned}
\left|V^{(t)}(s) - V^{(t-1)}(s)\right| &= \alpha_t \left|f_s^{(t)} - V^{(t-1)}(s)\right| \\
&= \alpha_t \sum_{l=0}^{t-1} \alpha_{l,t-1} \left|f_s^{(t)} - f_s^{(l)}\right| \\
&\leq \alpha_t \sum_{l=0}^{t-1} \alpha_{l,t-1} \sum_{j=l}^{t-1} \left|f_s^{(j+1)} - f_s^{(j)}\right|
\end{aligned}
\tag{31}
$$

The next lemma enables us to upper bound $\left|f_s^{(t+1)} - f_s^{(t)}\right|$ with $\left\|Q^{(t+1)}(s) - Q^{(t)}(s)\right\|_\infty$ and $\mathsf{KL}_s\big(\bar{\zeta}^{(t+1)} \,\|\, \bar{\zeta}^{(t)}\big)$ (as well as their $(t-1)-$th iteration counter parts). The proof is postponed to Appendix E.2.

**Lemma 12.** *For any $t \geq 0$, $\eta \leq (1-\gamma)/180$, we have*

$$
\begin{aligned}
\left|f_s^{(t+1)} - f_s^{(t)}\right| &\leq \left\|Q^{(t+1)}(s) - Q^{(t)}(s)\right\|_\infty + \left(\frac{3}{\eta} + \frac{4}{1-\gamma}\right) \mathsf{KL}_s\big(\bar{\zeta}^{(t+1)} \,\|\, \bar{\zeta}^{(t)}\big) \\
&\quad + \frac{12\eta}{1-\gamma}\left\|Q^{(t)}(s) - Q^{(t-1)}(s)\right\|_\infty + \frac{2}{1-\gamma}\mathsf{KL}_s\big(\bar{\zeta}^{(t)} \,\|\, \bar{\zeta}^{(t-1)}\big).
\end{aligned}
$$

Plugging the above lemma into (31),

$$
\begin{aligned}
&\left|V^{(t)}(s) - V^{(t-1)}(s)\right| \\
&\leq \alpha_t \sum_{l=0}^{t-1} \alpha_{l,t-1} \sum_{j=l}^{t-1} \left[\left\|Q^{(j+1)}(s) - Q^{(j)}(s)\right\|_\infty + \left(\frac{3}{\eta} + \frac{4}{1-\gamma}\right)\mathsf{KL}_s\big(\bar{\zeta}^{(j+1)} \,\|\, \bar{\zeta}^{(j)}\big)\right] \\
&\quad + \alpha_t \sum_{l=0}^{t-1} \alpha_{l,t-1} \sum_{j=l}^{t-1} \left[\frac{12\eta}{1-\gamma}\left\|Q^{(j)}(s) - Q^{(j-1)}(s)\right\|_\infty + \frac{2}{1-\gamma}\mathsf{KL}_s\big(\bar{\zeta}^{(j)} \,\|\, \bar{\zeta}^{(j-1)}\big)\right] \\
&\leq \alpha_t \sum_{l=0}^{t-1} \alpha_{l,t-1} \sum_{j=l}^{t-1} \left[\left(1 + \frac{12\eta}{1-\gamma}\right)\left\|Q^{(j+1)}(s) - Q^{(j)}(s)\right\|_\infty + \left(\frac{3}{\eta} + \frac{6}{1-\gamma}\right)\mathsf{KL}_s\big(\bar{\zeta}^{(j+1)} \,\|\, \bar{\zeta}^{(j)}\big)\right] \\
&\quad + \alpha_t \sum_{l=0}^{t-1} \alpha_{l,t-1}\left[\frac{12\eta}{1-\gamma}\left\|Q^{(l)}(s) - Q^{(l-1)}(s)\right\|_\infty + \frac{2}{1-\gamma}\mathsf{KL}_s\big(\bar{\zeta}^{(l)} \,\|\, \bar{\zeta}^{(l-1)}\big)\right] \\
&\leq \sum_{j=0}^{t-1} \alpha_{j+1} \sum_{l=0}^{j} \alpha_{l,t-1}\left[\left(1 + \frac{12\eta}{1-\gamma}\right)\left\|Q^{(j+1)}(s) - Q^{(j)}(s)\right\|_\infty + \left(\frac{3}{\eta} + \frac{6}{1-\gamma}\right)\mathsf{KL}_s\big(\bar{\zeta}^{(j+1)} \,\|\, \bar{\zeta}^{(j)}\big)\right] \\
&\quad + \alpha_t \sum_{l=0}^{t-2} \alpha_{l+1,t-1}\left[\frac{12\eta}{1-\gamma}\left\|Q^{(l+1)}(s) - Q^{(l)}(s)\right\|_\infty + \frac{2}{1-\gamma}\mathsf{KL}_s\big(\bar{\zeta}^{(l+1)} \,\|\, \bar{\zeta}^{(l)}\big)\right],
\end{aligned}
$$

where the last step is due to $\alpha_t \leq \alpha_j$ for all $j \leq t$. To continue, by definition of $\alpha$ we have $\alpha_t \alpha_{l+1,t-1} \leq \alpha_{l+1,t-1}(1 - \alpha_t) = \alpha_{l+1,t}$ for $0 \leq l < t$, and that

$$\alpha_{j+1} \sum_{l=0}^{j} \alpha_{l,t-1} = \alpha_{j+1} \sum_{l=0}^{j}\left(\prod_{i=l+1}^{t-1}(1 - \alpha_i) - \prod_{i=l}^{t-1}(1 - \alpha_i)\right)$$

$$= \alpha_{j+1} \prod_{i=j+1}^{t-1} (1 - \alpha_i)$$

$$\leq \alpha_{j+1} \prod_{i=j+2}^{t} (1 - \alpha_i) = \alpha_{j+1,t}.$$

Plugging into the inequality above gives

$$\left| V^{(t)}(s) - V^{(t-1)}(s) \right|$$

$$\leq \sum_{j=0}^{t-1} \alpha_{j+1,t} \left[ \left(1 + \frac{12\eta}{1-\gamma}\right) \left\| Q^{(j+1)}(s) - Q^{(j)}(s) \right\|_\infty + \left(\frac{3}{\eta} + \frac{6}{1-\gamma}\right) \mathsf{KL}_s \left(\bar{\zeta}^{(j+1)} \,\|\, \bar{\zeta}^{(j)}\right) \right]$$

$$+ \sum_{l=0}^{t-2} \alpha_{l+1,t} \left[ \frac{12\eta}{1-\gamma} \left\| Q^{(l+1)}(s) - Q^{(l)}(s) \right\|_\infty + \frac{2}{1-\gamma} \mathsf{KL}_s \left(\bar{\zeta}^{(l+1)} \,\|\, \bar{\zeta}^{(l)}\right) \right]$$

$$\leq \sum_{l=0}^{t-1} \alpha_{l+1,t} \left[ \left(1 + \frac{24\eta}{1-\gamma}\right) \left\| Q^{(l+1)}(s) - Q^{(l)}(s) \right\|_\infty + \frac{4}{\eta} \mathsf{KL}_s \left(\bar{\zeta}^{(l+1)} \,\|\, \bar{\zeta}^{(l)}\right) \right].$$

Plugging the above inequality into (30) leads to

$$\left| Q^{(t+1)}(s, a, b) - Q^{(t)}(s, a, b) \right|$$

$$\leq \gamma \mathop{\mathbb{E}}_{s' \sim P(\cdot|s,a,b))} \left\{ \sum_{l=0}^{t-1} \alpha_{l+1,t} \left[ \left(1 + \frac{24\eta}{1-\gamma}\right) \left\| Q^{(l+1)}(s') - Q^{(l)}(s') \right\|_\infty + \frac{4}{\eta} \mathsf{KL}_{s'} \left(\bar{\zeta}^{(l+1)} \,\|\, \bar{\zeta}^{(l)}\right) \right] \right\}.$$

When $\eta \leq \frac{(1-\gamma)^2}{48\gamma}$, we have $\gamma(1 + \frac{24\eta}{1-\gamma}) \leq \frac{1+\gamma}{2}$ and hence that

$$\left| Q^{(t+1)}(s, a, b) - Q^{(t)}(s, a, b) \right|$$

$$\leq \mathop{\mathbb{E}}_{s' \sim P(\cdot|s,a,b))} \left\{ \frac{1+\gamma}{2} \sum_{l=0}^{t-1} \alpha_{l+1,t} \left[ \left\| Q^{(l+1)}(s') - Q^{(l)}(s') \right\|_\infty + \frac{4}{\eta} \mathsf{KL}_{s'} \left(\bar{\zeta}^{(l+1)} \,\|\, \bar{\zeta}^{(l)}\right) \right] \right\}.$$

Let $x^{(t+1)} \in \mathcal{A}^{\mathcal{S}}$ and $y^{(t+1)} \in \mathcal{B}^{\mathcal{S}}$ be defined as

$$(x^{(t+1)}(s), y^{(t+1)}(s)) = \arg \max_{(a,b) \in \mathcal{A} \times \mathcal{B}} \left| Q^{(t+1)}(s, a, b) - Q^{(t)}(s, a, b) \right|.$$

It follows that $\forall \chi \in \Gamma(\rho)$, we have $\chi P_{x^{(t+1)}, y^{(t+1)}} \in \Gamma(\rho)$ and hence

$$\mathop{\mathbb{E}}_{s \sim \chi} \left[ \left\| Q^{(t+1)}(s) - Q^{(t)}(s) \right\|_\infty \right]$$

$$= \mathop{\mathbb{E}}_{\substack{s \sim \chi, \\ a = x^{(t+1)}(s), \\ b = y^{(t+1)}(s)}} \left[ \left| Q^{(t+1)}(s, a, b) - Q^{(t)}(s, a, b) \right| \right]$$

$$\leq \mathop{\mathbb{E}}_{s' \sim \chi P_{x^{(t+1)}, y^{(t+1)}}} \left[ \frac{1+\gamma}{2} \sum_{l=0}^{t-1} \alpha_{l+1,t} \left[ \left\| Q^{(l+1)}(s') - Q^{(l)}(s') \right\|_\infty + \frac{4}{\eta} \mathsf{KL}_{s'} \left(\bar{\zeta}^{(l+1)} \,\|\, \bar{\zeta}^{(l)}\right) \right] \right]$$

$$\leq \frac{1+\gamma}{2} \sum_{l=0}^{t-1} \alpha_{l+1,t} \left[ \left\| Q^{(l+1)}(s') - Q^{(l)}(s') \right\|_{\Gamma(\rho)} + \frac{4}{\eta} \cdot \left\| \frac{\chi P_{x^{(t+1)}, y^{(t+1)}}}{\rho} \right\|_\infty \mathsf{KL}_\rho \left(\bar{\zeta}^{(l+1)} \,\|\, \bar{\zeta}^{(l)}\right) \right]$$

$$\leq \frac{1+\gamma}{2} \sum_{l=0}^{t-1} \alpha_{l+1,t} \left[ \left\| Q^{(l+1)}(s') - Q^{(l)}(s') \right\|_{\Gamma(\rho)} + \frac{4\mathcal{C}_\rho}{\eta} \mathsf{KL}_\rho \left(\bar{\zeta}^{(l+1)} \,\|\, \bar{\zeta}^{(l)}\right) \right]. \tag{32}$$

Taking supremum over $\chi \in \Gamma(\rho)$ completes the proof.

When $t = 0$, we have $\left\| Q^{(0)}(s) - Q^{(1)}(s) \right\|_{\Gamma(\rho)} = \left\| Q^{(1)}(s) \right\|_{\Gamma(\rho)} \leq 2$. $\qquad \square$

### C.3   PROOF OF LEMMA 3

Note that if suffices to show for $t \geq 0$, $s \in \mathcal{S}$, $(a, b) \in \mathcal{A} \times \mathcal{B}$:

$$
\left| Q^{(t+1)}(s, a, b) - Q^{\star}_{\tau}(s, a, b) \right|
$$
$$
\leq \frac{1 + \gamma}{2} \cdot \mathop{\mathbb{E}}_{s' \sim P(s, a, b)} \left[ \sum_{l=0}^{t} \alpha_{l,t} \left[ \left\| Q^{(l)}(s') - Q^{\star}_{\tau}(s') \right\|_{\infty} + \frac{2\eta}{1 - \gamma} \left\| Q^{(l)}(s') - Q^{(l-1)}(s') \right\|_{\infty} \right] \right] + 2\alpha_{0,t}.
\tag{33}
$$

The remaining step follows a similar argument as (32) and is therefore omitted.

For $t \geq 0$, we have

$$
Q^{(t+1)}(s, a, b) - Q^{\star}_{\tau}(s, a, b) = \gamma \mathbb{E}_{s' \sim P(\cdot|s, a, b)} \left[ V^{(t)}(s') - V^{\star}_{\tau}(s') \right]
$$
$$
= \gamma \mathbb{E}_{s' \sim P(\cdot|s, a, b)} \left[ \sum_{l=0}^{t} \alpha_{l,t}(f^{(l)}_{s'} - f^{\star}_{s'}) \right].
\tag{34}
$$

We start by decomposing $f^{(t)}_s - f^{\star}_s$ as

$$
\begin{aligned}
f^{(t)}_s - f^{\star}_s &= f_s(Q^{(t)}, \bar{\mu}^{(t)}, \bar{\nu}^{(t)}) - f_s(Q^{\star}_{\tau}, \mu^{\star}_{\tau}, \nu^{\star}_{\tau}) \\
&= \left( f_s(Q^{(t)}, \bar{\mu}^{(t)}, \bar{\nu}^{(t)}) - f_s(Q^{(t)}, \bar{\mu}^{(t)}, \nu^{\star}_{\tau}) \right) + f_s(Q^{(t)}, \bar{\mu}^{(t)}, \nu^{\star}_{\tau}) - f_s(Q^{\star}_{\tau}, \mu^{\star}_{\tau}, \nu^{\star}_{\tau}) \\
&\leq \left( f_s(Q^{(t)}, \bar{\mu}^{(t)}, \bar{\nu}^{(t)}) - f_s(Q^{(t)}, \bar{\mu}^{(t)}, \nu^{\star}_{\tau}) \right) + f_s(Q^{\star}_{\tau}, \bar{\mu}^{(t)}, \nu^{\star}_{\tau}) - f_s(Q^{\star}_{\tau}, \mu^{\star}_{\tau}, \nu^{\star}_{\tau}) \\
&\quad + \left\| Q^{(t)}(s) - Q^{\star}_{\tau}(s) \right\|_{\infty} \\
&\leq f_s(Q^{(t)}, \bar{\mu}^{(t)}, \bar{\nu}^{(t)}) - f_s(Q^{(t)}, \bar{\mu}^{(t)}, \nu^{\star}_{\tau}) + \left\| Q^{(t)}(s) - Q^{\star}_{\tau}(s) \right\|_{\infty}.
\end{aligned}
$$

We bound the first two terms with the following lemma:

**Lemma 13.** *It holds for all $t \geq 0$, $s \in \mathcal{S}$ and $\nu(s) \in \Delta(\mathcal{B})$ that*

$$
\begin{aligned}
&f_s(Q^{(t)}, \bar{\mu}^{(t)}, \bar{\nu}^{(t)}) - f_s(Q^{(t)}, \bar{\mu}^{(t)}, \nu) \\
&= \left\langle \bar{\nu}^{(t)}(s) - \nu(s), Q^{(t)}(s)^{\top} \bar{\mu}^{(t)}(s) \right\rangle - \tau \mathcal{H}(\bar{\nu}^{(t)}(s)) + \tau \mathcal{H}(\nu^{\star}_{\tau}(s)) \\
&\leq \frac{2\eta}{1 - \gamma} \left\| Q^{(t)}(s) - Q^{(t-1)}(s) \right\|_{\infty} + \frac{2}{1 - \gamma} \left( \mathsf{KL}_s\big(\bar{\mu}^{(t)} \,\|\, \mu^{(t-1)}\big) + \mathsf{KL}_s\big(\mu^{(t-1)} \,\|\, \bar{\mu}^{(t-1)}\big) \right) \\
&\quad - \frac{1}{\eta}\Big(1 - \frac{4\eta}{1 - \gamma}\Big) \mathsf{KL}_s\big(\nu^{(t)} \,\|\, \bar{\nu}^{(t)}\big) - \frac{1 - \eta\tau}{\eta} \mathsf{KL}_s\big(\bar{\nu}^{(t)} \,\|\, \nu^{(t-1)}\big) \\
&\quad + \frac{1 - \eta\tau}{\eta} \mathsf{KL}_s\big(\nu \,\|\, \nu^{(t-1)}\big) - \frac{1}{\eta} \mathsf{KL}_s\big(\nu \,\|\, \nu^{(t)}\big).
\end{aligned}
$$

*Proof.* See Appendix E.3. □

Applying Lemma 13 with $\nu(s) = \nu^{\star}_{\tau}(s)$ gives

$$
\begin{aligned}
f^{(t)}_s - f^{\star}_s &\leq \left\| Q^{(t)}(s) - Q^{\star}_{\tau}(s) \right\|_{\infty} + \frac{2\eta}{1 - \gamma} \left\| Q^{(t)}(s) - Q^{(t-1)}(s) \right\|_{\infty} \\
&\quad + \frac{1 - \eta\tau}{\eta} \mathsf{KL}_s\big(\nu^{\star}_{\tau} \,\|\, \nu^{(t-1)}\big) - \frac{1}{\eta} \mathsf{KL}_s\big(\nu^{\star}_{\tau} \,\|\, \nu^{(t)}\big) \\
&\quad - \frac{1}{\eta}\Big(1 - \frac{4\eta}{1 - \gamma}\Big) \mathsf{KL}_s\big(\nu^{(t)} \,\|\, \bar{\nu}^{(t)}\big) - \frac{1 - \eta\tau}{\eta} \mathsf{KL}_s\big(\bar{\nu}^{(t)} \,\|\, \nu^{(t-1)}\big) \\
&\quad + \frac{2}{1 - \gamma} \left( \mathsf{KL}_s\big(\bar{\mu}^{(t)} \,\|\, \mu^{(t-1)}\big) + \mathsf{KL}_s\big(\mu^{(t-1)} \,\|\, \bar{\mu}^{(t-1)}\big) \right)
\end{aligned}
\tag{35}
$$

By a similar argument, we can derive

$$
\begin{aligned}
f_s^\star - f_s^{(t)} \leq{}& \left\|Q^{(t)}(s) - Q_\tau^\star(s)\right\|_\infty + \frac{2\eta}{1-\gamma}\left\|Q^{(t)}(s) - Q^{(t-1)}(s)\right\|_\infty \\
&+ \frac{1-\eta\tau}{\eta}\mathsf{KL}_s\big(\mu_\tau^\star \,\|\, \mu^{(t-1)}\big) - \frac{1}{\eta}\mathsf{KL}_s\big(\mu_\tau^\star \,\|\, \mu^{(t)}\big) \\
&- \frac{1}{\eta}\big(1 - \frac{4\eta}{1-\gamma}\big)\mathsf{KL}_s\big(\mu^{(t)} \,\|\, \bar\mu^{(t)}\big) - \frac{1-\eta\tau}{\eta}\mathsf{KL}_s\big(\bar\mu^{(t)} \,\|\, \mu^{(t-1)}\big) \\
&+ \frac{2}{1-\gamma}\Big(\mathsf{KL}_s\big(\bar\nu^{(t)} \,\|\, \nu^{(t-1)}\big) + \mathsf{KL}_s\big(\nu^{(t-1)} \,\|\, \bar\nu^{(t-1)}\big)\Big).
\end{aligned}
\tag{36}
$$

Computing (35) $+\frac{1-\gamma}{4}\cdot$ (36) gives

$$
\begin{aligned}
(1 &- \frac{1-\gamma}{4})(f_s^{(t)} - f_s^\star) \\
\leq{}& (1 + \frac{1-\gamma}{4})\Big[\left\|Q^{(t)}(s) - Q_\tau^\star(s)\right\|_\infty + \frac{2\eta}{1-\gamma}\left\|Q^{(t)}(s) - Q^{(t-1)}(s)\right\|_\infty\Big] \\
&+ \frac{1-\eta\tau}{\eta}\Big[\mathsf{KL}_s\big(\nu_\tau^\star \,\|\, \nu^{(t-1)}\big) + \frac{1-\gamma}{4}\mathsf{KL}_s\big(\mu_\tau^\star \,\|\, \mu^{(t-1)}\big)\Big] - \frac{1}{\eta}\Big[\mathsf{KL}_s\big(\nu_\tau^\star \,\|\, \nu^{(t)}\big) + \frac{1-\gamma}{4}\mathsf{KL}_s\big(\mu_\tau^\star \,\|\, \mu^{(t)}\big)\Big] \\
&+ \frac{2}{1-\gamma}\Big[\mathsf{KL}_s\big(\mu^{(t-1)} \,\|\, \bar\mu^{(t-1)}\big) + \frac{1-\gamma}{4}\mathsf{KL}_s\big(\nu^{(t-1)} \,\|\, \bar\nu^{(t-1)}\big)\Big] \\
&- \frac{1}{\eta}\big(1 - \frac{4\eta}{1-\gamma}\big)\Big[\frac{1-\gamma}{4}\mathsf{KL}_s\big(\mu^{(t)} \,\|\, \bar\mu^{(t)}\big) + \mathsf{KL}_s\big(\nu^{(t)} \,\|\, \bar\nu^{(t)}\big)\Big] \\
&+ \big(\frac{2}{1-\gamma} - \frac{1-\eta\tau}{\eta}\cdot\frac{1-\gamma}{4}\big)\mathsf{KL}_s\big(\bar\mu^{(t)} \,\|\, \mu^{(t-1)}\big) + \big(\frac{2}{1-\gamma}\cdot\frac{1-\gamma}{4} - \frac{1-\eta\tau}{\eta}\big)\mathsf{KL}_s\big(\bar\nu^{(t)} \,\|\, \nu^{(t-1)}\big).
\end{aligned}
\tag{37}
$$

With $0 < \eta \leq (1-\gamma)^2/16$, we have $\big(\frac{2}{1-\gamma} - \frac{1-\eta\tau}{\eta}\cdot\frac{1-\gamma}{4}\big) \leq 0$, $\big(\frac{2}{1-\gamma}\cdot\frac{1-\gamma}{4} - \frac{1-\eta\tau}{\eta}\big) \leq 0$, and

$$
\frac{1}{\eta}\big(1 - \frac{4\eta}{1-\gamma}\big)\cdot\frac{1-\gamma}{4} \geq \frac{2}{1-\gamma}\cdot\frac{1}{1-\eta\tau}.
$$

To proceed, we introduce a shorthand notation

$$
\begin{aligned}
G^{(t)}(s) ={}& \frac{1}{\eta}\Big[\mathsf{KL}_s\big(\nu_\tau^\star \,\|\, \nu^{(t)}\big) + \frac{1-\gamma}{4}\mathsf{KL}_s\big(\mu_\tau^\star \,\|\, \mu^{(t)}\big)\Big] \\
&+ \frac{2}{(1-\gamma)(1-\eta\tau)}\Big[\mathsf{KL}_s\big(\mu^{(t)} \,\|\, \bar\mu^{(t)}\big) + \mathsf{KL}_s\big(\nu^{(t)} \,\|\, \bar\nu^{(t)}\big)\Big].
\end{aligned}
$$

We can then write (37) as

$$
\begin{aligned}
(1 - \frac{1-\gamma}{4})(f_s^{(t)} - f_s^\star) \leq{}& (1 + \frac{1-\gamma}{4})\Big[\left\|Q^{(t)}(s) - Q_\tau^\star(s)\right\|_\infty + \frac{2\eta}{1-\gamma}\left\|Q^{(t)}(s) - Q^{(t-1)}(s)\right\|_\infty\Big] \\
&+ (1-\eta\tau)G^{(t-1)}(s) - G^{(t)}(s).
\end{aligned}
\tag{38}
$$

Note that when $t = 0$, we have

$$
\begin{aligned}
f_s^{(0)} - f_s^\star &= \tau\log|\mathcal{A}| - \tau\log|\mathcal{B}| - \mu_\tau^\star(s)^\top Q_\tau^\star(s)\nu_\tau^\star(s) - \tau\mathcal{H}(\mu_\tau^\star(s)) + \tau\mathcal{H}(\nu_\tau^\star(s)) \\
&= \max_{\mu(s)}\min_{\nu(s)} f_s(Q^{(0)}, \mu, \nu) - \max_{\mu(s)}\min_{\nu(s)} f_s(Q_\tau^\star, \mu, \nu) \\
&\leq \left\|Q^{(0)}(s) - Q_\tau^\star(s)\right\|_\infty.
\end{aligned}
\tag{39}
$$

Substitution of (38) and (39) into (34) gives

$$
\begin{aligned}
Q^{(t+1)}&(s,a,b) - Q_\tau^\star(s,a,b) \\
&= \gamma\mathbb{E}_{s'\sim P(\cdot|s,a,b)}\left[\sum_{l=0}^{t}\alpha_{l,t}(f_{s'}^{(l)} - f_{s'}^\star)\right] \\
&\leq \gamma\mathbb{E}_{s'\sim P(s,a,b)}\left[\alpha_{0,t}\left\|Q^{(0)}(s') - Q_\tau^\star(s')\right\|_\infty\right]
\end{aligned}
$$

$$+ \gamma \cdot \frac{1 + (1-\gamma)/4}{1 - (1-\gamma)/4} \mathop{\mathbb{E}}_{s' \sim P(s,a,b)} \left[ \sum_{l=1}^{t} \alpha_{l,t} \Big[ \big\| Q^{(l)}(s') - Q_\tau^\star(s') \big\|_\infty + \frac{2\eta}{1-\gamma} \big\| Q^{(l)}(s') - Q^{(l-1)}(s') \big\|_\infty \Big] \right]$$

$$+ \frac{\gamma}{1 - (1-\gamma)/4} \mathop{\mathbb{E}}_{s' \sim P(s,a,b)} \left[ (1 - \eta\tau) \sum_{l=1}^{t} \alpha_{l,t} G^{(l-1)}(s') - \sum_{l=1}^{t} \alpha_{l,t} G^{(l)}(s') \right].$$

Note that

$$(1 - \eta\tau) \sum_{l=1}^{t} \alpha_{l,t} G^{(l-1)}(s') - \sum_{l=1}^{t} \alpha_{l,t} G^{(l)}(s')$$

$$\leq \sum_{l=1}^{t-1} ((1 - \eta\tau)\alpha_{l+1,t} - \alpha_{l,t}) G^{(l)}(s') + \alpha_{1,t} G^{(0)}(s')$$

$$\leq \alpha_{1,t} G^{(0)}(s') \leq 2\alpha_{0,t}\eta\tau G^{(0)}(s') \leq 2\alpha_{0,t},$$

where the second step is due to

$$(1 - \eta\tau)\alpha_{l+1,t} - \alpha_{l,t} = ((1 - \eta\tau)\alpha_{l+1} - \alpha_l(1 - \alpha_{l+1})) \prod_{j=l+2}^{t} \alpha_j$$

$$\leq ((1 - \eta\tau)\alpha_{l+1} - \alpha_{l+1} + \alpha_l\alpha_{l+1}) \prod_{j=l+2}^{t} \alpha_j$$

$$= \alpha_{l+1}(\alpha_l - \eta\tau) \prod_{j=l+2}^{t} \alpha_j \leq 0. \tag{40}$$

So we conclude that

$$Q^{(t+1)}(s,a,b) - Q_\tau^\star(s,a,b)$$

$$\leq \gamma \cdot \frac{1 + (1-\gamma)/4}{1 - (1-\gamma)/4} \mathop{\mathbb{E}}_{s' \sim P(s,a,b)} \left[ \sum_{l=0}^{t} \alpha_{l,t} \Big[ \big\| Q^{(l)}(s') - Q_\tau^\star(s') \big\|_\infty + \frac{2\eta}{1-\gamma} \big\| Q^{(l)}(s') - Q^{(l-1)}(s') \big\|_\infty \Big] \right]$$

$$+ 2\alpha_{0,t}$$

$$\leq \frac{1+\gamma}{2} \cdot \mathop{\mathbb{E}}_{s' \sim P(s,a,b)} \left[ \sum_{l=0}^{t} \alpha_{l,t} \Big[ \big\| Q^{(l)}(s') - Q_\tau^\star(s') \big\|_\infty + \frac{2\eta}{1-\gamma} \big\| Q^{(l)}(s') - Q^{(l-1)}(s') \big\|_\infty \Big] \right]$$

$$+ 2\alpha_{0,t}.$$

The other side of (33) can be obtained by computing $\frac{1-\gamma}{4} \cdot$ (35) + (36) and following a similar argument, and is therefore omitted.

For $t = 0$, we have $\big| Q^{(1)}(s,a,b) - Q_\tau^\star(s,a,b) \big| \leq \gamma \max_{s' \in \mathcal{S}} |f_{s'}^{(0)} - f_{s'}^\star| \leq \frac{2\gamma}{1-\gamma}$.

## C.4   PROOF OF LEMMA 4

For $t \geq 1$, let

$$u_t = \eta \big\| Q_\tau^\star(s) - Q^{(t)}(s) \big\|_{\Gamma(\rho)} + \frac{12\eta^2}{(1-\gamma)^2} \big\| Q^{(t)}(s) - Q^{(t-1)}(s) \big\|_{\Gamma(\rho)}.$$

It follows that

$$u_1 \leq \frac{2\gamma\eta}{1-\gamma} + \frac{24\eta^2}{(1-\gamma)^3} \leq 1.$$

When $t \geq 1$, invoking Lemma 2 and Lemma 3 gives

$$
u_{t+1} \leq \left(1 - \frac{1-\gamma}{2}\right) \sum_{l=1}^{t} \alpha_{l,t} \left[\eta \|Q^{(l)} - Q_\tau^\star\|_{\Gamma(\rho)} + \left(\frac{2\eta^2}{1-\gamma} + \frac{12\eta^2}{(1-\gamma)^2}\right) \|Q^{(l)} - Q^{(l-1)}\|_{\Gamma(\rho)}\right]
$$

$$
+ \frac{48\eta\mathcal{C}_\rho}{(1-\gamma)^2} \sum_{l=1}^{t} \alpha_{l,t} \mathsf{KL}_\rho\big(\bar\zeta^{(l)} \,\|\, \bar\zeta^{(l-1)}\big) + 2\alpha_{0,t}\eta + \alpha_{0,t}\eta \|Q^{(0)} - Q_\tau^\star\|_{\Gamma(\rho)}
$$

$$
\leq \left(1 - \frac{1-\gamma}{3}\right) \sum_{l=1}^{t} \alpha_{l,t} u_l + \frac{48\eta\mathcal{C}_\rho}{(1-\gamma)^2} \sum_{l=1}^{t} \alpha_{l,t} \mathsf{KL}_\rho\big(\bar\zeta^{(l)} \,\|\, \bar\zeta^{(l-1)}\big) + \frac{4\eta}{1-\gamma}\alpha_{0,t}.
$$

$$(41)$$

Let

$$
\beta_{l,t} = \alpha_l \prod_{i=l+1}^{t} \left(1 - \frac{1-\gamma}{3} \cdot \alpha_i\right).
$$

It follows that for $t \geq 0$,

$$
\sum_{l=1}^{t+1} \alpha_{l,t+1} u_l
$$

$$
= (1 - \alpha_{t+1}) \sum_{l=1}^{t} \alpha_{l,t} u_l + \alpha_{t+1} u_{t+1}
$$

$$
\leq \left(1 - \frac{1-\gamma}{3} \cdot \alpha_{t+1}\right) \sum_{l=1}^{t} \alpha_{l,t} u_l + \alpha_{t+1} \frac{48\eta\mathcal{C}_\rho}{(1-\gamma)^2} \cdot \sum_{l=1}^{t} \alpha_{l,t} \mathsf{KL}_\rho\big(\bar\zeta^{(l)} \,\|\, \bar\zeta^{(l-1)}\big) + \frac{4\eta}{1-\gamma}\alpha_{t+1}\alpha_{0,t}
$$

$$
\leq \prod_{l=2}^{t+1} \left(1 - \frac{1-\gamma}{3} \cdot \alpha_l\right)\alpha_{1,1} u_1 + \frac{48\eta\mathcal{C}_\rho}{(1-\gamma)^2} \sum_{i=1}^{t} \beta_{i+1,t+1} \sum_{l=1}^{i} \alpha_{l,i} \mathsf{KL}_\rho\big(\bar\zeta^{(l)} \,\|\, \bar\zeta^{(l-1)}\big) + \frac{4\eta}{1-\gamma} \sum_{i=1}^{t} \alpha_{0,i}\beta_{i+1,t+1}
$$

$$
\leq \beta_{1,t+1} u_1 + \frac{48\eta\mathcal{C}_\rho}{(1-\gamma)^2} \sum_{l=1}^{t} \sum_{i=l}^{t} \alpha_{l,i}\beta_{i+1,t+1} \mathsf{KL}_\rho\big(\bar\zeta^{(l)} \,\|\, \bar\zeta^{(l-1)}\big) + \frac{4\eta}{1-\gamma} \sum_{i=1}^{t} \alpha_{0,i}\beta_{i+1,t+1}
$$

$$
\leq \frac{200\eta\mathcal{C}_\rho}{(1-\gamma)^2} \sum_{l=1}^{t} \beta_{l,t+1} \mathsf{KL}_\rho\big(\bar\zeta^{(l)} \,\|\, \bar\zeta^{(l-1)}\big) + \frac{18\eta}{1-\gamma}\beta_{0,t+1},
$$

$$(42)$$

where the last step is due to the following lemma. Similar lemma has appeared in prior works (see i.e., (Wei et al., 2021b, Lemma 36)). Our version features a simpler proof, which is postponed to Appendix E.4.

**Lemma 14.** *Let two sequences $\{\delta_i\}, \{\xi_i\}$ be defined as*

$$
\delta_i = 1 - c_1\alpha_i, \qquad and \qquad \xi_i = 1 - c_2\alpha_i,
$$

*where the constants $c_1, c_2$ satisfiy $0 < c_1 < c_2 < \frac{1}{2\alpha_i}$. For $l \leq t$, let $\delta_{l,t} = \alpha_l \prod_{i=l+1}^{d} \delta_i$ and $\xi_{l,t} = \alpha_l \prod_{i=l+1}^{d} \xi_i$. We have*

$$
\sum_{i=l}^{t} \xi_{l,i}\delta_{i+1,t} \leq \left(1 + \frac{2}{c_2 - c_1}\right)\delta_{l,t}.
$$

Substitution of (42) into (41) gives

$$
u_{t+1} \leq \left(1 - \frac{1-\gamma}{3}\right) \sum_{l=1}^{t} \alpha_{l,t} u_l + \frac{48\eta}{(1-\gamma)^2} \sum_{l=1}^{t} \alpha_{l,t} \mathsf{KL}\big(\bar\zeta^{(l)} \,\|\, \bar\zeta^{(l-1)}\big) + \frac{4\eta}{1-\gamma}\alpha_{0,t}
$$

$$
\leq \frac{200\eta\mathcal{C}_\rho}{(1-\gamma)^2} \sum_{l=1}^{t} \beta_{l,t} \mathsf{KL}\big(\bar\zeta^{(l)} \,\|\, \bar\zeta^{(l-1)}\big) + \frac{18\eta}{1-\gamma}\beta_{0,t} + \frac{48\eta\mathcal{C}_\rho}{(1-\gamma)^2} \sum_{l=1}^{t} \alpha_{l,t} \mathsf{KL}\big(\bar\zeta^{(l)} \,\|\, \bar\zeta^{(l-1)}\big) + \frac{4\eta}{1-\gamma}\alpha_{0,t}
$$

$$
\leq \frac{250\eta\mathcal{C}_\rho}{(1-\gamma)^2} \sum_{l=1}^{t} \beta_{l,t} \mathsf{KL}\big(\bar\zeta^{(l)} \,\|\, \bar\zeta^{(l-1)}\big) + \frac{22\eta}{1-\gamma}\beta_{0,t}.
$$

for $t \geq 1$. It is straightforward to verify that the above inequality holds for $t = 0$ as well. So we conclude that

$$
\begin{aligned}
\sum_{l=0}^{t} \lambda_{l+1,t+1} u_{l+1} &= \sum_{i=0}^{t} \lambda_{i+1,t+1} u_{i+1} \\
&\leq \sum_{i=0}^{t} \lambda_{i+1,t+1} \Big[ \frac{250\eta \mathcal{C}_\rho}{(1-\gamma)^2} \sum_{l=1}^{i} \beta_{l,i} \mathsf{KL}\big(\bar{\zeta}^{(l)} \,\|\, \bar{\zeta}^{(l-1)}\big) + \frac{22\eta}{1-\gamma} \beta_{0,i} \Big] \\
&= \frac{250\eta \mathcal{C}_\rho}{(1-\gamma)^2} \sum_{l=1}^{t} \sum_{i=l}^{t} \beta_{l,i} \lambda_{i+1,t+1} \mathsf{KL}\big(\bar{\zeta}^{(l)} \,\|\, \bar{\zeta}^{(l-1)}\big) + \frac{22\eta}{1-\gamma} \sum_{i=0}^{t} \beta_{0,i} \lambda_{i+1,t+1} \\
&\leq \frac{6250\eta \mathcal{C}_\rho}{(1-\gamma)^3} \sum_{l=1}^{t} \lambda_{l,t+1} \mathsf{KL}\big(\bar{\zeta}^{(l)} \,\|\, \bar{\zeta}^{(l-1)}\big) + \frac{550\eta}{(1-\gamma)^2} \lambda_{0,t+1} \\
&= \frac{6250\eta \mathcal{C}_\rho}{(1-\gamma)^3} \sum_{l=0}^{t-1} \lambda_{l+1,t+1} \mathsf{KL}\big(\bar{\zeta}^{(l+1)} \,\|\, \bar{\zeta}^{(l)}\big) + \frac{550\eta}{(1-\gamma)^2} \lambda_{0,t+1},
\end{aligned}
$$

where the penultimate step invokes Lemma 14.

## C.5 PROOF OF LEMMA 5

Taking logarithm on the both sides of the update rule (9b), we get

$$
\begin{cases}
\log \bar{\mu}^{(t+1)}(s) - (1 - \eta\tau) \log \mu^{(t)}(s) & \overset{1}{=} \eta Q^{(t)}(s) \bar{\nu}^{(t)}(s) \\
\log \bar{\nu}^{(t+1)}(s) - (1 - \eta\tau) \log \nu^{(t)}(s) & \overset{1}{=} -\eta Q^{(t)}(s)^\top \bar{\mu}^{(t)}(s)
\end{cases} . \tag{43}
$$

Subtracting (27) from (43) and taking inner product with $\bar{\zeta}^{(t+1)}(s) - \zeta_\tau^\star(s)$ gives

$$
\begin{aligned}
&\big\langle \log \bar{\zeta}^{(t+1)}(s) - (1 - \eta\tau) \log \zeta^{(t)}(s) - \eta\tau \log \zeta_\tau^\star(s), \bar{\zeta}^{(t+1)}(s) - \zeta_\tau^\star(s) \big\rangle \\
&= \eta \big\langle \bar{\mu}^{(t+1)}(s) - \mu_\tau^\star(s), Q^{(t)}(s) \bar{\nu}^{(t)}(s) - Q_\tau^\star(s) \nu_\tau^\star(s) \big\rangle \\
&\quad - \eta \big\langle \bar{\nu}^{(t+1)}(s) - \nu_\tau^\star(s), Q^{(t)}(s)^\top \bar{\mu}^{(t)}(s) - Q_\tau^\star(s)^\top \mu_\tau^\star(s) \big\rangle \\
&\leq \eta \big\langle \bar{\mu}^{(t+1)}(s) - \mu_\tau^\star(s), Q^{(t)}(s) \big( \bar{\nu}^{(t)}(s) - \nu_\tau^\star(s) \big) \big\rangle \\
&\quad - \eta \big\langle \bar{\nu}^{(t+1)}(s) - \nu_\tau^\star(s), Q^{(t)}(s)^\top \big( \bar{\mu}^{(t)}(s) - \mu_\tau^\star(s) \big) \big\rangle + 2\eta \big\| Q^{(t)}(s) - Q_\tau^\star(s) \big\|_\infty \\
&\leq \eta \big\langle \bar{\mu}^{(t+1)}(s) - \mu_\tau^\star(s), Q^{(t)}(s) \big( \bar{\nu}^{(t)}(s) - \bar{\nu}^{(t+1)}(s) \big) \big\rangle \\
&\quad - \eta \big\langle \bar{\nu}^{(t+1)}(s) - \nu_\tau^\star(s), Q^{(t)}(s)^\top \big( \bar{\mu}^{(t)}(s) - \bar{\mu}^{(t+1)}(s) \big) \big\rangle + 2\eta \big\| Q^{(t)}(s) - Q_\tau^\star(s) \big\|_\infty \\
&\leq \frac{2\eta}{1-\gamma} \Big( 2\mathsf{KL}_s\big(\zeta_\tau^\star \,\|\, \bar{\zeta}^{(t+1)}\big) + \mathsf{KL}_s\big(\bar{\zeta}^{(t+1)} \,\|\, \zeta^{(t)}\big) + \mathsf{KL}_s\big(\zeta^{(t)} \,\|\, \bar{\zeta}^{(t)}\big) \Big) + 2\eta \big\| Q^{(t)}(s) - Q_\tau^\star(s) \big\|_\infty .
\end{aligned}
$$

LHS can be written as

$$
\begin{aligned}
&\big\langle \log \bar{\zeta}^{(t+1)}(s) - (1 - \eta\tau) \log \zeta^{(t)}(s) - \eta\tau \log \zeta_\tau^\star(s), \bar{\zeta}^{(t+1)}(s) - \zeta_\tau^\star(s) \big\rangle \\
&= -\big\langle \log \bar{\zeta}^{(t+1)}(s) - (1 - \eta\tau) \log \zeta^{(t)}(s) - \eta\tau \log \zeta_\tau^\star(s), \zeta_\tau^\star(s) \big\rangle \\
&\quad + \big\langle \log \bar{\zeta}^{(t+1)}(s) - (1 - \eta\tau) \log \zeta^{(t)}(s) - \eta\tau \log \zeta_\tau^\star(s), \bar{\zeta}^{(t+1)}(s) \big\rangle \\
&= \mathsf{KL}_s\big(\zeta_\tau^\star \,\|\, \bar{\zeta}^{(t+1)}\big) - (1 - \eta\tau) \mathsf{KL}_s\big(\zeta_\tau^\star \,\|\, \zeta^{(t)}\big) \\
&\quad + (1 - \eta\tau) \mathsf{KL}_s\big(\bar{\zeta}^{(t+1)} \,\|\, \zeta^{(t)}\big) + \eta\tau \mathsf{KL}_s\big(\bar{\zeta}^{(t+1)} \,\|\, \zeta_\tau^\star\big).
\end{aligned}
$$

So we conclude that

$$
\Big( 1 - \frac{4\eta}{1-\gamma} \Big) \mathsf{KL}_s\big(\zeta_\tau^\star \,\|\, \bar{\zeta}^{(t+1)}\big) - (1 - \eta\tau) \mathsf{KL}_s\big(\zeta_\tau^\star \,\|\, \zeta^{(t)}\big)
$$

$$+ \left(1 - \eta\tau - \frac{2\eta}{1-\gamma}\right)\mathsf{KL}_s\big(\bar{\zeta}^{(t+1)} \,\|\, \zeta^{(t)}\big) + \eta\tau\mathsf{KL}_s\big(\bar{\zeta}^{(t+1)} \,\|\, \zeta_\tau^\star\big)$$

$$\leq \frac{2\eta}{1-\gamma}\mathsf{KL}_s\big(\zeta^{(t)} \,\|\, \bar{\zeta}^{(t)}\big) + 2\eta\big\|Q^{(t)}(s) - Q_\tau^\star(s)\big\|_\infty.$$

With $0 < \eta \leq \frac{1-\gamma}{8}$, we have

$$\frac{1}{2}\mathsf{KL}_s\big(\zeta_\tau^\star \,\|\, \bar{\zeta}^{(t+1)}\big) + \eta\tau\mathsf{KL}_s\big(\bar{\zeta}^{(t+1)} \,\|\, \zeta_\tau^\star\big)$$

$$\leq (1 - \eta\tau)\mathsf{KL}_s\big(\zeta_\tau^\star \,\|\, \zeta^{(t)}\big) + \frac{2\eta}{1-\gamma}\mathsf{KL}_s\big(\zeta^{(t)} \,\|\, \bar{\zeta}^{(t)}\big) + 2\eta\big\|Q^{(t)}(s) - Q_\tau^\star(s)\big\|_\infty.$$

## C.6  Proof of Lemma 6

We have

$$
\begin{aligned}
V_\tau^{\mu,\nu}(s) - V_\tau^\star(s) &= \mu(s)^\top Q_\tau^{\mu,\nu}(s)\nu(s) + \tau\mathcal{H}\big(\mu(s)\big) - \tau\mathcal{H}\big(\nu(s)\big) \\
&\quad - \mu_\tau^\star(s)^\top Q_\tau^\star(s)\nu_\tau^\star(s) - \tau\mathcal{H}\big(\mu_\tau^\star(s)\big) + \tau\mathcal{H}\big(\nu_\tau^\star(s)\big) \\
&= \mu(s)^\top Q_\tau^{\mu,\nu}(s)\nu(s) - \mu(s)^\top Q_\tau^\star(s)\nu(s) + f_s(Q_\tau^\star, \mu, \nu) - f_s(Q_\tau^\star, \mu_\tau^\star, \nu_\tau^\star) \\
&= \gamma \mathop{\mathbb{E}}_{\substack{a\sim\mu(\cdot|s), \\ b\sim\nu(\cdot|s), \\ s'\sim P(\cdot|s,a,b)}} [V_\tau^{\mu,\nu}(s') - V_\tau^\star(s')] + f_s(Q_\tau^\star, \mu, \nu) - f_s(Q_\tau^\star, \mu_\tau^\star, \nu_\tau^\star).
\end{aligned}
$$

Applying the inequality recursively and averaging $s$ over $\rho$, we arrive at

$$V_\tau^{\mu,\nu}(\rho) - V_\tau^\star(\rho) = \frac{1}{1-\gamma} \mathop{\mathbb{E}}_{s'\sim d_\rho^{\mu,\nu}} \left[f_{s'}(Q_\tau^\star, \mu, \nu) - f_{s'}(Q_\tau^\star, \mu_\tau^\star, \nu_\tau^\star)\right], \tag{44}$$

which is the well-known performance difference lemma applied to the setting of Markov games. It follows that

$$
\begin{aligned}
V_\tau^{\mu_\tau^\dagger(\nu),\nu}(\rho) - V_\tau^\star(\rho) &= \frac{1}{1-\gamma} \mathop{\mathbb{E}}_{s'\sim d_\rho^{\mu_\tau^\dagger(\nu),\nu}} \left[f_{s'}(Q_\tau^\star, \mu_\tau^\dagger(\nu), \nu) - f_{s'}(Q_\tau^\star, \mu_\tau^\star, \nu_\tau^\star)\right] \\
&\leq \frac{1}{1-\gamma} \mathop{\mathbb{E}}_{s'\sim d_\rho^{\mu_\tau^\dagger(\nu),\nu}} \left[f_{s'}(Q_\tau^\star, \mu_\tau^\dagger(\nu), \nu) - f_{s'}(Q_\tau^\star, \mu, \nu_\tau^\star)\right] \\
&\leq \frac{1}{1-\gamma} \mathop{\mathbb{E}}_{s'\sim d_\rho^{\mu_\tau^\dagger(\nu),\nu}} \left[\max_{\mu',\nu'} \Big(f_{s'}(Q_\tau^\star, \mu', \nu) - f_{s'}(Q_\tau^\star, \mu, \nu')\Big)\right] \\
&\leq \frac{\mathcal{C}_{\rho,\tau}^\dagger}{1-\gamma} \mathop{\mathbb{E}}_{s\sim\rho} \left[\max_{\mu',\nu'} \Big(f_s(Q_\tau^\star, \mu', \nu) - f_s(Q_\tau^\star, \mu, \nu')\Big)\right].
\end{aligned} \tag{45}
$$

A similar argument gives $V_\tau^\star(\rho) - V_\tau^{\mu,\nu_\tau^\dagger(\mu)}(\rho) \leq \frac{\mathcal{C}_{\rho,\tau}^\dagger}{1-\gamma} \mathop{\mathbb{E}}_{s\sim\rho} \left[\max_{\mu',\nu'} \Big(f_s(Q_\tau^\star, \mu', \nu) - f_s(Q_\tau^\star, \mu, \nu')\Big)\right]$.

Summing the two inequalities proves (18). Alternatively, we continue from (45) and show that

$$
\begin{aligned}
V_\tau^{\mu_\tau^\dagger(\nu),\nu}(s) - V_\tau^\star(s) &\leq \frac{1}{1-\gamma} \mathop{\mathbb{E}}_{s'\sim d_s^{\mu_\tau^\dagger(\nu),\nu}} \left[\max_{\mu',\nu'} \Big(f_{s'}(Q_\tau^\star, \mu', \nu) - f_{s'}(Q_\tau^\star, \mu, \nu')\Big)\right] \\
&\leq \frac{\|1/\rho\|_\infty}{1-\gamma} \mathop{\mathbb{E}}_{s\sim\rho} \left[\max_{\mu',\nu'} \Big(f_s(Q_\tau^\star, \mu', \nu) - f_s(Q_\tau^\star, \mu, \nu')\Big)\right].
\end{aligned}
$$

Summing the inequality with the one for $V_\tau^\star(s) - V_\tau^{\mu,\nu_\tau^\dagger(\mu)}(s)$ and taking maximum over $s \in \mathcal{S}$ completes the proof for (19).

## D  Proof of key lemmas for the episodic setting

### D.1  Proof of Lemma 9

Following the similar argument of arriving (28), we have

$$\big\langle \log\zeta_h^{(t+1)}(s) - (1 - \eta\tau)\log\zeta_h^{(t)}(s) - \eta\tau\log\zeta_{h,\tau}^\star(s), \bar{\zeta}_h^{(t+1)}(s) - \zeta_{h,\tau}^\star(s) \big\rangle$$

$$\leq 2\eta \big\| Q_h^{(t+1)}(s) - Q_{h,\tau}^{\star}(s) \big\|_{\infty}.$$

We rewrite LHS as

$$\big\langle \log \zeta_h^{(t+1)}(s) - (1-\eta\tau)\log \zeta_h^{(t)}(s) - \eta\tau \log \zeta_{h,\tau}^{\star}(s), \bar{\zeta}_h^{(t+1)}(s) - \zeta_{h,\tau}^{\star}(s) \big\rangle$$

$$= -\big\langle \log \zeta_h^{(t+1)}(s) - (1-\eta\tau)\log \zeta_h^{(t)}(s) - \eta\tau \log \zeta_{h,\tau}^{\star}(s), \zeta_{h,\tau}^{\star}(s) \big\rangle$$

$$\qquad + \big\langle \log \bar{\zeta}_h^{(t+1)}(s) - (1-\eta\tau)\log \zeta_h^{(t)}(s) - \eta\tau \log \zeta_{h,\tau}^{\star}(s), \bar{\zeta}_h^{(t+1)}(s) \big\rangle$$

$$\qquad + \big\langle \log \zeta_h^{(t+1)}(s) - \log \bar{\zeta}_h^{(t+1)}(s), \bar{\zeta}_h^{(t+1)}(s) \big\rangle$$

$$= \mathsf{KL}_s\big(\zeta_{h,\tau}^{\star} \,\|\, \zeta_h^{(t+1)}\big) - (1-\eta\tau)\mathsf{KL}_s\big(\zeta_{h,\tau}^{\star} \,\|\, \zeta_h^{(t)}\big)$$

$$\qquad + (1-\eta\tau)\mathsf{KL}_s\big(\bar{\zeta}_h^{(t+1)} \,\|\, \zeta_h^{(t)}\big) + \eta\tau\mathsf{KL}_s\big(\bar{\zeta}_h^{(t+1)} \,\|\, \zeta_{h,\tau}^{\star}\big)$$

$$\qquad + \mathsf{KL}_s\big(\zeta_h^{(t+1)} \,\|\, \bar{\zeta}_h^{(t+1)}\big) - \big\langle \log \bar{\zeta}_h^{(t+1)}(s) - \log \zeta_h^{(t+1)}(s), \bar{\zeta}_h^{(t+1)}(s) - \zeta_h^{(t+1)}(s) \big\rangle.$$

Rearranging terms gives

$$\mathsf{KL}_s\big(\zeta_{h,\tau}^{\star} \,\|\, \zeta_h^{(t+1)}\big) - (1-\eta\tau)\mathsf{KL}_s\big(\zeta_{h,\tau}^{\star} \,\|\, \zeta_h^{(t)}\big) + (1-\eta\tau)\mathsf{KL}_s\big(\bar{\zeta}_h^{(t+1)} \,\|\, \zeta_h^{(t)}\big)$$

$$\qquad + \eta\tau\mathsf{KL}_s\big(\bar{\zeta}_h^{(t+1)} \,\|\, \zeta_{h,\tau}^{\star}\big) + \mathsf{KL}_s\big(\zeta_h^{(t+1)} \,\|\, \bar{\zeta}_h^{(t+1)}\big) \tag{46}$$

$$\qquad - \big\langle \log \bar{\zeta}_h^{(t+1)}(s) - \log \zeta_h^{(t+1)}(s), \bar{\zeta}_h^{(t+1)}(s) - \zeta_h^{(t+1)}(s) \big\rangle$$

$$\leq 2\eta \big\| Q^{(t+1)}(s) - Q_{\tau}^{\star}(s) \big\|_{\infty}.$$

Note that

$$\big\langle \log \bar{\mu}_h^{(t+1)}(s) - \log \mu_h^{(t+1)}(s), \bar{\mu}_h^{(t+1)}(s) - \mu_h^{(t+1)}(s) \big\rangle$$

$$= \eta\big\langle Q_h^{(t)}(s)\bar{\nu}_h^{(t)}(s) - Q_h^{(t+1)}(s)\bar{\nu}_h^{(t+1)}(s), \bar{\mu}_h^{(t+1)}(s) - \mu_h^{(t+1)}(s) \big\rangle \tag{47}$$

$$\leq \eta\big\| Q_h^{(t)}(s)\bar{\nu}_h^{(t)}(s) - Q_h^{(t+1)}(s)\bar{\nu}_h^{(t+1)}(s) \big\|_1 \big\| \bar{\mu}_h^{(t+1)}(s) - \mu_h^{(t+1)}(s) \big\|_1.$$

We bound $\big\| Q_h^{(t)}(s)\bar{\nu}_h^{(t)}(s) - Q_h^{(t+1)}(s)\bar{\nu}_h^{(t+1)}(s) \big\|_1$ as

$$\big\| Q_h^{(t)}(s)\bar{\nu}_h^{(t)}(s) - Q_h^{(t+1)}(s)\bar{\nu}_h^{(t+1)}(s) \big\|_1$$

$$\leq \big\| Q_h^{(t+1)}(s)\big(\bar{\nu}_h^{(t)}(s) - \bar{\nu}_h^{(t+1)}(s)\big) \big\|_1 + \big\| \big(Q_h^{(t)}(s) - Q_h^{(t+1)}(s)\big)\bar{\nu}_h^{(t)}(s) \big\|_1$$

$$\leq 2H\big\| \bar{\nu}_h^{(t)}(s) - \bar{\nu}_h^{(t+1)}(s) \big\|_1 + \big\| Q_h^{(t)}(s) - Q_h^{(t+1)}(s) \big\|_{\infty}$$

$$\leq 2H\big\| \bar{\nu}_h^{(t+1)}(s) - \nu_h^{(t)}(s) \big\|_1 + 2H\big\| \nu_h^{(t)}(s) - \bar{\nu}_h^{(t)}(s) \big\|_1 + \big\| Q_h^{(t)}(s) - Q_h^{(t+1)}(s) \big\|_{\infty}.$$

Plugging the above inequality into (47) and invoking Young's inequality yields

$$\big\langle \log \bar{\mu}_h^{(t+1)}(s) - \log \mu_h^{(t+1)}(s), \bar{\mu}_h^{(t+1)}(s) - \mu_h^{(t+1)}(s) \big\rangle$$

$$\leq \eta H\Big( \big\| \bar{\nu}_h^{(t+1)}(s) - \nu_h^{(t)}(s) \big\|_1^2 + \big\| \nu_h^{(t)}(s) - \bar{\nu}_h^{(t)}(s) \big\|_1^2 + 2\big\| \bar{\mu}_h^{(t+1)}(s) - \mu_h^{(t+1)}(s) \big\|_1^2 \Big)$$

$$\qquad + \eta\big\| Q_h^{(t)}(s) - Q_h^{(t+1)}(s) \big\|_{\infty} \big\| \bar{\mu}_h^{(t+1)}(s) - \mu_h^{(t+1)}(s) \big\|_1$$

$$\leq 2\eta H\mathsf{KL}_s\big(\bar{\nu}_h^{(t+1)} \,\|\, \nu_h^{(t)}\big) + 2\eta H\mathsf{KL}_s\big(\nu_h^{(t)} \,\|\, \bar{\nu}_h^{(t)}\big) + 4\eta H\mathsf{KL}_s\big(\mu_h^{(t+1)} \,\|\, \bar{\mu}_h^{(t+1)}\big)$$

$$\qquad + 2\eta^2 H\big\| Q_h^{(t)}(s) - Q_h^{(t+1)}(s) \big\|_{\infty},$$

where the last step results from Pinsker's inequality and Lemma 8. Similarly, we have

$$\big\langle \log \bar{\nu}_h^{(t+1)}(s) - \log \nu_h^{(t+1)}(s), \bar{\nu}_h^{(t+1)}(s) - \nu_h^{(t+1)}(s) \big\rangle$$

$$\leq 2\eta H\mathsf{KL}_s\big(\bar{\mu}_h^{(t+1)} \,\|\, \mu_h^{(t)}\big) + 2\eta H\mathsf{KL}_s\big(\mu_h^{(t)} \,\|\, \bar{\mu}_h^{(t)}\big) + 4\eta H\mathsf{KL}_s\big(\nu_h^{(t+1)} \,\|\, \bar{\nu}_h^{(t+1)}\big)$$

$$\qquad + 2\eta^2 H\big\| Q_h^{(t)}(s) - Q_h^{(t+1)}(s) \big\|_{\infty}.$$

Summing the above two inequalities gives

$$\big\langle \log \bar{\zeta}_h^{(t+1)}(s) - \log \zeta_h^{(t+1)}(s), \bar{\zeta}_h^{(t+1)}(s) - \zeta_h^{(t+1)}(s) \big\rangle$$

$$\leq 2\eta H \mathsf{KL}_s\big(\bar{\zeta}_h^{(t+1)} \,\|\, \zeta_h^{(t)}\big) + 2\eta H \mathsf{KL}_s\big(\zeta_h^{(t)} \,\|\, \bar{\zeta}_h^{(t)}\big) + 4\eta H \mathsf{KL}_s\big(\zeta_h^{(t+1)} \,\|\, \bar{\zeta}_h^{(t+1)}\big)$$
$$+ 4\eta^2 H \big\|Q_h^{(t)}(s) - Q_h^{(t+1)}(s)\big\|_\infty$$
$$\leq 2\eta H \mathsf{KL}_s\big(\bar{\zeta}_h^{(t+1)} \,\|\, \zeta_h^{(t)}\big) + 2\eta H \mathsf{KL}_s\big(\zeta_h^{(t)} \,\|\, \bar{\zeta}_h^{(t)}\big) + 4\eta H \mathsf{KL}_s\big(\zeta_h^{(t+1)} \,\|\, \bar{\zeta}_h^{(t+1)}\big)$$
$$+ \frac{\eta}{2}\Big(\big\|Q_h^{(t)}(s) - Q_{h,\tau}^\star(s)\big\|_\infty + \big\|Q_h^{(t+1)}(s) - Q_{h,\tau}^\star(s)\big\|_\infty\Big),$$

where the second step invokes triangular inequality and the fact that $\eta \leq \frac{1}{8H}$. Plugging the above inequality into (46) gives

$$\mathsf{KL}_s\big(\zeta_{h,\tau}^\star \,\|\, \zeta_h^{(t+1)}\big) - (1 - \eta\tau)\mathsf{KL}_s\big(\zeta_{h,\tau}^\star \,\|\, \zeta_h^{(t)}\big) + (1 - \eta(\tau + 2H))\mathsf{KL}_s\big(\bar{\zeta}_h^{(t+1)} \,\|\, \zeta_h^{(t)}\big)$$
$$+ \eta\tau\mathsf{KL}_s\big(\bar{\zeta}_h^{(t+1)} \,\|\, \zeta_{h,\tau}^\star\big) + (1 - 4\eta H)\mathsf{KL}_s\big(\zeta_h^{(t+1)} \,\|\, \bar{\zeta}_h^{(t+1)}\big) - 2\eta H \mathsf{KL}_s\big(\zeta_h^{(t)} \,\|\, \bar{\zeta}_h^{(t)}\big)$$
$$\leq \frac{5\eta}{2}\big\|Q_h^{(t+1)}(s) - Q_\tau^\star(s)\big\|_\infty + \frac{\eta}{2}\big\|Q_h^{(t)}(s) - Q_\tau^\star(s)\big\|_\infty.$$

With $\eta \leq \frac{1}{8H}$, we have $(1 - \eta\tau)(1 - 4\eta H) \geq 2\eta H$ and $1 - \eta(\tau + 2H) \geq 0$. It follows that

$$\mathsf{KL}_s\big(\zeta_{h,\tau}^\star \,\|\, \zeta_h^{(t+1)}\big) + (1 - 4\eta H)\mathsf{KL}_s\big(\zeta_h^{(t+1)} \,\|\, \bar{\zeta}_h^{(t+1)}\big) + \eta\tau\mathsf{KL}_s\big(\bar{\zeta}_h^{(t+1)} \,\|\, \zeta_{h,\tau}^\star\big)$$
$$\leq (1 - \eta\tau)\mathsf{KL}_s\big(\zeta_{h,\tau}^\star \,\|\, \zeta_h^{(t)}\big) + 2\eta H \mathsf{KL}_s\big(\zeta_h^{(t)} \,\|\, \bar{\zeta}_h^{(t)}\big)$$
$$+ \frac{5\eta}{2}\big\|Q_h^{(t+1)}(s) - Q_\tau^\star(s)\big\|_\infty + \frac{\eta}{2}\big\|Q_h^{(t)}(s) - Q_\tau^\star(s)\big\|_\infty$$
$$\leq (1 - \eta\tau)\Big(\mathsf{KL}_s\big(\zeta_{h,\tau}^\star \,\|\, \zeta_h^{(t)}\big) + (1 - 4\eta H)\mathsf{KL}_s\big(\zeta_h^{(t)} \,\|\, \bar{\zeta}_h^{(t)}\big)\Big)$$
$$+ \frac{5\eta}{2}\big\|Q_h^{(t+1)}(s) - Q_\tau^\star(s)\big\|_\infty + \frac{\eta}{2}\big\|Q_h^{(t)}(s) - Q_\tau^\star(s)\big\|_\infty.$$

Therefore, it holds for $0 \leq t_1 < t_2$ that

$$\mathsf{KL}_s\big(\zeta_{h,\tau}^\star \,\|\, \zeta_h^{(t_2)}\big) + (1 - 4\eta H)\mathsf{KL}_s\big(\zeta_h^{(t_2)} \,\|\, \bar{\zeta}_h^{(t_2)}\big) + \eta\tau\mathsf{KL}_s\big(\bar{\zeta}_h^{(t_2)} \,\|\, \zeta_{h,\tau}^\star\big)$$
$$\leq (1 - \eta\tau)^{t_2 - t_1}\Big(\mathsf{KL}_s\big(\zeta_{h,\tau}^\star \,\|\, \zeta_h^{(t_1)}\big) + (1 - 4\eta H)\mathsf{KL}_s\big(\zeta_h^{(t_1)} \,\|\, \bar{\zeta}_h^{t_1}\big)\Big)$$
$$+ \sum_{t'=t_1+1}^{t_2} (1 - \eta\tau)^{t_2 - l}\Big[\frac{5\eta}{2}\big\|Q_h^{(l)}(s) - Q_\tau^\star(s)\big\|_\infty + \frac{\eta}{2}\big\|Q_h^{(l-1)}(s) - Q_\tau^\star(s)\big\|_\infty\Big]$$
$$\leq (1 - \eta\tau)^{t_2 - t_1}\Big(\mathsf{KL}_s\big(\zeta_{h,\tau}^\star \,\|\, \zeta_h^{(t_1)}\big) + (1 - 4\eta H)\mathsf{KL}_s\big(\zeta_h^{(t_1)} \,\|\, \bar{\zeta}_h^{(t_1)}\big)\Big)$$
$$+ 4\eta \sum_{l=t_1}^{t_2} (1 - \eta\tau)^{t_2 - l}\big\|Q_h^{(l)}(s) - Q_\tau^\star(s)\big\|_\infty.$$

## D.2   PROOF OF LEMMA 10

For $t_2 > 0$, we have

$$Q_{h-1}^{(t_2)}(s, a, b) - Q_{h-1,\tau}^\star(s, a, b)$$
$$= \mathop{\mathbb{E}}_{s' \sim P_{h-1}(\cdot|s,a,b)}\Big[V_h^{(t_2 - 1)}(s') - V_{h,\tau}^\star(s')\Big]$$
$$= \mathop{\mathbb{E}}_{s' \sim P_{h-1}(\cdot|s,a,b)}\Big[(1 - \eta\tau)^{t_2 - t_1}\big(V_h^{(t_1 - 1)}(s') - V_{h,\tau}^\star(s')\big)$$
$$+ \eta\tau \sum_{l=t_1}^{t_2 - 1}(1 - \eta\tau)^{t_2 - 1 - l}\big(f_{s'}(Q^{(t_1)}, \bar{\mu}_h^{(t_1)}, \bar{\nu}_h^{(t_1)}) - f_{s'}(Q_{h,\tau}^\star, \mu_{h,\tau}^\star, \nu_{h,\tau}^\star)\big)\Big]$$
$$\leq (1 - \eta\tau)^{t_2 - t_1}2H + \mathop{\mathbb{E}}_{s' \sim P_{h-1}(\cdot|s,a,b)}\Big[\eta\tau \sum_{l=t_1}^{t_2 - 1}(1 - \eta\tau)^{t_2 - 1 - l}\big(f_{s'}(Q_h^{(l)}, \bar{\mu}_h^{(l)}, \bar{\nu}_h^{(l)}) - f_{s'}(Q_{h,\tau}^\star, \mu_{h,\tau}^\star, \nu_{h,\tau}^\star)\big)\Big].$$
$$\tag{48}$$

We start by decomposing $f_s^{(t)} - f_s^\star$ as

$$
\begin{aligned}
&f_s(Q_h^{(t)}, \bar\mu_h^{(t)}, \bar\nu_h^{(t)}) - f_s(Q_{h,\tau}^\star, \mu_{h,\tau}^\star, \nu_{h,\tau}^\star) \\
&= \Big( f_s(Q_h^{(t)}, \bar\mu_h^{(t)}, \bar\nu_h^{(t)}) - f_s(Q_h^{(t)}, \bar\mu_h^{(t)}, \nu_{h,\tau}^\star) \Big) + f_s(Q_h^{(t)}, \bar\mu_h^{(t)}, \nu_{h,\tau}^\star) - f_s(Q_{h,\tau}^\star, \mu_{h,\tau}^\star, \nu_{h,\tau}^\star) \\
&\le \Big( f_s(Q_h^{(t)}, \bar\mu_h^{(t)}, \bar\nu_h^{(t)}) - f_s(Q_h^{(t)}, \bar\mu_h^{(t)}, \nu_{h,\tau}^\star) \Big) + f_s(Q_\tau^\star, \bar\mu^{(t)}, \nu_{h,\tau}^\star) - f_s(Q_{h,\tau}^\star, \mu_{h,\tau}^\star, \nu_{h,\tau}^\star) \\
&\quad + \big\| Q_h^{(t)}(s) - Q_{h,\tau}^\star(s) \big\|_\infty \\
&\le f_s(Q_h^{(t)}, \bar\mu_h^{(t)}, \bar\nu_h^{(t)}) - f_s(Q_h^{(t)}, \bar\mu_h^{(t)}, \nu_{h,\tau}^\star) + \big\| Q_h^{(t)}(s) - Q_{h,\tau}^\star(s) \big\|_\infty.
\end{aligned}
$$

Note that Lemma 13 can be applied to the episodic setting by simply replacing $1/(1-\gamma)$ with $H$, which yields

$$
\begin{aligned}
&f_s(Q_h^{(t)}, \bar\mu_h^{(t)}, \bar\nu_h^{(t)}) - f_s(Q_{h,\tau}^\star, \mu_{h,\tau}^\star, \nu_{h,\tau}^\star) \\
&\le \big\| Q_h^{(t)}(s) - Q_{h,\tau}^\star(s) \big\|_\infty + 2\eta H \big\| Q_h^{(t)}(s) - Q_h^{(t-1)}(s) \big\|_\infty \\
&\quad + \frac{1-\eta\tau}{\eta} \mathsf{KL}_s\big(\nu_{h,\tau}^\star \,\|\, \nu_h^{(t-1)}\big) - \frac{1}{\eta} \mathsf{KL}_s\big(\nu_{h,\tau}^\star \,\|\, \nu_h^{(t)}\big) \\
&\quad - \frac{1}{\eta}\big(1 - 4\eta H\big) \mathsf{KL}_s\big(\nu_h^{(t)} \,\|\, \bar\nu_h^{(t)}\big) - \frac{1-\eta\tau}{\eta} \mathsf{KL}_s\big(\bar\nu_h^{(t)} \,\|\, \nu_h^{(t-1)}\big) \\
&\quad + 2H\Big( \mathsf{KL}_s\big(\bar\mu_h^{(t)} \,\|\, \mu_h^{(t-1)}\big) + \mathsf{KL}_s\big(\mu_h^{(t-1)} \,\|\, \bar\mu_h^{(t-1)}\big) \Big).
\end{aligned}
\tag{49}
$$

By a similar argument,

$$
\begin{aligned}
&f_s(Q_{h,\tau}^\star, \mu_{h,\tau}^\star, \nu_{h,\tau}^\star) - f_s(Q_h^{(t)}, \bar\mu_h^{(t)}, \bar\nu_h^{(t)}) \\
&\le \big\| Q_h^{(t)}(s) - Q_{h,\tau}^\star(s) \big\|_\infty + 2\eta H \big\| Q_h^{(t)}(s) - Q_h^{(t-1)}(s) \big\|_\infty \\
&\quad + \frac{1-\eta\tau}{\eta} \mathsf{KL}_s\big(\mu_{h,\tau}^\star \,\|\, \mu_h^{(t-1)}\big) - \frac{1}{\eta} \mathsf{KL}_s\big(\mu_{h,\tau}^\star \,\|\, \mu_h^{(t)}\big) \\
&\quad - \frac{1}{\eta}\big(1 - 4\eta H\big) \mathsf{KL}_s\big(\mu_h^{(t)} \,\|\, \bar\mu_h^{(t)}\big) - \frac{1-\eta\tau}{\eta} \mathsf{KL}_s\big(\bar\mu_h^{(t)} \,\|\, \mu_h^{(t-1)}\big) \\
&\quad + 2H\Big( \mathsf{KL}_s\big(\bar\nu_h^{(t)} \,\|\, \nu_h^{(t-1)}\big) + \mathsf{KL}_s\big(\nu_h^{(t-1)} \,\|\, \bar\nu_h^{(t-1)}\big) \Big).
\end{aligned}
\tag{50}
$$

Computing (49) $+ \frac{2}{3} \cdot$ (50) gives

$$
\begin{aligned}
&\frac{1}{3}\big[ f_s(Q_h^{(t)}, \bar\mu_h^{(t)}, \bar\nu_h^{(t)}) - f_s(Q_{h,\tau}^\star, \mu_{h,\tau}^\star, \nu_{h,\tau}^\star) \big] \\
&\le \frac{5}{3}\big[ \big\| Q_h^{(t)}(s) - Q_{h,\tau}^\star(s) \big\|_\infty + 2\eta H \big\| Q_h^{(t)}(s) - Q_h^{(t-1)}(s) \big\|_\infty \big] \\
&\quad + \frac{1-\eta\tau}{\eta} \Big[ \mathsf{KL}_s\big(\nu_{h,\tau}^\star \,\|\, \nu_h^{(t-1)}\big) + \frac{2}{3}\mathsf{KL}_s\big(\mu_{h,\tau}^\star \,\|\, \mu_h^{(t-1)}\big) \Big] - \frac{1}{\eta}\Big[ \mathsf{KL}_s\big(\nu_{h,\tau}^\star \,\|\, \nu_h^{(t)}\big) + \frac{2}{3}\mathsf{KL}_s\big(\mu_{h,\tau}^\star \,\|\, \mu_h^{(t)}\big) \Big] \\
&\quad + 2H\Big[ \mathsf{KL}_s\big(\mu_h^{(t-1)} \,\|\, \bar\mu_h^{(t-1)}\big) + \frac{2}{3}\mathsf{KL}_s\big(\nu_h^{(t-1)} \,\|\, \bar\nu_h^{(t-1)}\big) \Big] \\
&\quad - \frac{1}{\eta}\big(1 - 4\eta H\big)\Big[ \frac{2}{3}\mathsf{KL}_s\big(\mu_h^{(t)} \,\|\, \bar\mu_h^{(t)}\big) + \mathsf{KL}_s\big(\nu_h^{(t)} \,\|\, \bar\nu_h^{(t)}\big) \Big] \\
&\quad + \Big( 2H - \frac{1-\eta\tau}{\eta} \cdot \frac{2}{3} \Big) \mathsf{KL}_s\big(\bar\mu^{(t)} \,\|\, \mu^{(t-1)}\big) + \Big( 2H \cdot \frac{2}{3} - \frac{1-\eta\tau}{\eta} \Big) \mathsf{KL}_s\big(\bar\nu^{(t)} \,\|\, \nu^{(t-1)}\big).
\end{aligned}
\tag{51}
$$

With $\eta \le \frac{1}{8H}$, we have

$$
2H - \frac{1-\eta\tau}{\eta} \cdot \frac{2}{3} \le 0, \quad 2H \cdot \frac{2}{3} - \frac{1-\eta\tau}{\eta} \le 0, \quad \text{and} \quad \frac{1}{\eta}(1-\eta\tau)(1-4\eta H) \cdot \frac{2}{3} \ge 2H.
$$

Let

$$
G_h^{(t)}(s) = \mathsf{KL}_s\big(\nu_{h,\tau}^\star \,\|\, \nu_h^{(t)}\big) + \frac{2}{3}\mathsf{KL}_s\big(\mu_{h,\tau}^\star \,\|\, \mu_h^{(t)}\big)
$$

$$+ \frac{2}{3}(1 - 4\eta H)\Big[\mathsf{KL}_s\big(\mu_h^{(t)} \,\|\, \bar{\mu}_h^{(t)}\big) + \mathsf{KL}_s\big(\nu_h^{(t)} \,\|\, \bar{\nu}_h^{(t)}\big)\Big].$$

We can simplify (51) as

$$f_s(Q_h^{(t)}, \bar{\mu}_h^{(t)}, \bar{\nu}_h^{(t)}) - f_s(Q_{h,\tau}^\star, \mu_{h,\tau}^\star, \nu_{h,\tau}^\star)$$
$$\leq 5\big[\big\|Q_h^{(t)}(s) - Q_{h,\tau}^\star(s)\big\|_\infty + 2\eta H\big\|Q_h^{(t)}(s) - Q_h^{(t-1)}(s)\big\|_\infty\big] + \frac{1 - \eta\tau}{\eta}G_h^{(t-1)}(s) - \frac{1}{\eta}G_h^{(t)}(s).$$

Plugging the above inequality into (48) gives

$$Q_{h-1}^{(t_2)}(s, a, b) - Q_{h-1,\tau}^\star(s, a, b)$$
$$\leq (1 - \eta\tau)^{t_2 - t_1} 2H$$
$$\quad + \mathop{\mathbb{E}}_{s' \sim P_{h-1}(\cdot|s,a,b)}\left[5\eta\tau \sum_{l=t_1}^{t_2-1}(1 - \eta\tau)^{t_2-1-l}\big(\big\|Q_h^{(l)}(s') - Q_{h,\tau}^\star(s')\big\|_\infty + 2\eta H\big\|Q_h^{(l)}(s') - Q_h^{(l-1)}(s')\big\|_\infty\big)\right]$$
$$\quad + \mathop{\mathbb{E}}_{s' \sim P_{h-1}(\cdot|s,a,b)}\left[\tau(1 - \eta\tau)^{t_2-t_1}G_h^{(t_1-1)}(s')\right]$$
$$\leq (1 - \eta\tau)^{t_2-t_1}2H$$
$$\quad + 10\eta\tau \mathop{\mathbb{E}}_{s' \sim P_{h-1}(\cdot|s,a,b)}\left[\sum_{l=t_1-1}^{t_2-1}(1 - \eta\tau)^{t_2-1-l}\big\|Q_h^{(l)}(s') - Q_{h,\tau}^\star(s')\big\|_\infty\right]$$
$$\quad + \tau(1 - \eta\tau)^{t_2-t_1} \mathop{\mathbb{E}}_{s' \sim P_{h-1}(\cdot|s,a,b)}\left[\mathsf{KL}_{s'}\big(\zeta_{h,\tau}^\star \,\|\, \zeta_h^{(t_1-1)}\big) + (1 - 4\eta H)\mathsf{KL}_{s'}\big(\zeta_h^{(t_1-1)} \,\|\, \bar{\zeta}_h^{(t_1-1)}\big)\right].$$

# E  PROOF OF AUXILIARY LEMMAS

## E.1  PROOF OF LEMMA 11

We first single out a set of bounds for $V^{(t)}$ and $Q^{(t)}$, which can be obtained by a simple induction:

$$\forall(s, a, b) \in \mathcal{S} \times \mathcal{A} \times \mathcal{B}, \quad \begin{cases} -\frac{\tau\log|\mathcal{B}|}{1-\gamma} \leq V^{(t)}(s) \leq \frac{1+\tau\log|\mathcal{A}|}{1-\gamma} \\ -\frac{\gamma\tau\log|\mathcal{B}|}{1-\gamma} \leq Q^{(t)}(s, a, b) \leq \frac{1+\gamma\tau\log|\mathcal{A}|}{1-\gamma} \end{cases}. \tag{52}$$

We invoke the following lemma to bound several key quantities that will be helpful in the analysis.

**Lemma 15** ((Mei et al., 2020, Lemma 24)). *Let $\pi, \pi' \in \Delta(\mathcal{A})$ such that $\pi(a) \propto \exp(\theta(a))$, $\pi'(a) \propto \theta'(a)$ for some $\theta, \theta' \in \mathbb{R}^{|\mathcal{A}|}$. It holds that*

$$\big\|\pi - \pi'\big\|_1 \leq \big\|\theta - \theta'\big\|_\infty.$$

With this lemma in mind, for any $t \geq 0$, it follows that

$$\big\|\bar{\mu}^{(t+1)}(s) - \mu^{(t+1)}(s)\big\|_1 \leq \min_{c \in \mathbb{R}}\big\|\log\bar{\mu}^{(t+1)}(s) - \log\mu^{(t+1)}(s) - c \cdot \mathbf{1}\big\|_\infty$$
$$\leq \eta\big\|Q^{(t)}(s)\bar{\nu}^{(t)}(s) - Q^{(t+1)}(s)\bar{\nu}^{(t+1)}(s)\big\|_\infty$$
$$\leq \eta \cdot \frac{1 + \gamma\tau(\log|\mathcal{A}| + \log|\mathcal{B}|)}{1 - \gamma} \leq \frac{2\eta}{1 - \gamma},$$

and a similar argument reveals that

$$\big\|\bar{\nu}^{(t+1)}(s) - \nu^{(t+1)}(s)\big\|_1 \leq \frac{2\eta}{1 - \gamma}.$$

Next we make note of the fact that when $t \geq 1$,

$$\bar{\mu}^{(t+1)}(a|s) \propto \mu^{(t)}(a|s)^{1-\eta\tau}\exp(\eta Q^{(t)}(s)\bar{\nu}^{(t)}(s))$$
$$\propto \bar{\mu}^{(t)}(a|s)^{1-\eta\tau}\exp\Big(\eta\big[Q^{(t)}(s)\bar{\nu}^{(t)}(s) + (1 - \eta\tau)(Q^{(t)}(s)\bar{\nu}^{(t)}(s) - Q^{(t-1)}(s)\bar{\nu}^{(t-1)}(s))\big]\Big)$$
$$\propto \bar{\mu}^{(t)}(a|s)\exp(\eta w^{(t)}(a)),$$
$$\tag{53}$$

where

$$w^{(t)} = Q^{(t)}(s)\bar{\nu}^{(t)}(s) + (1 - \eta\tau)\big(Q^{(t)}(s)\bar{\nu}^{(t)}(s) - Q^{(t-1)}(s)\bar{\nu}^{(t-1)}(s)\big) - \tau\log\bar{\mu}^{(t)}(s)$$

satisfies

$$\|w^{(t)}\|_\infty$$
$$\leq \|Q^{(t)}(s)\bar{\nu}^{(t)}(s)\|_\infty + \|\tau\log\bar{\mu}^{(t)}(s)\|_\infty + (1-\eta\tau)\|Q^{(t)}(s)\bar{\nu}^{(t)}(s) - Q^{(t-1)}(s)\bar{\nu}^{(t-1)}(s)\|_\infty$$
$$\leq \frac{2}{1-\gamma} + \frac{2}{1-\gamma} + \frac{2(1-\eta\tau)}{1-\gamma} \leq \frac{6}{1-\gamma},$$

where the second step is due to the following bound:

$$\forall t \geq 0, s \in \mathcal{S}, \qquad \max\left\{\|\log\zeta^{(t)}(s)\|_\infty, \|\log\bar{\zeta}^{(t)}(s)\|_\infty\right\} \leq \frac{2}{(1-\gamma)\tau}. \tag{54}$$

Recall that when $t = 0$, we have $\bar{\mu}^{(t+1)} = \bar{\mu}^{(0)}$. So we have

$$\forall s \in \mathcal{S}, t \geq 0, \qquad \|\bar{\mu}^{(t+1)}(s) - \bar{\mu}^{(t)}(s)\|_1 \leq \frac{6\eta}{1-\gamma}.$$

It remains to prove the claim (54).

*Proof.* It is worth noting that $\mu^{(t)}(s)$ can be always written as $\mu^{(t)}(a|s) \propto \exp(w^{(t)}(a)/\tau)$ for some $w^{(t)} \in \mathbb{R}^{|\mathcal{A}|}$ satisfying

$$\forall a \in \mathcal{A}, \qquad -\frac{\gamma\tau\log|\mathcal{B}|}{1-\gamma} \leq w^{(t)}(a) \leq \frac{1+\gamma\tau\log|\mathcal{A}|}{1-\gamma}.$$

To see this, note that the claim trivially holds for $t = 0$ with $w^{(0)} = \mathbf{0}$. When the statement holds for some $t \geq 0$, we have

$$\mu^{(t+1)}(a|s) \propto \mu^{(t)}(a|s)^{1-\eta\tau}\exp(\eta Q^{(t+1)}(s)\bar{\nu}^{(t+1)}(s))$$
$$\propto \exp\big(((1-\eta\tau)w^{(t)} + \eta\tau Q^{(t+1)}(s)\bar{\nu}^{(t+1)}(s))/\tau\big)$$
$$\propto \exp\big(w^{(t+1)}/\tau\big),$$

with $w^{(t+1)} = (1-\eta\tau)w^{(t)} + \eta\tau Q^{(t+1)}(s)\bar{\nu}^{(t+1)}(s)$. We conclude that the claim holds for $t+1$ by recalling (52). It then follows straightforwardly that

$$\frac{\mu^{(t)}(a_1)}{\mu^{(t)}(a_2)} = \exp\left(\frac{w^{(t)}(a_1) - w^{(t)}(a_2)}{\tau}\right) \leq \exp\left(\frac{1+\gamma\tau(\log|\mathcal{A}|+\log|\mathcal{B}|)}{(1-\gamma)\tau}\right)$$

for any $a_1, a_2 \in \mathcal{A}$. This allows us to show that

$$\min_{a \in \mathcal{A}} \mu^{(t)}(a) \geq \frac{1}{|\mathcal{A}|\exp\left(\frac{1+\gamma\tau(\log|\mathcal{A}|+\log|\mathcal{B}|)}{(1-\gamma)\tau}\right)}\sum_{a \in \mathcal{A}}\mu^{(t)}(a) = \frac{1}{|\mathcal{A}|\exp\left(\frac{1+\gamma\tau(\log|\mathcal{A}|+\log|\mathcal{B}|)}{(1-\gamma)\tau}\right)},$$

which gives

$$\|\log\mu^{(t)}\|_\infty \leq \frac{1+\gamma\tau(\log|\mathcal{A}|+\log|\mathcal{B}|)}{(1-\gamma)\tau} + \log|\mathcal{A}| \leq \frac{1}{(1-\gamma)\tau} + \frac{\log|\mathcal{A}|+\gamma\log|\mathcal{B}|}{1-\gamma}$$
$$\leq \frac{2}{(1-\gamma)\tau}.$$

$\square$

### E.2 PROOF OF LEMMA 12

We decompose the term $f_s(Q^{(t+1)}, \bar{\mu}^{(t+1)}, \bar{\nu}^{(t+1)}) - f_s(Q^{(t)}, \bar{\mu}^{(t)}, \bar{\nu}^{(t)})$ as follows:

$$f_s(Q^{(t+1)}, \bar{\mu}^{(t+1)}, \bar{\nu}^{(t+1)}) - f_s(Q^{(t)}, \bar{\mu}^{(t)}, \bar{\nu}^{(t)})$$
$$= f_s(Q^{(t+1)}, \bar{\mu}^{(t+1)}, \bar{\nu}^{(t+1)}) - f_s(Q^{(t)}, \bar{\mu}^{(t+1)}, \bar{\nu}^{(t+1)}) + f_s(Q^{(t)}, \bar{\mu}^{(t+1)}, \bar{\nu}^{(t+1)}) - f_s(Q^{(t)}, \bar{\mu}^{(t)}, \bar{\nu}^{(t)})$$

$$= \bar{\mu}^{(t+1)}(s)^\top \Big( Q^{(t+1)}(s) - Q^{(t)}(s) \Big) \bar{\nu}^{(t+1)}(s)$$
$$+ f_s(Q^{(t)}, \bar{\mu}^{(t+1)}, \bar{\nu}^{(t)}) - f_s(Q^{(t)}, \bar{\mu}^{(t)}, \bar{\nu}^{(t)}) + f_s(Q^{(t)}, \bar{\mu}^{(t)}, \bar{\nu}^{(t+1)}) - f_s(Q^{(t)}, \bar{\mu}^{(t)}, \bar{\nu}^{(t)})$$
$$+ \Big[ f_s(Q^{(t)}, \bar{\mu}^{(t+1)}, \bar{\nu}^{(t+1)}) + f_s(Q^{(t)}, \bar{\mu}^{(t)}, \bar{\nu}^{(t)}) - f_s(Q^{(t)}, \bar{\mu}^{(t+1)}, \bar{\nu}^{(t)}) - f_s(Q^{(t)}, \bar{\mu}^{(t)}, \bar{\nu}^{(t+1)}) \Big]$$

Note that $\big| \bar{\mu}^{(t+1)}(s)^\top (Q^{(t+1)}(s) - Q^{(t)}(s)) \bar{\nu}^{(t+1)}(s) \big| \le \big\| Q^{(t+1)}(s) - Q^{(t)}(s) \big\|_\infty$. For the terms in the bracket, we have

$$\Big| \Big[ f_s(Q^{(t)}, \bar{\mu}^{(t+1)}, \bar{\nu}^{(t+1)}) + f_s(Q^{(t)}, \bar{\mu}^{(t)}, \bar{\nu}^{(t)}) - f_s(Q^{(t)}, \bar{\mu}^{(t+1)}, \bar{\nu}^{(t)}) - f_s(Q^{(t)}, \bar{\mu}^{(t)}, \bar{\nu}^{(t+1)}) \Big] \Big|$$
$$= \Big| \big( \bar{\mu}^{(t+1)}(s) - \bar{\mu}^{(t)}(s) \big)^\top Q^{(t)}(s) \big( \bar{\nu}^{(t+1)}(s) - \bar{\nu}^{(t)}(s) \big) \Big|$$
$$\le \frac{2}{1-\gamma} \mathsf{KL}_s \big( \bar{\zeta}^{(t+1)} \,\|\, \bar{\zeta}^{(t)} \big).$$

It remains to bound the two difference terms $\big| f_s(Q^{(t)}, \bar{\mu}^{(t+1)}, \bar{\nu}^{(t)}) - f_s(Q^{(t)}, \bar{\mu}^{(t)}, \bar{\nu}^{(t)}) \big|$ and $\big| f_s(Q^{(t)}, \bar{\mu}^{(t)}, \bar{\nu}^{(t+1)}) - f_s(Q^{(t)}, \bar{\mu}^{(t)}, \bar{\nu}^{(t)}) \big|$. To proceed, we show that

$$f_s(Q^{(t)}, \bar{\mu}^{(t)}, \bar{\nu}^{(t)}) - f_s(Q^{(t)}, \bar{\mu}^{(t+1)}, \bar{\nu}^{(t)})$$
$$= \big\langle \bar{\mu}^{(t)}(s) - \bar{\nu}^{(t+1)}(s), Q^{(t)}(s)^\top \bar{\mu}^{(t)}(s) \big\rangle + \tau \mathcal{H}(\bar{\mu}^{(t)}(s)) - \tau \mathcal{H}(\bar{\mu}^{(t+1)}(s))$$
$$= \big\langle \bar{\mu}^{(t)}(s) - \bar{\mu}^{(t+1)}(s), Q^{(t)}(s)^\top \bar{\nu}^{(t)}(s) + (1-\eta\tau)\big( Q^{(t)}(s)\bar{\nu}^{(t)}(s) - Q^{(t-1)}(s)\bar{\nu}^{(t-1)}(s) \big) \big\rangle$$
$$\quad + \tau \mathcal{H}(\bar{\mu}^{(t)}(s)) - \tau \mathcal{H}(\bar{\mu}^{(t+1)}(s))$$
$$\quad - (1-\eta\tau)\big\langle \bar{\mu}^{(t)}(s) - \bar{\mu}^{(t+1)}(s), Q^{(t)}(s)\bar{\nu}^{(t)}(s) - Q^{(t-1)}(s)\bar{\nu}^{(t-1)}(s) \big\rangle$$
$$= -\frac{1}{\eta} \mathsf{KL}_s \big( \bar{\mu}^{(t)} \,\|\, \bar{\mu}^{(t+1)} \big) - \frac{1-\eta\tau}{\eta} \mathsf{KL}_s \big( \bar{\mu}^{(t+1)} \,\|\, \bar{\mu}^{(t)} \big)$$
$$\quad - (1-\eta\tau)\big\langle \bar{\mu}^{(t)}(s) - \bar{\mu}^{(t+1)}(s), Q^{(t)}(s)\bar{\nu}^{(t)}(s) - Q^{(t-1)}(s)\bar{\nu}^{(t-1)}(s) \big\rangle \tag{55}$$

Here, the third step results from Lemma 17 along with (53). Recall from previous discussion (cf. (53)) that $\bar{\mu}^{(t+1)}(a|s) \propto \bar{\mu}^{(t)}(a|s) \exp(\eta w^{(t)}(s))$ with some $w^{(t)} \in \mathbb{R}^{|\mathcal{B}|}$ satisfying

$$\big\| w^{(t)} \big\|_\infty \le \frac{6}{1-\gamma}.$$

We can ensure that $\| \eta w^{(t)} \|_\infty \le 1/30$ with $\eta^{-1} \ge \frac{180}{1-\gamma}$, and the next lemma guarantees $\mathsf{KL}_s \big( \bar{\mu}^{(t)} \,\|\, \bar{\mu}^{(t+1)} \big) \le 2\mathsf{KL}_s \big( \bar{\mu}^{(t+1)} \,\|\, \bar{\mu}^{(t)} \big)$ in this case.

**Lemma 16.** *Let $w \in \mathbb{R}^{|\mathcal{A}|}$, $\pi, \pi' \in \Delta(\mathcal{A})$ satisfy, for each $a \in \mathcal{A}$, $\pi'(a) \propto \pi(a)\exp(w(a))$ with $\|w\|_\infty \le \frac{1}{30}$. It holds that*
$$\mathsf{KL}\big( \pi \,\|\, \pi' \big) \le 2\mathsf{KL}\big( \pi' \,\|\, \pi \big).$$

Therefore, we can continue (55) by showing that

$$\big| f_s(Q^{(t)}, \bar{\mu}^{(t+1)}, \bar{\nu}^{(t)}) - f_s(Q^{(t)}, \bar{\mu}^{(t)}, \bar{\nu}^{(t)}) \big|$$
$$\le \frac{1}{\eta} \mathsf{KL}_s \big( \bar{\mu}^{(t)} \,\|\, \bar{\mu}^{(t+1)} \big) + \frac{1-\eta\tau}{\eta} \mathsf{KL}_s \big( \bar{\mu}^{(t+1)} \,\|\, \bar{\mu}^{(t)} \big)$$
$$\quad + \big\| \bar{\mu}^{(t+1)}(s) - \bar{\mu}^{(t)}(s) \big\|_1 \big\| Q^{(t)}(s)\bar{\nu}^{(t)}(s) - Q^{(t-1)}(s)\bar{\nu}^{(t-1)}(s)) \big\|_\infty$$
$$\le \frac{3}{\eta} \mathsf{KL}_s \big( \bar{\mu}^{(t+1)} \,\|\, \bar{\mu}^{(t)} \big) + \big\| \bar{\mu}^{(t+1)}(s) - \bar{\mu}^{(t)}(s) \big\|_1 \big\| Q^{(t)}(s) - Q^{(t-1)}(s) \big\|_\infty$$
$$\quad + \big\| Q^{(t)}(s) \big\|_\infty \big\| \bar{\mu}^{(t+1)}(s) - \bar{\mu}^{(t)}(s) \big\|_1 \big\| \bar{\nu}^{(t)}(s) - \bar{\nu}^{(t-1)}(s) \big\|_1$$
$$\le \Big( \frac{3}{\eta} + \frac{2}{1-\gamma} \Big) \mathsf{KL}_s \big( \bar{\mu}^{(t+1)} \,\|\, \bar{\mu}^{(t)} \big) + \frac{2}{1-\gamma} \mathsf{KL}_s \big( \bar{\mu}^{(t)} \,\|\, \bar{\mu}^{(t-1)} \big) + \frac{6\eta}{1-\gamma} \big\| Q^{(t)}(s) - Q^{(t-1)}(s) \big\|_\infty$$

One can bound $\big| f_s(Q^{(t)}, \bar{\mu}^{(t)}, \bar{\nu}^{(t)}) - f_s(Q^{(t)}, \bar{\mu}^{(t)}, \bar{\nu}^{(t+1)}) \big|$ with similar argument. Putting all pieces together, we arrive at

$$\big| f_s(Q^{(t+1)}, \bar{\mu}^{(t+1)}, \bar{\nu}^{(t+1)}) - f_s(Q^{(t)}, \bar{\mu}^{(t)}, \bar{\nu}^{(t)}) \big|$$

$$\leq \left\| Q^{(t+1)}(s) - Q^{(t)}(s) \right\|_{\infty} + \left( \frac{3}{\eta} + \frac{4}{1-\gamma} \right) \mathsf{KL}_s\big(\bar{\zeta}^{(t+1)} \,\|\, \bar{\zeta}^{(t)}\big) + \frac{2}{1-\gamma} \mathsf{KL}_s\big(\bar{\zeta}^{(t)} \,\|\, \bar{\zeta}^{(t-1)}\big)$$
$$+ \frac{12\eta}{1-\gamma} \left\| Q^{(t)}(s) - Q^{(t-1)}(s) \right\|_{\infty}.$$

### E.3 PROOF OF LEMMA 13

$$\big\langle \bar{\nu}^{(t)}(s) - \nu_{\tau}^{\star}(s), Q^{(t)}(s)^{\top} \bar{\mu}^{(t)}(s) \big\rangle - \tau\mathcal{H}(\bar{\nu}^{(t)}(s)) + \tau\mathcal{H}(\nu_{\tau}^{\star}(s))$$
$$= \big\langle \bar{\nu}^{(t)}(s) - \nu^{(t)}(s), Q^{(t)}(s)^{\top} \bar{\mu}^{(t)}(s) - Q^{(t-1)}(s)^{\top} \bar{\mu}^{(t-1)}(s) \big\rangle$$
$$\quad + \big\langle \bar{\nu}^{(t)}(s) - \nu^{(t)}(s), Q^{(t-1)}(s)^{\top} \bar{\mu}^{(t-1)}(s) \big\rangle - \tau\mathcal{H}(\bar{\nu}^{(t)}(s)) + \tau\mathcal{H}(\nu^{(t)}(s))$$
$$\quad + \big\langle \nu^{(t)}(s) - \nu_{\tau}^{\star}(s), Q^{(t)}(s)^{\top} \bar{\mu}^{(t)}(s) \big\rangle - \tau\mathcal{H}(\nu^{(t)}(s)) + \tau\mathcal{H}(\mu_{\tau}^{\star}(s))$$
$$= \big\langle \bar{\nu}^{(t)}(s) - \nu^{(t)}(s), Q^{(t)}(s)^{\top} \bar{\mu}^{(t)}(s) - Q^{(t-1)}(s)^{\top} \bar{\mu}^{(t-1)}(s) \big\rangle$$
$$\quad + \frac{1-\eta\tau}{\eta} \mathsf{KL}_s\big(\nu^{(t)} \,\|\, \nu^{(t-1)}\big) - \frac{1}{\eta} \mathsf{KL}_s\big(\nu^{(t)} \,\|\, \bar{\nu}^{(t)}\big) - \frac{1-\eta\tau}{\eta} \mathsf{KL}_s\big(\bar{\nu}^{(t)} \,\|\, \nu^{(t-1)}\big)$$
$$\quad + \frac{1-\eta\tau}{\eta} \mathsf{KL}_s\big(\nu_{\tau}^{\star} \,\|\, \nu^{(t-1)}\big) - \frac{1}{\eta} \mathsf{KL}_s\big(\nu_{\tau}^{\star} \,\|\, \nu^{(t)}\big) - \frac{1-\eta\tau}{\eta} \mathsf{KL}_s\big(\nu^{(t)} \,\|\, \nu^{(t-1)}\big)$$
$$\leq \big\| \bar{\nu}^{(t)}(s) - \nu^{(t)}(s) \big\|_1 \big\| Q^{(t)}(s)^{\top} \bar{\mu}^{(t)}(s) - Q^{(t-1)}(s)^{\top} \bar{\mu}^{(t-1)}(s) \big\|_{\infty}$$
$$\quad - \frac{1}{\eta} \mathsf{KL}_s\big(\nu^{(t)} \,\|\, \bar{\nu}^{(t)}\big) - \frac{1-\eta\tau}{\eta} \mathsf{KL}_s\big(\bar{\nu}^{(t)} \,\|\, \nu^{(t-1)}\big) + \frac{1-\eta\tau}{\eta} \mathsf{KL}_s\big(\nu_{\tau}^{\star} \,\|\, \nu^{(t-1)}\big) - \frac{1}{\eta} \mathsf{KL}_s\big(\nu_{\tau}^{\star} \,\|\, \nu^{(t)}\big).$$
$$(56)$$

The second step results from the following three-point lemma:

**Lemma 17** (Regularized 3-point lemma). *Let $x \in \Delta(\mathcal{A})$ be defined as*

$$x(a) \propto y(a)^{1-\eta\tau} \exp(-\eta w(a))$$

*for some $w \in \mathbb{R}^{|\mathcal{A}|}$ and $y \in \Delta(\mathcal{A})$. It holds for all $z \in \Delta(\mathcal{A})$ that*

$$\frac{\eta}{1-\eta\tau} \Big[ \langle x - z, w \rangle - \tau\mathcal{H}(x) + \tau\mathcal{H}(z) \Big] = \mathsf{KL}\big(z \,\|\, y\big) - \frac{1}{1-\eta\tau} \mathsf{KL}\big(z \,\|\, x\big) - \mathsf{KL}\big(x \,\|\, y\big).$$

We bound the first term in (56) as follows:

$$\big\| \bar{\nu}^{(t)}(s) - \nu^{(t)}(s) \big\|_1 \big\| Q^{(t)}(s)^{\top} \bar{\mu}^{(t)}(s) - Q^{(t-1)}(s)^{\top} \bar{\mu}^{(t-1)}(s) \big\|_{\infty}$$
$$\leq \big\| \bar{\nu}^{(t)}(s) - \nu^{(t)}(s) \big\|_1 \Big( \big\| \big(Q^{(t)}(s) - Q^{(t-1)}(s)\big)^{\top} \bar{\mu}^{(t-1)}(s) \big\|_{\infty} + \big\| Q^{(t)}(s)\big(\bar{\mu}^{(t)}(s) - \bar{\mu}^{(t-1)}(s)\big) \big\|_{\infty} \Big)$$
$$\leq \big\| \bar{\nu}^{(t)}(s) - \nu^{(t)}(s) \big\|_1 \big\| Q^{(t)}(s) - Q^{(t-1)}(s) \big\|_{\infty} + \frac{2}{1-\gamma} \big\| \bar{\nu}^{(t)}(s) - \nu^{(t)}(s) \big\|_1 \big\| \bar{\mu}^{(t)}(s) - \bar{\mu}^{(t-1)}(s) \big\|_1$$
$$\leq \frac{2\eta}{1-\gamma} \big\| Q^{(t)}(s) - Q^{(t-1)}(s) \big\|_{\infty} + \frac{1}{1-\gamma} \Big[ 2\big\| \bar{\nu}^{(t)}(s) - \nu^{(t)}(s) \big\|_1^2$$
$$\quad + \big\| \bar{\mu}^{(t)}(s) - \mu^{(t-1)}(s) \big\|_1^2 + \big\| \mu^{(t-1)}(s) - \bar{\mu}^{(t-1)}(s) \big\|_1^2 \Big]$$
$$\leq \frac{2\eta}{1-\gamma} \big\| Q^{(t)}(s) - Q^{(t-1)}(s) \big\|_{\infty} + \frac{4}{1-\gamma} \mathsf{KL}_s\big(\nu^{(t)} \,\|\, \bar{\nu}^{(t)}\big)$$
$$\quad + \frac{2}{1-\gamma} \mathsf{KL}\big(\bar{\mu}^{(t)}(s) \,\|\, \mu^{(t-1)}(s)\big) + \frac{2}{1-\gamma} \mathsf{KL}\big(\mu^{(t-1)}(s) \,\|\, \bar{\mu}^{(t-1)}(s)\big).$$

Substitution of the above inequality into (56) completes the proof.

### E.4 PROOF OF LEMMA 14

We have

$$\delta_{l,t} = \alpha_l \prod_{i=l+1}^{t} (1 - c_1 \alpha_i)$$

$$= \alpha_l \prod_{i=l+1}^{t} (1 - c_2\alpha_i + (c_2 - c_1)\alpha_i)$$

$$= \alpha_l(c_2 - c_1)\alpha_{l+1} \prod_{i=l+2}^{t} (1 - c_2\alpha_i + (c_2 - c_1)\alpha_i) + \alpha_l(1 - c_2\alpha_{l+1}) \prod_{i=l+2}^{t} (1 - c_2\alpha_i + (c_2 - c_1)\alpha_i)$$

$$= \alpha_l \sum_{i=l+1}^{t} (c_2 - c_1)\alpha_i \cdot \prod_{j=l+1}^{i} (1 - c_2\alpha_j) \cdot \prod_{k=i+1}^{t} (1 - c_1\alpha_k) + \alpha_l \prod_{i=l+1}^{t} (1 - c_2\alpha_i)$$

$$= (c_2 - c_1) \sum_{i=l+1}^{t} \xi_{l,i}\delta_{i,t} + \xi_{l,t}.$$

Rearranging terms,

$$\sum_{i=l}^{t} \xi_{l,i}\delta_{i+1,t} = \alpha_l\delta_{l+1,t} + \sum_{i=l+1}^{t} \xi_{l,i}\delta_{i+1,t}$$

$$= \frac{\alpha_{l+1}}{1 - c_1\alpha_{l+1}}\delta_{l,t} + \sum_{i=l+1}^{t} \xi_{l,i}\delta_{i,t} \cdot \frac{\alpha_{i+1}}{\alpha_i(1 - c_1\alpha_{i+1})}$$

$$\leq \delta_{l,t} + 2 \sum_{i=l+1}^{t} \xi_{l,i}\delta_{i,t} = \delta_{l,t} + \frac{2}{c_2 - c_1}(\delta_{l,t} - \xi_{l,t}) \leq \left(1 + \frac{2}{c_2 - c_1}\right)\delta_{l,t},$$

where the inequality is due to $\alpha_{l+1} \leq \alpha \leq 1/2$ and $1 - c_1\alpha_l \geq 1/2$ for all $l \geq 1$.

### E.5 PROOF OF LEMMA 16

*Proof.* For any $x > -1$, it holds that

$$\log(1 + x) \leq x - \frac{x^2}{2} + \frac{x^3}{3}$$

$$\leq x - \frac{x^2}{2} + \frac{|x^3|}{3} = x - \left(\frac{1}{2} - \frac{|x|}{3}\right)x^2,$$

and that

$$\log(1 + x) \geq x - \frac{x^2}{2} + \frac{x^3}{3(1 + x)^3}$$

$$\geq x - \frac{x^2}{2} - \frac{|x^3|}{3(1 + x)^3} = x - \left(\frac{1}{2} + \frac{|x|}{3(1 + x)^3}\right)x^2.$$

Therefore, when $x > -\frac{1}{10}$, we have $(1 + x)^3 > \frac{2}{3}$ and thus

$$x - \left(\frac{1}{2} + \frac{|x|}{2}\right)x^2 \leq \log(1 + x) \leq x - \left(\frac{1}{2} - \frac{|x|}{3}\right)x^2.$$

Let $c$ be a shorthand notation for $\|w\|_\infty$. The following lemma is standard (see, e.g., (Mei et al., 2020, Lemma 23), (Cen et al., 2021a, Lemma 3)), which ensures that $\|\log\pi - \log\pi'\|_\infty \leq 2c$.

**Lemma 18.** *Let* $\pi, \pi' \in \Delta(\mathcal{A})$ *satisfy* $\pi(a) \propto \exp(\theta(a))$ *and* $\pi'(a) \propto \exp(\theta'(a))$ *for some* $\theta, \theta' \in \mathbb{R}^{|\mathcal{A}|}$. *It holds that*

$$\|\log\pi - \log\pi'\|_\infty \leq 2\|\theta - \theta'\|_\infty.$$

Since $c < 1/30$, we have

$$\left|\frac{\pi(a)}{\pi'(a)} - 1\right| = \left|\exp\left(\log\frac{\pi(a)}{\pi'(a)}\right) - \exp(0)\right| \leq |\log\pi(a) - \log\pi'(a)|\max\left\{1, \frac{\pi(a)}{\pi'(a)}\right\}$$

$$\leq 2c\exp(|2c|) \leq 3c, \quad \forall a \in \mathcal{A}.$$

Therefore, we can bound $\mathsf{KL}\big(\pi \,\|\, \pi'\big)$ as

$$
\begin{aligned}
\mathsf{KL}\big(\pi \,\|\, \pi'\big) &= \sum_{a \in \mathcal{A}} \pi(a) \log \frac{\pi(a)}{\pi'(a)} \\
&\leq \sum_{a \in \mathcal{A}} \pi(a) \Big( \frac{\pi(a)}{\pi'(a)} - 1 - \big(\frac{1}{2} - c\big)\Big(\frac{\pi(a)}{\pi'(a)} - 1\Big)^2 \Big) \\
&= \chi^2(\pi; \pi') - \big(\frac{1}{2} - c\big) \sum_{a \in \mathcal{A}} \pi(a)\Big(\frac{\pi(a)}{\pi'(a)} - 1\Big)^2 \qquad (57) \\
&\leq \chi^2(\pi; \pi') - \big(\frac{1}{2} - c\big)(1 - 3c) \sum_{a \in \mathcal{A}} \pi'(a)\Big(\frac{\pi(a)}{\pi'(a)} - 1\Big)^2 \\
&= \Big(1 - \big(\frac{1}{2} - c\big)(1 - 3c)\Big)\chi^2(\pi; \pi').
\end{aligned}
$$

On the other hand, we have

$$
\begin{aligned}
\mathsf{KL}\big(\pi' \,\|\, \pi\big) &= \sum_{a \in \mathcal{A}} \pi'(a) \log \frac{\pi'(a)}{\pi(a)} \\
&\geq \sum_{a \in \mathcal{A}} \pi'(a) \Big( \frac{\pi'(a)}{\pi(a)} - 1 - \frac{1 + 3c}{2}\Big(\frac{\pi'(a)}{\pi(a)} - 1\Big)^2 \Big) \\
&= \chi^2(\pi'; \pi) - \frac{1 + 3c}{2} \sum_{a \in \mathcal{A}} \pi'(a)\Big(\frac{\pi'(a)}{\pi(a)} - 1\Big)^2 \qquad (58) \\
&\geq \chi^2(\pi'; \pi) - \frac{(1 + 3c)^2}{2} \sum_{a \in \mathcal{A}} \pi(a)\Big(\frac{\pi'(a)}{\pi(a)} - 1\Big)^2 \\
&= \big(1 - \frac{(1 + 3c)^2}{2}\big)\chi^2(\pi'; \pi).
\end{aligned}
$$

By definition, we have

$$
\begin{aligned}
\chi^2(\pi; \pi') &= \sum_{a \in \mathcal{A}} \pi'(a)\Big(\frac{\pi(a)}{\pi'(a)} - 1\Big)^2 = \sum_{a \in \mathcal{A}} \frac{\big(\pi(a) - \pi'(a)\big)^2}{\pi'(a)} \\
&\leq \big\|\pi/\pi'\big\|_\infty \sum_{a \in \mathcal{A}} \frac{\big(\pi'(a) - \pi(a)\big)^2}{\pi(a)} \qquad (59) \\
&\leq (1 + 3c)\chi^2(\pi'; \pi).
\end{aligned}
$$

Combining (57), (58) and (59) gives

$$
\mathsf{KL}\big(\pi \,\|\, \pi'\big) \leq (1 + 3c) \cdot \frac{1 - \big(1/2 - c\big)(1 - 3c)}{1 - (1 + 3c)^2/2} \mathsf{KL}\big(\pi' \,\|\, \pi\big).
$$

It is straightforward to verify that the factor is less than 2 when $c \leq 1/30$. $\qquad \square$

## E.6 PROOF OF LEMMA 17

*Proof.* We have

$$
\begin{aligned}
\mathsf{KL}\big(z \,\|\, y\big) &= -\mathcal{H}(z) + \mathcal{H}(y) - \big\langle z - y, \log y \big\rangle \\
&= -\mathcal{H}(z) + \mathcal{H}(x) - \big\langle z - x, \log y \big\rangle - \mathcal{H}(x) + \mathcal{H}(y) - \big\langle x - y, \log y \big\rangle \\
&= -\mathcal{H}(z) + \mathcal{H}(x) - \big\langle z - x, \log x \big\rangle - \mathcal{H}(x) + \mathcal{H}(y) - \big\langle x - y, \log y \big\rangle - \big\langle z - x, \log y - \log x \big\rangle \\
&= \mathsf{KL}\big(z \,\|\, x\big) + \mathsf{KL}\big(x \,\|\, y\big) - \frac{\eta}{1 - \eta\tau}\big\langle z - x, w + \tau \log x \big\rangle.
\end{aligned}
$$

Rearranging terms gives

$$
\frac{\eta}{1 - \eta\tau}\big\langle x - z, w \big\rangle = \mathsf{KL}\big(z \,\|\, y\big) - \mathsf{KL}\big(z \,\|\, x\big) - \mathsf{KL}\big(x \,\|\, y\big) + \frac{\eta\tau}{1 - \eta\tau}\big\langle z - x, \log x \big\rangle.
$$

Adding $\frac{\eta\tau}{1-\eta\tau}(-\mathcal{H}(x) + \mathcal{H}(z))$ to both sides, we are left with

$$\frac{\eta}{1-\eta\tau}\Big[\langle x - z, w\rangle - \tau\mathcal{H}(x) + \tau\mathcal{H}(z)\Big] = \mathsf{KL}\big(z\,\|\,y\big) - \mathsf{KL}\big(z\,\|\,x\big) - \mathsf{KL}\big(x\,\|\,y\big)$$

$$-\frac{\eta\tau}{1-\eta\tau}\big(-\mathcal{H}(z) + \mathcal{H}(x) - \langle z - x, \log x\rangle\big)$$

$$= \mathsf{KL}\big(z\,\|\,y\big) - \frac{1}{1-\eta\tau}\mathsf{KL}\big(z\,\|\,x\big) - \mathsf{KL}\big(x\,\|\,y\big).$$

$\square$

## F    FURTHER DISCUSSION REGARDING APPROXIMATE ALGORITHMS

In this section we verify the convergence of the proposed method equipped with inexact value updates in the infinite-horizon setting, where (10) in Algorithm 1 is replaced by

$$\begin{cases} Q^{(t+1)}(s,a,b) &= r(s,a,b) + \gamma\mathbb{E}_{s'\sim P(\cdot|s,a,b)}\big[V^{(t)}(s')\big] + \alpha_t\delta^{(t)}(s,a,b) \\ V^{(t+1)}(s) &= (1-\alpha_{t+1})V^{(t)}(s) \\ &\quad +\alpha_{t+1}\big[\bar{\mu}^{(t+1)}(s)^\top Q^{(t+1)}(s)\bar{\nu}^{(t+1)}(s) + \tau\mathcal{H}\big(\bar{\mu}^{(t+1)}(s)\big) - \tau\mathcal{H}\big(\bar{\nu}^{(t+1)}(s)\big)\big] \end{cases}.$$

or equivalently

$$Q^{(t+1)}(s,a,b) = (1-\alpha_t)Q^{(t)}(s,a,b)$$
$$+ \alpha_t\Big[r(s,a,b) + \gamma\mathbb{E}_{s'\sim P(\cdot|s,a,b)}\big[\bar{\mu}^{(t)}(s)^\top Q^{(t)}(s)\bar{\nu}^{(t)}(s) + \tau\mathcal{H}\big(\bar{\mu}^{(t)}(s)\big) - \tau\mathcal{H}\big(\bar{\nu}^{(t)}(s)\big)\big] + \delta^{(t)}(s,a,b)\Big].$$

Here, $\delta(s,a,b)^{(t+1)} \in \mathbb{R}$ represents the error due to approximate evaluation. For simplicity we focus on the case where the policy update rules (9a), (9b) remain unchanged. The following theorems reveal that the algorithm converges linearly to the QRE until it reaches an error floor determined by $\big\|\delta^{(i)}\big\|_{\Gamma(\rho)}$:

**Theorem 5.** *With $0 < \eta \le \frac{(1-\gamma)^3}{32000\mathcal{C}_\rho}$, and $\alpha_i = \eta\tau$, we have*

$$\max\Big\{\mathsf{KL}_\rho\big(\zeta_\tau^\star\,\|\,\zeta^{(t)}\big), \frac{1}{2}\mathsf{KL}_\rho\big(\zeta_\tau^\star\,\|\,\bar{\zeta}^{(t)}\big), 3\eta\,\mathbb{E}_{s\sim\rho}\big[\big\|Q^{(t)}(s) - Q_\tau^\star(s)\big\|_\infty\big]\Big\}$$

$$\le \frac{3000}{(1-\gamma)^2\tau}\Big(1 - \frac{(1-\gamma)\eta\tau}{4}\Big)^t + \frac{1500}{(1-\gamma)\tau}\max_{0\le i\le t}\big\|\delta^{(i)}\big\|_{\Gamma(\rho)}.$$

**Theorem 6.** *With $0 < \eta \le \frac{(1-\gamma)^3}{32000\mathcal{C}_\rho}$, and $\alpha_i = \eta\tau$, we have*

$$\max_{s\in\mathcal{S},\mu,\nu}\Big(V_\tau^{\mu,\bar{\nu}^{(t)}}(s) - V_\tau^{\bar{\mu}^{(t)},\nu}(s)\Big) \le \frac{2\|1/\rho\|_\infty}{1-\gamma}\max\Big\{\frac{8}{(1-\gamma)^2\tau}, \frac{1}{\eta}\Big\}$$

$$\cdot\Big[\frac{3000}{(1-\gamma)^2\tau}\Big(1 - \frac{(1-\gamma)\eta\tau}{4}\Big)^t + \frac{1500}{(1-\gamma)\tau}\max_{0\le i\le t}\big\|\delta^{(i)}\big\|_{\Gamma(\rho)}\Big],$$

*and*

$$\max_{\mu,\nu}\Big(V_\tau^{\mu,\bar{\nu}^{(t)}}(\rho) - V_\tau^{\bar{\mu}^{(t)},\nu}(\rho)\Big) \le \frac{2\mathcal{C}_{\rho,\tau}^\dagger}{1-\gamma}\max\Big\{\frac{8}{(1-\gamma)^2\tau}, \frac{1}{\eta}\Big\}\Big(1 - \frac{(1-\gamma)\eta\tau}{4}\Big)^t$$

$$\cdot\Big[\frac{3000}{(1-\gamma)^2\tau}\Big(1 - \frac{(1-\gamma)\eta\tau}{4}\Big)^t + \frac{1500}{(1-\gamma)\tau}\max_{0\le i\le t}\big\|\delta^{(i)}\big\|_{\Gamma(\rho)}\Big].$$

We remark that $\big\|\delta^{(t)}\big\|_{\Gamma(\rho)}$ can be bounded either by $\epsilon_{\mathsf{stat}}$ or $\mathcal{C}_\rho\epsilon_{\mathsf{stat}}$ with evaluation error guarantee $\max_{s\in\mathcal{S}}\big\|\delta^{(t)}(s)\big\|_\infty \le \epsilon_{\mathsf{stat}}$ and $\mathbb{E}_{s\sim\rho}\big[\big\|\delta^{(t)}(s)\big\|_\infty\big] \le \epsilon_{\mathsf{stat}}$ respectively.

The remaining part of this section outlines the proof for the above Theorems. For simplicity, we only highlight the key difference from the previous proof due to evaluation error and omit the proof for corresponding lemmas. We first remark that Lemma 1 depends solely on the policy update rules and hence still holds. The error propagation of $\{\delta^{(l)}\}$ is captured by the following lemmas which parallels Lemma 2 and Lemma 16:

**Lemma 19.** *With $0 < \eta \leq \min\{(1-\gamma)/180, (1-\gamma)^2/48\}$, it holds for all $t \geq 1$ that*

$$\left\|Q^{(t+1)} - Q^{(t)}\right\|_{\Gamma(\rho)} \leq \frac{1+\gamma}{2} \sum_{l=1}^{t} \alpha_{l,t} \left\|Q^{(l)} - Q^{(l-1)}\right\|_{\Gamma(\rho)} + \frac{4\mathcal{C}_\rho}{\eta} \cdot \sum_{l=1}^{t} \alpha_{l,t} \mathsf{KL}_\rho\left(\bar{\zeta}^{(l)} \,\|\, \bar{\zeta}^{(l-1)}\right)$$

$$+ \alpha_t \left\|\delta^{(t)}\right\|_{\Gamma(\rho)} + \alpha_{t-1} \left\|\delta^{(t-1)}\right\|_{\Gamma(\rho)}. \tag{60}$$

*When $t = 0$, we have $\left\|Q^{(1)}(s) - Q^{(0)}(s)\right\|_{\Gamma(\rho)} \leq 2 + \alpha_0 \left\|\delta^{(0)}\right\|_{\Gamma(\rho)}$.*

**Lemma 20.** *With $0 < \eta \leq (1-\gamma)^2/16$, it holds for all $t \geq 1$ that*

$$\left\|Q^{(t+1)} - Q_\tau^\star\right\|_{\Gamma(\rho)}$$

$$\leq \frac{1+\gamma}{2} \cdot \sum_{l=0}^{t} \alpha_{l,t}\left(\left\|Q^{(l)} - Q_\tau^\star\right\|_{\Gamma(\rho)} + \frac{2\eta}{1-\gamma}\left\|Q^{(l)} - Q^{(l-1)}\right\|_{\Gamma(\rho)}\right) + 2\alpha_{0,t} + \alpha_t\left\|\delta^{(t)}\right\|_{\Gamma(\rho)} \tag{61}$$

*When $t = 0$, we have $\left\|Q^{(1)} - Q_\tau^\star\right\|_{\Gamma(\rho)} \leq \frac{2\gamma}{1-\gamma} + \alpha_0\left\|\delta^{(0)}\right\|_{\Gamma(\rho)}$.*

Following the similar argument in Lemma 4, we can show that

**Lemma 21.** *Under the assumption of Lemma 19 and 20, it holds for all $t \geq 0$ that*

$$\sum_{l=0}^{t} \lambda_{l+1,t+1}\left[\eta\left\|Q_\tau^\star - Q^{(l+1)}\right\|_{\Gamma(\rho)} + \frac{12\eta^2}{(1-\gamma)^2}\left\|Q^{(l+1)} - Q^{(l)}\right\|_{\Gamma(\rho)}\right]$$

$$\leq \frac{6250\eta\mathcal{C}_\rho}{(1-\gamma)^3} \sum_{l=0}^{t-1} \lambda_{l+1,t+1}\mathsf{KL}\left(\bar{\zeta}^{(l+1)} \,\|\, \bar{\zeta}^{(l)}\right) + \frac{550\eta}{(1-\gamma)^2}\lambda_{0,t+1} + 60\eta \sum_{l=0}^{t} \lambda_{l+1,t+1}\alpha_l\left\|\delta^{(l)}\right\|_{\Gamma(\rho)}.$$

With $\alpha_l = \eta\tau$ for $l \geq 1$, we have

$$\sum_{l=0}^{t} \lambda_{l+1,t+1}\alpha_l\left\|\delta^{(l)}\right\|_{\Gamma(\rho)} \leq \lambda_{1,t+1}\left\|\delta^{(0)}\right\|_{\Gamma(\rho)} + \max_{1\leq i\leq t}\left\|\delta^{(i)}\right\|_{\Gamma(\rho)} \sum_{l=1}^{t} \lambda_{l+1,t+1}\alpha_l$$

$$\leq \lambda_{1,t+1}\left\|\delta^{(0)}\right\|_{\Gamma(\rho)} + \frac{4}{1-\gamma}\max_{1\leq i\leq t}\left\|\delta^{(i)}\right\|_{\Gamma(\rho)} \leq \frac{5}{1-\gamma}\max_{0\leq i\leq t}\left\|\delta^{(i)}\right\|_{\Gamma(\rho)}.$$

It is then straightforward to put together the above lemmas in a similar way to the proof in Appendix A to obtain Theorem 5 and 6.

## G  FURTHER DISCUSSION REGARDING WEI ET AL. (2021B)

This section demonstrates how the last-iterate convergence result in Wei et al. (2021b, Theorem 2) in terms of the Euclidean distance to the set of NEs can be translated to that of the duality gap. Given any policy pair $\zeta = (\mu, \nu)$ and a NE $\zeta^\star = (\mu^\star, \nu^\star)$, we can invoke performance difference lemma (44) and obtain:

$$V^{\mu,\nu}(\rho) - V^\star(\rho) = \frac{1}{1-\gamma} \mathop{\mathbb{E}}_{s'\sim d_\rho^{\mu,\nu}} \left[\mu(s')^\top Q^\star(s')\nu(s') - \mu^\star(s')^\top Q^\star(s')\nu^\star(s')\right]$$

$$\leq \frac{1}{1-\gamma} \mathop{\mathbb{E}}_{s'\sim d_\rho^{\mu,\nu}} \left[\max_{\mu'} \mu'(s')^\top Q^\star(s')\nu(s') - \mu^\star(s')^\top Q^\star(s')\nu^\star(s')\right]$$

$$= \frac{1}{1-\gamma} \mathop{\mathbb{E}}_{s'\sim d_\rho^{\mu,\nu}} \left[\max_{\mu'} \mu'(s')^\top Q^\star(s')\nu(s') - \max_{\mu'} \mu'(s')^\top Q^\star(s')\nu^\star(s')\right]$$

$$\leq \frac{1}{1-\gamma} \mathop{\mathbb{E}}_{s'\sim d_\rho^{\mu,\nu}} \left[\max_{\mu'} \mu'(s')^\top Q^\star(s')\left(\nu(s') - \nu^\star(s')\right)\right]$$

$$\leq \frac{1}{(1-\gamma)^2} \mathop{\mathbb{E}}_{s'\sim d_\rho^{\mu,\nu}} \left[\left\|\nu(s') - \nu^\star(s')\right\|_1\right].$$

Setting $\mu$ to the best-response policy of $\nu$, i.e., $\mu = \mu^\dagger(\nu) := \arg\max_\mu V^{\mu,\nu}(\rho)$, we get

$$\max_{\mu'} V^{\mu',\nu}(\rho) - V^\star(\rho) = V^{\mu^\dagger(\nu),\nu}(\rho) - V^\star(\rho)$$

$$\leq \frac{1}{(1-\gamma)^2} \mathop{\mathbb{E}}_{s' \sim d_\rho^{\mu^\dagger(\nu),\nu}} \left[ \left\| \nu(s') - \nu^\star(s') \right\|_1 \right]$$

$$\leq \frac{\left\| d_\rho^{\mu^\dagger(\nu),\nu} \right\|_\infty}{(1-\gamma)^2} \sum_{s \in \mathcal{S}} \left\| \nu(s) - \nu^\star(s) \right\|_1.$$

Similarly, we have

$$V^\star(\rho) - \min_{\nu'} V^{\mu,\nu'}(\rho) \leq \frac{\left\| d_\rho^{\mu,\nu^\dagger(\mu)} \right\|_\infty}{(1-\gamma)^2} \sum_{s \in \mathcal{S}} \left\| \mu(s') - \mu^\star(s') \right\|_1.$$

Taken together, the duality gap can be bounded by the policy's $\ell_1$ distance to NE $(\mu^\star, \nu^\star)$ as

$$\max_{\mu',\nu'} \left[ V^{\mu',\nu}(\rho) - V^{\mu,\nu'}(\rho) \right] \leq \frac{1}{(1-\gamma)^2} \sum_{s \in \mathcal{S}} \left( \left\| \nu(s') - \nu^\star(s') \right\|_1 + \left\| \mu(s') - \mu^\star(s') \right\|_1 \right)$$

$$\leq \frac{|\mathcal{S}|^{1/2}(|\mathcal{A}| + |\mathcal{B}|)^{1/2}}{(1-\gamma)^2} \left[ \sum_{s \in \mathcal{S}} \left( \left\| \nu(s') - \nu^\star(s') \right\|_2^2 + \left\| \mu(s') - \mu^\star(s') \right\|_2^2 \right) \right]^{1/2},$$

where the second step results from Cauchy-Schwarz inequality. Finally, recall from Wei et al. (2021b, Theorem 2) that it takes at most

$$\mathcal{O}\left( \frac{|\mathcal{S}|^2}{\eta^4 c^4 (1-\gamma)^4 \epsilon^2} \right)$$

iterations to ensure

$$\frac{1}{|\mathcal{S}|} \sum_{s \in \mathcal{S}} \left( \left\| \nu(s') - \nu^\star(s') \right\|_2^2 + \left\| \mu(s') - \mu^\star(s') \right\|_2^2 \right) \leq \epsilon^2,$$

with $\eta^2 = \mathcal{O}((1-\gamma)^5 |\mathcal{S}|^{-1})$. Putting pieces together and minimizing the bound over $\eta$, this leads to an iteration complexity of

$$\mathcal{O}\left( \frac{|\mathcal{S}|^5(|\mathcal{A}| + |\mathcal{B}|)^{1/2}}{(1-\gamma)^{16} c^4 \epsilon^2} \right)$$

to achieve $\epsilon$-NE in a last-iterate fashion.

