# OpenReview forum: "Faster Last-iterate Convergence of Policy Optimization in Zero-Sum Markov Games"
_ICLR.cc/2023/Conference — ICLR 2023 poster_

### Official Review · Reviewer_Ki1t · 2022-10-12

**Confidence:** 4
**Correctness:** 4
**Technical Novelty And Significance:** 3
**Empirical Novelty And Significance:** 3
**Recommendation:** 6

**Clarity, Quality, Novelty And Reproducibility:**

The paper doesn't come with an appendix and I can't verify the correctness of the paper.

**Strength And Weaknesses:**

The paper doesn't come with an appendix and I can't verify the correctness of the paper. The discussion about the technical highlight is also very limited. The authors may want to move the episodic case to the appendix and discuss more the proof sketch and technical contribution, including the limits of the existing results. I think it would be super helpful for readers.

**Summary Of The Paper:**

This paper discusses policy optimization in two-player zero-sum Markov games with sharp last iterate guarantee.

**Summary Of The Review:**

This paper discusses policy optimization in two-player zero-sum Markov games with a sharp last iterate guarantee. The paper doesn't come with an appendix and covers very little about the proof techniques in the main text.

---

> ### Author Response · Authors · 2022-11-18
> **Response to Reviewer Ki1t**
>
> We thank the reviewer for the useful feedback! We want to start by mentioning that due to page limit, the proofs are provided in the supplemental material of the submission. We understand the reviewer might have missed it, and now have included in the pdf of the paper directly in the revision. The high-level proof sketches are provided in Appendix A and Appendix B, respectively, for the infinite-horizon and finite-horizon cases. We would greatly appreciate if you can re-evaluate the technical correctness of our paper and consider updating your recommendation accordingly.

---

> > ### Comment · Reviewer_Ki1t · 2022-11-18
> > **Score updated**
> >
> > The new version looks good to me. I think this year's ICLR asked authors to put everything in the main pdf so I didn't realize this is supplementary material.

---

### Official Review · Reviewer_HSt5 · 2022-10-24

**Confidence:** 4
**Correctness:** 3
**Technical Novelty And Significance:** 2
**Empirical Novelty And Significance:** Not applicable
**Recommendation:** 5

**Clarity, Quality, Novelty And Reproducibility:**

Just one minor comment:

The value-update rule used in the submission was first proposed in the following paper instead of Wei et al. (2021).

Y. Bai, C. Jin, and T. Yu. Near-optimal reinforcement learning with self-play. Advances in neural information processing systems, 33:2159–2170, 2020.

**Strength And Weaknesses:**

# Strength

- The iteration complexity improves over the best previous results by a factor of $H$.
- The paper is mostly clearly written.


# Weakness (and questions)

- The proposed algorithm simply combines the entropy-regularized OMWU method (Cen et al., 2021b) with the V-learning style value update rule (Bai et al., 2020). The techniques used in the proofs also look quite similar to the previous ones. So it's a bit unclear to me how much algorithmic and technical novelty this submission possess beyond blending the existing ideas.

- The interaction protocol considered in this paper is very strong, which basically enables running any dynamic-programming style algorithms. In this case, why isn't it a better choice to run DP-style algorithms, which is algorithmically simpler, theoretically easier to analyze, and oftentimes have better iteration complexity.

- The results in this paper seem to highly rely on the gradient feedback being exact to guarantee the linear convergence rate on the strongly convexified problem, which makes it hard to generalize to the sample-based setting.





**Summary Of The Paper:**

This paper studies computing Nash equilibria in two-player zero-sum Markov games, by proposing a single-loop and symmetric algorithm with last-iterate convergence guarantee. The derived iteration complexity improves over previous works by a factor of $H$.

**Summary Of The Review:**

Despite the improved iteration complexity, there is still unneglectable ambiguity regarding the rationality of the setting considered and the novelty of the algorithm design and analysis. I am willing to raise the score if the above issues are well addressed.

---

> ### Author Response · Authors · 2022-11-18
> **Response to Reviewer HSt5 (2/2)**
>
>
> > The results in this paper seem to highly rely on the gradient feedback being exact to guarantee the linear convergence rate on the strongly convexified problem, which makes it hard to generalize to the sample-based setting.
>
> Thanks for the comment! It is possible to generalize our analysis to the bandit setting by incorporating error analysis w.r.t. approximate policy evaluations, which leads to a last-iterate convergence result with some error floor determined by the approximation error. **In Appendix F of the revised paper, we have included additional results** that characterize the convergence of the proposed method with approximate value updates in the infinite-horizon setting (which is arguably more challenging than the finite-horizon setting). The approximation results allow one to use efficient sampling protocols to conduct inexact policy evaluation, which can in turn lead to sample-based results.
>
> We would also like to clarify that introducing entropy regularization does not necessarily make the problem strongly-convex strongly-concave, especially with a small regularization parameter $\tau$ typically considered in this work.
>
> > The value-update rule used in the submission was first proposed in the following paper instead of Wei et al. (2021).
> > Y. Bai, C. Jin, and T. Yu. Near-optimal reinforcement learning with self-play. Advances in neural information processing systems, 33:2159–2170, 2020.
>
> Thanks for the comment! We have updated our references and modified our statements accordingly.

---

> ### Author Response · Authors · 2022-11-18
> **Response to Reviewer HSt5 (1/2)**
>
> > - The iteration complexity improves over the best previous results by a factor of $H$.
> > - The paper is mostly clearly written.
>
>
> We thank the reviewer's positive assessment of our paper! We would like to remind that our work also improves upon the best existing results in the infinite-horizon setting from $\widetilde{O}(1/\epsilon^2)$ to $\widetilde{O}(1/\epsilon)$, along with better dependency on other problem parameters. In fact, the infinite-horizon setting is more challenging than the finite-horizon one, and significant more efforts are necessary to lead to this improvement.
>
>
> > The proposed algorithm simply combines the entropy-regularized OMWU method (Cen et al., 2021b) with the V-learning style value update rule (Bai et al., 2020). The techniques used in the proofs also look quite similar to the previous ones. So it's a bit unclear to me how much algorithmic and technical novelty this submission possess beyond blending the existing ideas.
>
> Thanks for the comment!
>
> **Algorithmic Novely:** We agree that it is a natural idea to solve two-player zero-sum Markov games with OMWU-type method in combination with critic update rules. That being said, many aspects remain unclear, e.g., (i) whether or not this combination will ever lead to provable convergence, (ii) proper choices of learning rates, and (iii) possible differences between the finite-horizon setting and the infinite-horizon setting. Therefore, we believe that it is equally important to show how simple and natural algorithm design can lead to reliable convergence.
>
> **Technical Novelty:** We would like to point out that the techniques in the work of (Cen et al., 2021b) focus exlusively on the matrix game setting, and generalize to the Markov setting by leveraging contraction properties of value iterations.
> - While Lemma 1 and Lemma 9 in this work take inspiration from (Cen et al., 2021b), the remaining proof tackling instability and approximation errors w.r.t $Q$ (e.g., Lemma 2,3,4) separate itself from the prior work of (Cen et al., 2021b) by directly characterizing the interaction between policy updates and value updates --- which are treated in a decoupled manner in (Cen et al., 2021b).
> - In addition, we adopt a novel decomposition in Lemma 1 that yields a bonus term $\mathsf{KL}\_\rho(\bar{\zeta}^{(t+1)}\\,\\|\\,\bar{\zeta}^{(t)})$ instead of $\mathsf{KL}\_\rho(\bar{\zeta}^{(t+1)}\\,\\|\\,{\zeta}^{(t)})$ in prior literatures, which is necessary for the proof.
>
> Regarding the value update rule, we make note that most previous literature have adopted the linear rescaled learning rate $\alpha\_t = \frac{H+1}{H+t}$, while our analysis shows it is possible to set a constant learning rate $\alpha\_t = \eta\tau$ and hence leads to faster convergence. This is enabled by the introduction of regularization as well as a per-state geometrically weighted regret bound (see proof for Lemma 3).
>
> > The interaction protocol considered in this paper is very strong, which basically enables running any dynamic-programming style algorithms. In this case, why isn't it a better choice to run DP-style algorithms, which is algorithmically simpler, theoretically easier to analyze, and oftentimes have better iteration complexity.
>
> Thanks for the comment! We would like to clarify that the interaction protocol (i.e., assuming exact policy evaluation) in this work has been widely used in prior policy optimization literature, see e.g., (Agarwal et al, 2020; Wei et al, 2021), with the hope of shedding light to the understanding of the optimization dynamic of policy optimization methods by setting aside the statistical considerations.
>
> From either agent's perspective, the protocal in use is on par with the typical actor-critic method for single-agent RL. We agree with the reviewer that DP-style algorithms are theoretically easier to analyze, and oftentimes have better iteration complexity --- but not necessarily algorithmically simpler especially in the game setting (for example, the periodic policy initialization due to nested-loop design, the widely-used trick of introducing two separate evaluations of $Q$ for upper bound and lower bound estimates, etc). In fact, it is the simple algorithm design and the easy implementation that make policy optimization methods especially appealing in the practice, which motivate our study of their theoretical properties.

---

> ### Author Response · Authors · 2022-12-02
> **Response to Reviewer HSt5: have we addressed your concerns?**
>
> We are eager to find out if our response and the revised paper have successfully addressed your concerns raised in the review. Please engage with us, and let us know how we can clarify further to increase your perception of our work to recommendation. Thank you in advance!

---

### Official Review · Reviewer_1xjS · 2022-10-24

**Confidence:** 3
**Correctness:** 3
**Technical Novelty And Significance:** 3
**Empirical Novelty And Significance:** Not applicable
**Recommendation:** 6

**Clarity, Quality, Novelty And Reproducibility:**

**Quality**: The theoretical results are strong, and the technical lemmas seem correct to me as I checked. The quality of the work is good. A missing part could be the simulation results to verify the theory as mentioned in the weaknesses.

**Clarity**: In general, I found the presentation clear and easy to follow, including the problems and settings, related work discussions, and comparisons to existing results. The clarify is good. A missing part is the explanation for difficulties of using larger learning rates could be better presented and clarified.

**Originality**: The policy update is from entropy-regularized OMWU Cen2021b as noted, and the critic update Eq. (11) is motivated from Wei2021 as also mentioned. Generalizing from matrix games to Markov games and showing that combining these two updates can work seem non-trivial to me. The technical originality is incremental but seems enough.

**Strength And Weaknesses:**

**Strenghs**:

1. This work studied an important problem of finding equilibria in two-player zero-sum Markov games. The presentation is clear and easy to follow. Existing related works are also discussed well enough.

2. The results obtained are strong and promising, achieving the same iteration complexity of $\tilde{O}\left( \frac{1}{\epsilon} \right)$ as for PI- and VI-based methods while using only single-loop updates.

3. The techniques are also insightful. It shows that the existing contraction techniques used in matrix games in Cen2021b can be generalized to Markov games and can be combined with critic updates to work, which seems non-trivial and novel to my knowledge.

**Weaknesses**:

I have some questions after reading the paper.

1. If $\eta \le \frac{(1 - \gamma)^3 }{32000 |\mathcal{S}| } $, then the linear rate also seems to be very small in Eq. (12a) since $\eta$ also appears there. The authors did mention that $|\mathcal{S}|$ can be replaced with concentration coefficient in Remark 1 and Appendix A, but how smaller that replacement could be is still unclear from the paper. Could you elaborate more on what in the current analysis makes the learning rate $\eta$ have to be very small in Theorem 1? It seems to me bounding the state distribution in $V$ is the bottleneck (Theorem 2 in Appendix A).

2. It would be great to use some simulation results to verify the theoretical results.

    - It would be curious to verify and observe the linear rate itself (in terms of $t$), as well as how slow it could be (in terms of $\eta$);
    - It would be also interesting to see if in practice larger learning rates than $\eta \le \frac{(1 - \gamma)^3 }{32000 |\mathcal{S}| } $ could be used and still achieve convergence, which means the analysis has space for improvement; or larger learning rates lead to non-convergence, which means it has to be what is used in Theorem 1 (in this case, I think the authors should do better jobs in explaining the difficulties of using large learning rates here).

3. I did not find any assumption of unique NE in the draft (the authors did mention that other works require unique NE assumptions, and this seems to say that they do not need this kind of assumption). Could you comment on what is the quality of NE for the convergence results in the paragraph below Theorem 1 ("Last-iterate convergence to $\epsilon$-optimal NE")?


**Summary Of The Paper:**

The work "Faster Last-iterate Convergence of Policy Optimization in Zero-Sum Markov Games" studies two-player zero-sum Markov games, where the goals are to find the Nash Equilibrium (NE) or Quantal Response Equilibrium (QRE).

The authors proposed actor-critic methods (two similar variants, one for discounted and one for episodic settings), where the actor update is the entropy-regularized optimistic multiplicative weights update (OMWU) method in Cen2021b, and the critic update is slower (as shown in Eq. (10) in Algorithm 1). The proposed algorithms are single-loop, symmetric, with finite-time last-iterate convergence toward equilibria.

In particular, Theorem 1 shows that Algorithm 1 enjoys linear convergence toward QRE, and by setting the temperature $\tau$ for the entropy the algorithm can find $\epsilon$-NE with $\tilde{O}\left( \frac{ |\mathcal{S}| }{(1 - \gamma)^5 \epsilon} \right)$ iteration complexity.

The above results outperform existing works as summarized in Tables 1 and 2, achieving the same desirable iteration complexity of $\tilde{O}\left( \frac{1}{\epsilon} \right)$ as in PI- and VI-based methods while using only single-loop updates.

**Summary Of The Review:**

Overall, this work studies an important problem and the theoretical results are strong. The techniques are from or motivated by prior works, but extending those results from matrix games to Markov games requires non-trivial work. The only missing part is lack of empirical supports.


~~~~~after rebuttal~~~~~

Thank you for the comments. I would like to maintain my current score.

---

> ### Author Response · Authors · 2022-11-18
> **Response to Reviewer 1xjS**
>
>
> Thanks for the positive assessment of our paper!
>
> > If $\eta \le \frac{(1-\gamma)^3}{32000|\mathcal{S}|}$ , then the linear rate also seems to be very small in Eq. (12a) since $\eta$ appears there. The authors did mention that $|\mathcal{S}|$ can be replaced with concentration coefficient in Remark 1 and Appendix A, but how smaller that replacement could be is still unclear from the paper. Could you elaborate more on what in the current analysis makes the learning rate have to be very small in Theorem 1? It seems to me bounding the state distribution in $V$ is the bottleneck (Theorem 2 in Appendix A).
>
> Thanks for the comment!
>
> The main reason for adopting a small learning rate $\eta$ in the infinite-horizon case is that we need to cancel the error/instability terms w.r.t. $Q^{(t)}$ on the RHS of Lemma 1 with the "bonus" terms $\\{\mathsf{KL}\_\rho(\bar{\zeta}^{(t+1)}\\,\\|\\,\bar{\zeta}^{(t)})\\}\_{t=0,\cdots}$ on the LHS. Further decomposition of these error terms reveals dependency on $\\{\eta\mathsf{KL}\_{\chi^{(l)}}(\bar{\zeta}^{(l+1)}\\,\\|\\,\bar{\zeta}^{(l)})\\}\_{l=0,\cdots}$ where $\chi^{(l)}$ is a state distribution determined by the error propagation (see proof for Lemma 2 for details). It is hence necesary to use a small $\eta$ to control $\\{\eta\mathsf{KL}\_{\chi^{(l)}}(\bar{\zeta}^{(l+1)}\\,\\|\\,\bar{\zeta}^{(l)})\\}\_{t=0,\cdots}$ with $\\{\mathsf{KL}\_\rho(\bar{\zeta}^{(t+1)}\\,\\|\\,\bar{\zeta}^{(t)})\\}\_{l=0,\cdots}$.
>
> We would like to further point out that this technical issue is no longer present in the finite-horizon setting: the convergence of policy pair $(\mu\_H, \nu\_H)$ is guaranteed with large learning rates as $Q\_H^{(t)}$ being fixed in the last step, which consequently leads to convergence of $Q\_{H-1}^{(t)}$ --- and ultimately the convergence of $\\{\mu\_h\\}\_{h=1}^H$ and $\\{\nu\_h\\}\_{h=1}^H$ as a whole. This implies that we can expect a larger learning rate in the infinite-horizon setting as well by assuming the existence of some absorbing states, which leads to a convergence rate depending on the reachablility of these states. Assumptions of this kind, however, are less common in the policy optimization literature and hence we refrain from digression into that discussion.
>
> > It would be great to use some simulation results to verify the theoretical results.
> > - It would be curious to verify and observe the linear rate itself (in terms of $t$), as well as how slow it could be (in terms of $\eta$);
> > - It would be also interesting to see if in practice larger learning rates than $\eta \le \frac{(1-\gamma)^3}{32000|\mathcal{S}|}$ could be used and still achieve convergence, which means the analysis has space for improvement; or larger learning rates lead to non-convergence, which means it has to be what is used in Theorem 1 (in this case, I think the authors should do better jobs in explaining the difficulties of using large learning rates here).
>
> Thanks for the suggestion! We plan to include numerical simulations for proof-of-concept purpose in the next revision.
>
> > I did not find any assumption of unique NE in the draft (the authors did mention that other works require unique NE assumptions, and this seems to say that they do not need this kind of assumption). Could you comment on what is the quality of NE for the convergence results in the paragraph below Theorem 1 ("Last-iterate convergence to $\epsilon$-optimal NE")?
>
> Thanks for the comment! The reviewer is correct that our analysis is not built upon the unique NE assumption (some prior analysis on matrix games using OMWU, e.g. (Daskalakis and Panageas (2018); Wei et al. (2021)), requires the uniqueness assumption). Intuitively speaking, we expect the QRE to converge to the NE with more randomness among the set of all NEs when $\tau \to 0$. That being said, we are not aware of any explicit characterization of this limiting behavior, which appears to be one of the long-standing problems in the game theory literature.

---

### Official Review · Reviewer_mJWC · 2022-10-27

**Confidence:** 2
**Correctness:** 4
**Technical Novelty And Significance:** 3
**Empirical Novelty And Significance:** Not applicable
**Recommendation:** 8

**Clarity, Quality, Novelty And Reproducibility:**

This paper is overall clear to readers. It also has significant novelty. The detailed proof in the Appendix can reproduce the main results in the paper.

**Strength And Weaknesses:**

Strength: This paper presents a novel algorithm for policy optimization in zero-sum Markov games, which can lead to an improvement over the existing rate. Moreover, the proposed algorithm is single-loop and symmetric, which are good properties for an algorithm. The overall algorithm design seems to be inspired by the optimistic mirror descent algorithm. But given that the algorithm and the corresponding result are novel to the research area of policy optimization in multi-agent RL, the contribution of this paper is significant.

I only have one question regarding this paper: is it possible to extend this algorithm to a more realistic bandit setting, where both the reward and the transition are unknown to the learner? If it is not, what is the major challenge?

**Summary Of The Paper:**

This paper focuses on two-player zero-sum Markov games and studies equilibrium finding algorithms in the infinite-horizon discounted setting and the finite-horizon episodic setting. The authors propose a novel single-loop policy optimization method with symmetric updates from both agents. They show the proposed method achieves a sublinear last-iterate convergence to the Nash equilibrium by controlling the amount of regularization in the full-information tabular setting. Their results improve upon the best-known iteration complexities.

**Summary Of The Review:**

See the Strength And Weaknesses section.

---

> ### Author Response · Authors · 2022-11-18
> **Response to Reviewer mJWC**
>
> We thank the reviewer for the positive assessment of our paper!
>
> > Is it possible to extend this algorithm to a more realistic bandit setting, where both the reward and the transition are unknown to the learner? If it is not, what is the major challenge?
>
> It is possible to generalize our analysis to the bandit setting by incorporating error analysis w.r.t. approximate policy evaluations, which leads to a last-iterate convergence result with some error floor determined by the approximation error. **In Appendix F of the revised paper, we have included additional results** that characterize the convergence of the proposed method with approximate value updates in the infinite-horizon setting (which is arguably more challenging than the finite-horizon setting). The approximation results allow one to use efficient sampling protocols to conduct inexact policy evaluation, which can in turn lead to sample-based results.

---

### Decision · Program_Chairs · 2023-01-20

**Decision:**

Accept: poster

**Justification For Why Not Higher Score:**

NA

**Justification For Why Not Lower Score:**

NA

**Metareview: Summary, Strengths And Weaknesses:**

This work studies two player zero-sum Markov games, and develops a policy mirror ascent-type algorithm. Convergence analysis is conducted, which establishes the resultant algorithm achieves a Nash equilibrium depending on the amount of regularization.

The reviewers were nearly unanimous in confirming the merits of the work relative to prior art.

Weaknesses of the paper include a lack of experimental validation, and an ablation study of the technical conditions required for convergence in practice. Nonetheless, I recommend this paper be accepted.

**Note From Pc:**

if the above contains the word "oral" or "spotlight" please see: "oral" presentation means -> notable-top-5% and "spotlight" means -> notable-top-25%. As stated in our emails, we are disassociating presentation type from AC recommendations